# A solvable high-dimensional model where nonlinear autoencoders learn structure invisible to PCA while test loss misaligns with generalization

Vicente Conde Mendes [* 1]   Lorenzo Bardone [* 1]   Cédric Koller [* 1]   Jorge Medina Moreira [* 1]   Vittorio Erba [1]
Emanuele Troiani [1]   Lenka Zdeborová [1]

## Abstract

Many real-world datasets contain hidden structure that cannot be detected by simple linear correlations between input features. For example, latent factors may influence the data in a coordinated way, even though their effect is invisible to covariance-based methods such as PCA. In practice, nonlinear neural networks often succeed in extracting such hidden structure in unsupervised and self-supervised learning. However, constructing a minimal high-dimensional model where this advantage can be rigorously analyzed has remained an open theoretical challenge.

We introduce a tractable high-dimensional spiked model with two latent factors: one visible to covariance, and one statistically dependent yet uncorrelated, appearing only in higher-order moments. PCA and linear autoencoders fail to recover the latter, while a minimal nonlinear autoencoder provably extracts both. We analyze both the population risk and empirical risk minimization. Our model also provides a tractable example where self-supervised test loss is poorly aligned with representation quality: nonlinear autoencoders recover latent structure that linear methods miss, even though their reconstruction loss is higher.

## 1. Introduction

Understanding how high-dimensional unsupervised learning systems extract latent structure from data has become a central theme in theoretical machine learning. Over the

*Equal contribution [1]Statistical Physics of Computation Laboratory, EPFL, Lausanne, Switzerland. Correspondence to: Lorenzo Bardone <lorenzo.bardone@epfl.ch>, Jorge Medina Moreira <jorge.medinamoreira@epfl.ch>.

*Proceedings of the 43rd International Conference on Machine Learning*, Seoul, South Korea. PMLR 306, 2026. Copyright 2026 by the author(s).
    Code available at: https://github.com/SPOC-group/advantage_nonlinearity

past decades, the spiked covariance model and its variants (Baik et al., 2005; Johnstone & Lu, 2009; Donoho et al., 2018; Deshpande & Montanari, 2014; Lelarge & Miolane, 2017) have served as a fertile theoretical testbed for understanding algorithmic and information-theoretic limits of unsupervised learning in high dimensions. In this class of models, nonlinear methods provide little or no improvement over linear correlation-based ones (Krzakala et al., 2013; Deshpande & Montanari, 2015; Bandeira et al., 2016).

Modern unsupervised and self-supervised neural networks, however, routinely extract latent structure that is invisible to linear correlation, see e.g. (Vincent et al., 2008; Song & Ermon, 2019; He et al., 2022). At the same time, our theoretical understanding of how nonlinear architectures exploit higher-order statistical dependence remains limited. Although it is empirically well known that nonlinear models can outperform their linear counterparts and PCA in unsupervised representation learning, establishing a minimal high-dimensional data model where this advantage can be rigorously demonstrated and analyzed has remained elusive. A major reason is the lack of data models on which correlation-based methods such as PCA fail, while simple shallow—and thus analyzable—nonlinear neural networks provably succeed. In this work, we fill this gap by introducing a structured high-dimensional data model where nonlinear representation learners provably outperform linear baselines, while learning remains as tractable to analyze as in the spiked covariance model.

**The spiked cumulant model.**  We are inspired by recent work of (Bardone & Goldt, 2024; Szekely et al., 2024), who introduced the spiked cumulant model in which a dataset of $n$ samples in dimension $d$ is generated as

$$\mathbf{x}_\mu = \lambda_\mu \frac{\mathbf{u}^\star}{\sqrt{d}} + S\left(\nu_\mu \frac{\mathbf{v}^\star}{\sqrt{d}} + \mathbf{z}_\mu\right), \quad \mu = 1, \dots, n, \quad (1)$$

where $\mathbf{z}_\mu, \mathbf{u}^\star, \mathbf{v}^\star \sim \mathcal{N}(0, \mathbb{I}_d)$, and

$$S = \mathbb{I} - \frac{\mathbb{E}[\nu^2]}{1 + \mathbb{E}[\nu^2] + \sqrt{1 + \mathbb{E}[\nu^2]}} \frac{\mathbf{v}^\star \mathbf{v}^{\star\top}}{d}, \quad (2)$$

is a whitening matrix. The latent variables $\lambda_\mu$ and $\nu_\mu$ are drawn from a joint probability distribution $P_{\text{latents}}$ indepen-

dently for each sample, and we denoted by $\mathbb{E}$ the average over this distribution. In the whole work, we consider centered latent variables $\mathbb{E}[\lambda] = \mathbb{E}[\nu] = 0$. Authors of (Bardone & Goldt, 2024; Szekely et al., 2024) studied learning in the case of independent latent variables, where extracting the spike $\mathbf{v}^\star$ from the higher-order correlations is computationally hard and requires large sample complexity. They also considered the case where $\lambda_\mu$ and $\nu_\mu$ are linearly correlated, in which case even a simple PCA efficiently recovers a correlation with both spikes $\mathbf{u}^\star$ and $\mathbf{v}^\star$. This can be seen from the population covariance of the data model Equation (1) and Equation (2), which is given by

$$\mathrm{Cov}(\mathbf{x}|\mathbf{u}^\star, \mathbf{v}^\star) = \mathbb{I} + \mathbb{E}[\lambda^2]\frac{\mathbf{u}^\star \mathbf{u}^{\star\top}}{d}$$
$$+ \sqrt{\mathbb{E}[\lambda^2]\mathbb{E}[\nu^2]}\mathbb{E}[\lambda\nu]\frac{\mathbf{u}^\star \mathbf{v}^{\star\top}S + S\mathbf{v}^\star \mathbf{u}^{\star\top}}{d}. \quad (3)$$

Motivated by realistic data, where strong statistical dependencies often exist but are rarely visible through linear correlations, we will focus on the regime where $\lambda_\mu$ and $\nu_\mu$ are dependent but uncorrelated $\mathbb{E}[\lambda\nu] = 0$. In that case, the covariance Equation (3) includes no information on the spike $\mathbf{v}^\star$ and consequently that spike is invisible to PCA. Unlike in the spiked covariance model, here the hidden spike is invisible to all second-order statistics but, as we will see, remains recoverable through higher-order dependence.

**Definition 1.1** (Correlation exponent). We define the *correlation exponent* $k^\star$ of a distribution $P_{\text{latents}}(\lambda, \nu)$ as the smallest $k \in \mathbb{N}$ such that $\mathbb{E}[\lambda^k \nu] \neq 0$.

In high dimension, the spike $\mathbf{v}^\star$ is invisible to PCA (and thus also to linear autoencoders) whenever the correlation exponent $k^\star \geq 2$. We show in Section 3 that specialized algorithms such as approximate message passing are able to efficiently recover a correlation with both the spikes $\mathbf{u}^\star$ and $\mathbf{v}^\star$ whenever the correlation exponent is finite, $k^\star < \infty$ and the number of samples is larger than $\alpha_c d$ where $\alpha_c = \Theta(1)$ and $d$ is the dimension of the data. We then show in Sections 4 and 5 that the simplest non-linear neural networks, i.e., autoencoders with a single hidden unit and a suitable non-linearity, are also able to recover a correlation with both the spikes whenever $k^\star < \infty$. The first key result of our work can be summarized as follows:

**For the spiked cumulant model with $2 \leq k^\star < \infty$, nonlinear autoencoders learn structure invisible to PCA.**

**The analyzed autoencoder.** Building on the high-dimensional analysis of learning with simple autoencoders on data generated by spiked-covariance models (Refinetti & Goldt, 2022; Cui & Zdeborová, 2023), we study learning with the minimal nonlinear autoencoder architecture

$$\hat{\mathbf{x}}_\mu = \frac{\mathbf{w}}{\sqrt{d}}\sigma\left(\frac{\mathbf{w}^\top \mathbf{x}_\mu}{\sqrt{d}}\right), \quad (4)$$

where $\mathbf{w} \in \mathbb{R}^d$ is the learnable weight vector, $\sigma(\cdot)$ is a fixed nonlinear activation function, and tied encoder-decoder weights are used.

The model is trained by minimizing either the population or the empirical reconstruction loss. At the population level, the objective reads

$$\mathcal{L}(\mathbf{w}) = \mathbb{E}_\mathbf{x}\left\|\mathbf{x} - \frac{\mathbf{w}}{\sqrt{d}}\sigma\left(\frac{\mathbf{w}^\top \mathbf{x}}{\sqrt{d}}\right)\right\|^2, \quad (5)$$

where the expectation is taken over the data-generating distribution. We analyze the population risk in the high-dimensional limit $d \to \infty$ using recent advances for the supervised learning of the single index model using stochastic gradient descent (Ben Arous et al., 2021; Veiga et al., 2022; Ben Arous et al., 2022). This allows us to characterize how the ability to learn depends on the dependence structure between the latent variables $(\lambda, \nu)$ and on the non-linearity.

In practice, learning proceeds by minimizing the empirical risk

$$\mathcal{L}_n(\mathbf{w}) = \frac{1}{n}\sum_{\mu=1}^n \left\|\mathbf{x}_\mu - \frac{\mathbf{w}}{\sqrt{d}}\sigma\left(\frac{\mathbf{w}^\top \mathbf{x}_\mu}{\sqrt{d}}\right)\right\|^2, \quad (6)$$

typically using gradient descent. We analyze the minimum of the empirical risk in the high-dimensional limit $n, d \to \infty$ at sample complexity scaling linearly with the dimension $\alpha = n/d$ using the advances from (Cui & Zdeborová, 2023).

**Validation in self-supervised learning.** A fundamental difficulty in unsupervised and self-supervised learning is the absence of ground-truth labels for evaluating representation quality. In modern self-supervised learning, representation quality is rarely assessed through reconstruction or test loss; instead, it is evaluated almost exclusively via downstream tasks or linear probing (Grill et al., 2020; He et al., 2022; Devlin et al., 2019; Balestriero & Lecun, 2024). This widespread practice reflects an implicit recognition that training or test loss on the self-supervised objective may be poorly aligned with representation quality.

At the same time, training and model selection are still commonly guided by the validation loss of the self-supervised objective, through early stopping and the tuning of regularization and architectural hyperparameters such as weight decay, width–depth trade-offs, or attention head count. This implicitly assumes that lower validation loss correlates with better learned representations. Recent work has shown that decreasing test loss does not necessarily indicate improved generalization: models can enter a regime where test loss improves while representations become increasingly biased toward the training data, making early stopping unreliable (Garnier-Brun et al., 2025). More broadly, the conditions under which loss-based evaluation and training control are

reliable—or fundamentally misleading—in self-supervised learning remain poorly understood.

To understand the interplay between test loss and representation quality, it is useful to have a clear and tractable example of this phenomenon. In Section 6, we show that the spiked cumulant model with correlation exponent $2 \leq k^\star < \infty$ provides such a setting. Linear autoencoders achieve lower reconstruction error on held-out data than nonlinear autoencoders, yet fail to recover the hidden spike. This is expected, since linear autoencoders are equivalent to PCA and are optimal for squared reconstruction error by the Eckart–Young theorem (Eckart & Young, 1936). In contrast, nonlinear autoencoders recover both spikes despite exhibiting higher test loss. The second key result of our work can thus be summarized as follows:

**For the spiked cumulant model with $2 \leq k^\star < \infty$, lower test loss misaligns with better representation quality.**

More broadly, this result highlights the need for evaluation criteria in unsupervised learning that probe latent structure rather than solely measuring reconstruction performance. For the task considered in our paper, we introduce a simple downstream task that allows principled evaluation without direct access to the ground-truth spikes. For a small subset of samples, we assume access to labels depending on the latent representation, for instance $y_\mu = \text{sign}(\mathbf{x}_\mu^\top \mathbf{v}^\star)$. Given a learned representation $\hat{\mathbf{w}}$, prediction on a new sample is performed as $\hat{y}_{\text{new}} = \text{sign}(\mathbf{x}_{\text{new}}^\top \hat{\mathbf{w}})$. In practice, one would fine-tune a model on the labeled data to construct such an estimator. Performance on this downstream task reveals the advantage of nonlinear representation learning in regimes where reconstruction-based evaluation fails.

## 2. Linear autoencoder implements PCA

It is well known that for $\sigma = \text{id}$, minimizing the squared reconstruction loss yields PCA (Bourlard & Kamp, 1988), equivalently the best rank-one approximation in Frobenius norm (Eckart & Young, 1936). Since this case provides a natural baseline for our analysis, we restate the result below in a form tailored to our setting (proof in Appendix A.1).

**Proposition 2.1** (Linear autoencoder yields PCA). *Consider Equation (5) with $\sigma = \text{id}$ and any distribution over $\mathbf{x}$ (even empirical). Then any global minimizer $\hat{\mathbf{w}}$ satisfies $\|\hat{\mathbf{w}}\|^2 = d$ and $\hat{\mathbf{w}}/\sqrt{d}$ is a leading eigenvector of the covariance $\mathbb{E}_{\mathbf{x}}[\mathbf{x}\mathbf{x}^\top]$.*

As a consequence, training a linear autoencoder on either the population loss Equation (5) or the empirical loss Equation (6) yields weights aligned with the leading eigenspace of the corresponding covariance matrix (see Appendix A.1). For our data model, the population covariance is given by Equation (3), whose top eigenvector is $\mathbf{u}^\star$. When $k^\star \geq 2$,

linear autoencoders therefore fail to recover any nontrivial correlation with the hidden direction $\mathbf{v}^\star$. Note that even in case of empirical covariance, we can quantify that with high probability the weights will not have more than random overlap with $\mathbf{v}^\star$, see Lemma A.1. Finally, since the optimal PCA solution is a symmetric rank-one projector, tying encoder and decoder weights incurs no loss of optimality compared to the untied setting (Baldi & Hornik, 1989).

## 3. Information theoretic baseline

In this section we study the Bayes-optimal estimator, gaining insight on the intrinsic statistical and computational difficulty of retrieval of both spikes. We consider the spiked cumulant model Equation (1) in the high-dimensional regime where the number of samples $n$ is linearly proportional to the dimensionality, i.e. $d, n \to \infty$ with fixed ratio $\alpha = n/d$.

**Bayes-optimal estimator.** The Bayes optimal (BO) estimator of the spikes is defined as $\hat{\mathbf{w}}_{\text{BO}}(\{\mathbf{x}_\mu\}_{\mu=1}^n) \in \mathbb{R}^{d \times 2}$ such that $\|\hat{\mathbf{w}}_{\text{BO}}(\{\mathbf{x}_\mu\}_{\mu=1}^n) - [\mathbf{u}^\star, \mathbf{v}^\star]\|_F^2$ is minimized on average over the observed data, where with square bracket we mean the horizontal stack of the two spikes (seen as column vectors). It is a direct consequence of the Bayes formula that such BO estimator is given by the posterior average, i.e. the mean of the probability distribution

$$P_{\text{post}}([\mathbf{u}, \mathbf{v}]|\{\mathbf{x}_\mu\}_{\mu=1}^n) \propto$$
$$P_{\text{data}}(\{\{\mathbf{x}_\mu\}_{\mu=1}^n|[\mathbf{u}, \mathbf{v}])P_{\text{prior}}([\mathbf{u}, \mathbf{v}]), \quad (7)$$

where we denoted by $P_{\text{prior}}$ the distribution of the spikes (isotropic Gaussian in our case), and by $P_{\text{data}}$ the conditional distribution of observing a certain dataset given the spikes. The degree of recovery achieved by the BO estimator, measured through its cosine similarity

$$\theta_s = \frac{|(\hat{\mathbf{w}}_{\text{BO}}^\top)_s \mathbf{s}|}{\|(\hat{\mathbf{w}}_{\text{BO}}^\top)_s\| \|\mathbf{s}\|} \quad (8)$$

with each spike $\mathbf{s} \in \{\mathbf{u}^\star, \mathbf{v}^\star\}$ and $(\hat{\mathbf{w}}_{\text{BO}}^\top)_s$ the first/second column of $(\hat{\mathbf{w}}_{\text{BO}}^\top)$, can be characterized in the high-dimensional limit as follows.

**Result 3.1** (Asymptotic characterization of Bayes optimal estimation). *Consider the spiked cumulant model, defined in Section 1, in the high-dimensional limit $n, d \to \infty$ with $\alpha = n/d$. Call $\hat{\mathbf{w}}_{\text{BO}} \in \mathbb{R}^{d \times 2}$ the mean of Equation (7). Define the order parameters*

$$q^{\text{BO}} = \hat{\mathbf{w}}_{\text{BO}}^\top [\mathbf{u}^\star, \mathbf{v}^\star]/d \in \mathbb{R}^{2 \times 2}. \quad (9)$$

*Then, for $d \to \infty$, the order parameters concentrate onto their mean w.r.t. the data distribution, and their limiting values satisfy the dimension-independent extremization problem*

$$q^{\text{BO}} = \text{argmax}_q \phi^{\text{BO}}(q) \quad (10)$$

*We give the precise form of $\phi^{\text{BO}}$ in Appendix C, Equation (163), and it depends only on $\alpha$ and $P_{\text{latents}}$. Additionally, the cosine similarities concentrate to*

$$\theta_u^{\text{BO}} = \sqrt{q_{11}^{\text{BO}}} \quad and \quad \theta_v^{\text{BO}} = \sqrt{q_{22}^{\text{BO}}}. \tag{11}$$

Result 3.1 is derived in Appendix C using the replica method from statistical physics. This is a well-established method used to obtain a dimension-independent characterization of high-dimensional learning problems which has been applied to the spiked covariance model in (Lesieur et al., 2017; 2015) before being established rigorously for that model (Miolane, 2017).

**Characterization of the weak recovery of $\mathbf{v}^\star$.** Result 3.1 gives the performance of the information-theoretically optimal estimator for the spikes, but gives no insight on whether such estimator is computable with efficient algorithms. In Appendix C, we introduce an Approximate-Message-Passing (AMP) algorithm for this problem. AMP is a class of iterative algorithms that are widely believed (Gamarnik et al., 2022), and in some settings proven (Celentano et al., 2020), to be optimal among large classes of efficient algorithms. Following this conjecture, we consider AMP as a proxy for the optimal efficiently-achievable performance, and characterize under which properties of $P_{\text{latents}}$ it weakly recovers the spike $\mathbf{v}^\star$, i.e. obtains $\theta_v > 0$ in the high-dimensional limit.

**Result 3.2** (Weak recovery threshold of AMP). *Under the same setting of Result 3.1, assume that $\mathbb{E}[\lambda], \mathbb{E}[\nu] = 0$, $\mathbb{E}[\lambda^2], \mathbb{E}[\nu^2] > 0$ and that the correlation exponent $k^\star$ is finite. Then, the AMP algorithm defined in Appendix C achieves non-zero cosine similarity with both spikes for all $\alpha > \alpha_{\text{weak}}^{\text{AMP}}$ with*

$$\alpha_{\text{weak}}^{\text{AMP}} = \frac{f(\mathbb{E}[\lambda^2], \mathbb{E}[\nu^2], \mathbb{E}[\lambda\nu])}{\mathbb{E}[\lambda^2]^2}, \tag{12}$$

*where $f = 1$ for $k^\star \geq 2$, and $f < 1$ for $k^\star = 1$ (its precise expression is given in Appendix C, Equation (179)).*

Most importantly, Result 3.2 uncovers which correlation structure in the data enables the recovery of the hidden spike $\mathbf{v}^\star$, namely a non-null joint moment of the form $\mathbb{E}[\lambda^k \nu]$ for some finite $k$, thus justifying our Definition 1.1. We also stress that the weak recovery of both spikes happens at the same value of sample ratio $\alpha$: as soon as $\mathbf{u}^\star$ is weakly recovered, the statistical dependence between the latents allows to weakly recover $\mathbf{v}^\star$.

The value at which weak-recovery happens is also notable: for $k^\star \geq 2$, it is the same as that of the Baik-Ben Arous-Péché transition (Baik et al., 2005) for the leading eigenvalue of the empirical covariance of the data: only for

$\alpha > \alpha_{\text{weak}}^{\text{AMP}}$ the covariance Equation (3) develops an outlying eigenvalue, with eigenvector correlated with $\mathbf{u}^\star$. For correlation exponent $k^\star = 1$, the additional information in the covariance leads to an improved weak-recovery threshold. In contrast to our single-view setting with dependent but uncorrelated latents, related work studies computational thresholds in multimodal spiked models with correlated modalities (Tabanelli et al., 2025).

Result 3.2 is derived in Appendix C.2 by first arguing that the iterate of AMP can be tracked through a dimension-independent equation for the overlap $q_{\text{AMP}}^t = \hat{\mathbf{w}}_t^\top [\mathbf{u}^\star, \mathbf{v}^\star]$ (the so-called state evolution, see for e.g. (Bayati & Montanari, 2011; Berthier et al., 2020)), and then studying the linear stability of the uninformed initialization $q_{\text{AMP}}^{t=0} = \mathbb{O}_{d \times 2}$, where $\mathbb{O}$ denotes the null matrix.

In Figure 1 we compare the numerical solution of Equation (10) with a run of the AMP algorithm for a specific example of $P_{\text{latents}}$ with correlation exponent 2, obtaining a good match. This suggests that Result 3.2 applies also to the BO estimator, pointing to the absence of statistical-to-computational gaps (Gamarnik et al., 2022) for this specific setting.

We finally remark that Result 3.2 does not explicitly provide negative results in the case of $k^\star = +\infty$. We argue in Appendix C.1 that if the latent variables are independent, AMP does not weakly recover $\mathbf{v}^\star$ for any $n = \mathcal{O}(d)$, as can be expected based on results of (Bardone & Goldt, 2024). The more subtle case of dependent latents with $k^\star = +\infty$ is more elusive, and its analysis is left for future work.

## 4. Autoencoder: population gradient flow

In this section, we study the performance of the nonlinear autoencoder through an analysis of the gradient flow dynamics of the population loss Equation (5), subject to a spherical constraint on $\mathbf{w}$. The key aim is to describe which properties of the activation function $\sigma$ and which dependence structure in $P_{\text{latents}}$ enable efficient learning. To characterize the properties of the learned weights, and in particular whether the weights develop a non-null projection along the spike $\mathbf{v}^\star$, we follow the approach from (Bardone & Goldt, 2024; Ricci et al., 2025) and rewrite Equation (5) as

$$\mathcal{L}(\mathbf{w}) = \mathop{\mathbb{E}}_{\mathbf{x} \sim \mathcal{N}(0, \mathbb{I})} \left[ L(\mathbf{x}) \left\| \mathbf{x} - \frac{\mathbf{w}}{\sqrt{d}} \sigma\left(\frac{\mathbf{w}^\top \mathbf{x}}{\sqrt{d}}\right) \right\|^2 \right] \tag{13}$$

in which we introduced the likelihood ratio $L(\mathbf{x}) = \frac{\mathrm{d}\mathbb{P}_x}{\mathrm{d}\mathcal{N}(0,\text{id})}(\mathbf{x})$, where $\mathrm{d}\mathbb{P}_x$ is the probability density of a variable that is distributed according the *spiked cumulant model* defined in Section 1, with latent variables distributed according to $P_{\text{latents}}(\lambda, \nu)$. Then, we decompose in Hermite basis (see A.2 for details) both $L$ and an effective nonlinearity that results from Equation (13), namely the function

| Activation $\sigma$: | **Linear** $z$ | **Quadratic** $z^2$ | **ReLU** | **tanh** | **ELU** | **Swish** | **Sigmoid** | **GELU** |
|---|---|---|---|---|---|---|---|---|
| Recovery of $\mathbf{u}^\star$ | ✓₀ | ✗₂ | ✓₁ | ✓₂ | ✓₁ | ✓₁ | ✓₁ | ✓₁ |
| Recovery of $\mathbf{v}^\star$ if $k^\star = 2$ | ✗₀ | ✗₂ | ✓₁ | ✗₁ | ✓₁ | ✓₁ | ✓₁ | ✓₁ |
| Recovery of $\mathbf{v}^\star$ if $k^\star = 3$ | ✗₀ | ✗₂ | ✗₁ | ✓₂ | ✓₂ | ✓₂ | ✓₂ | ✓₂ |

*Table 1.* **Classification of activation functions by success or failure in weakly recovering the spikes $\mathbf{u}^\star$ and $\mathbf{v}^\star$ in logarithmic time.**
Symbols ✓₀ and ✗₀ follow from Proposition 2.1, while all other symbols correspond to the predictions of Theorem 4.2. All cases except ✗₁ hold for any latent law $P_{\text{latents}}$ with the specified $k^\star$; ✗₁ indicates the existence of counterexamples, i.e. latent distributions satisfying the assumptions of the third item of Theorem 4.2 which implies that $\mathbf{v}^\star$ is not weakly recovered (see Appendix A.4 for more details).

$z \to \sigma^2(z) - 2z\sigma(z)$. This leads to the following lemma.

**Lemma 4.1** (Hermite expansion of the population loss).
*Let $\mathcal{L}(\mathbf{w})$ be the population risk Equation (13). Assume the spherical constraint $\|\mathbf{w}\| = \sqrt{d}$, and let $\mathbf{u}^\star, \mathbf{v}^\star \in \mathbb{R}^d$ satisfy $\|\mathbf{u}^\star\| = \|\mathbf{v}^\star\| = \sqrt{d}$ and $\mathbf{u}^\star \perp \mathbf{v}^\star$. Consider the likelihood ratio, which depends only on two projections linked by the distribution of the latent variables $P_{\text{latents}}$*

$$L(\mathbf{x}) = \tilde{L}\left(\frac{\mathbf{x}^\top \mathbf{u}^\star}{\sqrt{d}}, \frac{\mathbf{x}^\top \mathbf{v}^\star}{\sqrt{d}}\right), \qquad \tilde{L} \in \mathcal{L}^2(\mathcal{N}(0, \mathbb{I}_2)), \tag{14}$$

*and assume that $\sigma(\cdot)^2 \in \mathcal{L}^2(\mathcal{N}(0,1))$ and $(z \mapsto z\sigma(z)) \in \mathcal{L}^2(\mathcal{N}(0,1))$, where $\mathcal{L}^2(\gamma)$ denotes the space of square integrable functions with scalar product weighted by the measure $\gamma$. Define overlaps*

$$m_u := \frac{\mathbf{w}^\top \mathbf{u}^\star}{d}, \qquad m_v := \frac{\mathbf{w}^\top \mathbf{v}^\star}{d}, \tag{15}$$

*and Hermite coefficients (probabilists' Hermite polynomials are denoted with $\text{He}_k$)*

$$c_{i,j}^L := \mathop{\mathbb{E}}_{(Z_1, Z_2) \sim \mathcal{N}(0, \mathbb{I}_2)} \left[ \text{He}_i(Z_1) \text{He}_j(Z_2) \tilde{L}(Z_1, Z_2) \right],$$
$$c_k^{\sigma^2} := \mathop{\mathbb{E}}_{Z \sim \mathcal{N}(0,1)} \left[ \sigma(Z)^2 \text{He}_k(Z) \right],$$
$$c_k^{z\sigma} := \mathop{\mathbb{E}}_{Z \sim \mathcal{N}(0,1)} \left[ Z\sigma(Z) \text{He}_k(Z) \right].$$

*Then $\mathcal{L}(\mathbf{w})$ depends on $\mathbf{w}$ only through $(m_u, m_v)$ and admits the expansion*

$$\mathcal{L}(\mathbf{w}) = C + \sum_{k \geq 0} \sum_{i=0}^{k} \frac{c_{i,k-i}^L}{i!(k-i)!} \left( c_k^{\sigma^2} - 2c_k^{z\sigma} \right) m_u^i m_v^{k-i}, \tag{16}$$

*where the constant is*

$$C = \mathop{\mathbb{E}}_{\mathbf{x} \sim \mathcal{N}(0, \mathbb{I}_d)} \left[ L(\mathbf{x}) \|\mathbf{x}\|^2 \right] = \mathbb{E}_{\mathbf{x}} \|\mathbf{x}\|^2. \tag{17}$$

*Moreover, for any $r < 1$ the series in Equation (16) converges absolutely and uniformly on $\{(m_u, m_v) : |m_u|, |m_v| \leq r\}$.*

The proof can be found in Appendix A.2. Lemma 4.1 gives an expansion of the population loss that allows to interpret early stage dynamics of spherical gradient flow, leading to an understanding on which conditions on the activation $\sigma$ and the latent variables distribution $P_{\text{latents}}$ allow for recovery of a positive correlation with $\mathbf{v}^\star$. Note that the Hermite coefficients of the likelihood ratio depend on $P_{\text{latents}}$ in the following way:

$$c_{i,j}^L \propto \mathbb{E}\left[ \text{He}_i(\lambda + Z_u) \text{He}_j\left(\eta\nu + \sqrt{1 - \eta^2} Z_v\right) \right], \tag{18}$$

where $\eta = 1/\sqrt{1 + \mathbb{E}[\nu^2]}$ and the averages are taken over $P_{\text{latents}}(\lambda, \nu)$ and two independent standard Gaussian $Z_u, Z_v$. Our assumption on centered and uncorrelated latents, together with the whitening of spike $v$ imply the following:

$$c_{1,0}^L = c_{0,1}^L = c_{1,1}^L = c_{0,2}^L = 0. \tag{19}$$

**Theorem 4.2** (Gradient flow dynamics on population loss).
*Denote $\tilde{\mathbf{w}} := \mathbf{w}/\sqrt{d}$. Suppose that $\mathcal{L}$ is continuously differentiable 4 times and satisfies Lemma 4.1. Consider spherical gradient flow dynamics on the population loss, i.e. the following dynamical system:*

$$\frac{d}{dt} \tilde{\mathbf{w}}(t) = -\nabla_{sph} \mathcal{L}(\tilde{\mathbf{w}}) = -\left( id - \tilde{\mathbf{w}}\tilde{\mathbf{w}}^\top \right) \nabla \mathcal{L}(\tilde{\mathbf{w}}) \tag{20}$$

*starting from initialization $\tilde{\mathbf{w}}(0) \sim Unif(\mathbb{S}^{d-1})$. Then:*

✓₁ *Assume that $C_2 := c_{2,0}^L \left( c_2^{\sigma^2} - 2c_2^{z\sigma} \right) < 0$ and $C_3 := c_{2,1}^L \left( c_3^{\sigma^2} - 2c_3^{z\sigma} \right) \neq 0$. Then, with high probability in the high dimensional limit $d \to \infty$, gradient flow dynamics reaches **weak recovery** of $\mathbf{u}^\star$ and $\mathbf{v}^\star$ in logarithmic time, i.e., for any $\delta > 0$ there exist $\epsilon_1, \epsilon_2 > 0$, independent of $d$ and $T = \Theta(\log d)$ such that the probability that $|m_u(T)| \geq \epsilon_1, |m_v(T)| \geq \epsilon_2$ is larger than $1 - \delta$ for $d$ large enough.*

✓₂ *If $C_2 < 0$ but $C_3 = 0$ the dynamics still leads to a **weak recovery** of $\mathbf{u}^\star$ and $\mathbf{v}^\star$ in logarithmic time if $C_{(3,1)} := c_{3,1}^L(c_4^{\sigma^2} - 2c_4^{z\sigma}) \neq 0$.*

**✗₁** *If $C_2 < 0$ but $C_{p,1} := c_{p,1}^L(c_{p+1}^{\sigma^2} - 2c_{p+1}^{z\sigma}) = 0$ for all $p \in \mathbb{N}$, and $C_{2,2} := c_{2,2}^L(c_4^{\sigma^2} - 2c_4^{z\sigma}) > 2C_2$, then early stage dynamics will lead to the **weak recovery only of the $\mathbf{u}^\star$ spike** in logarithmic time $T = \Theta(\log d)$: for any $\delta_1, \delta_2 > 0$ there exist $\epsilon_1 > 0$, independent of $d$ and $T = \Theta(\log d)$ such that the probability that $|m_u(T)| \geq \epsilon_1, |m_v(T)| \leq d^{-1/2+\delta_2}$ is larger than $1 - \delta_1$ for $d$ large enough.*

**✗₂** *If instead $C_2 \geq 0$ then, with high probability in the high dimensional limit $d \to \infty$, gradient flow dynamics **does not reach weak recovery of $\mathbf{u}^\star$ and $\mathbf{v}^\star$ for any time that scales as $o(\sqrt{d})$**, i.e. we show that for any $t_d = o(\sqrt{d})$ there exists a sequence $(A_d)_{d\in\mathbb{N}}$ such that $\lim_{d\to\infty} A_d = 0$ and $|m_u(t)| \leq A_d, |m_v(t)| \leq A_d$ for $t \in [0, t_d]$ for all $d$.*

The complete proof is given in the Appendix A.3, and in the following we sketch the main idea.

*Idea of the proof.* By Lemma 4.1 we know that $\mathcal{L}$ depends only on $m_u$, $m_v$, so the high dimensional gradient flow can be reduced to a system of two coupled ODEs that describe the evolution in time of $(m_u, m_v)$. Moreover, since $\tilde{\mathbf{w}}(0) \sim \text{Unif}(\mathbb{S}^{d-1})$, $m_u(0), m_v(0) \approx 1/\sqrt{d}$ with high probability. So, close to initialization, the dynamics will be driven by the low order terms in Equation (16). Since we have Equation (19), the leading order term for $m_u$ dynamics comes from the quadratic term in the loss

$$\dot{m}_u(t) \approx -C_2 m_u(t) \tag{21}$$

hence if $C_2 < 0$ we have that $m_u$ grows exponentially and in logarithmic time will reach threshold $\epsilon$ (which needs to be chosen small enough to ensure that the expansion of the loss works). Regarding the ODE for $m_v$ we have the following approximation:

$$\dot{m}_v(t) \approx -C_3 m_u^2(t) - C_{(3,1)} m_u^3(t) \\ + \left( C_2 - \frac{C_{(2,2)}}{2} \right) m_u^2(t) m_v(t) \tag{22}$$

where the term $C_2 m_u^2 m_v$ comes from the spherical constraint $-\tilde{\mathbf{w}}\tilde{\mathbf{w}}^\top$ in Equation (20). We then prove that if one among $C_3$ and $C_{(3,1)}$ is nonzero, then $m_v$ will grow (or decrease, depending on the sign) almost as fast as $m_u$, reaching weak recovery in logarithmic time. On the other hand, if they are both 0 (and we need to assume no other term of the form $m_u^p$ to be present), then the dynamics are driven by $(C_2 - C_{(2,2)}/2)m_u^2(t)m_v(t)$ which will be attracting towards 0 if the coefficient $C_2 - C_{(2,2)}/2 < 0$. Finally if $C_2 \geq 0$, then $m_u$ does not have the drive to grow exponentially, hence both $m_u$ and $m_v$ need to take longer to reach weak recovery. □

**Connection with the correlation exponent.** Theorem 4.2 can be interpreted through the lens of the *correlation exponent* via the identity $c_{p,1}^L = \mathbb{E}[\lambda^p \nu]$, which is proved in Lemma A.3 in the appendix. This identity implies that cases ✓₁ and ✓₂ establish weak recovery of $\mathbf{u}^\star$ and $\mathbf{v}^\star$ when $k^\star = 2, 3$, whereas case ✗₁ shows that if $k^\star = \infty$, gradient flow on the population loss cannot reach weak recovery of $\mathbf{v}^\star$ on the same time scale as for $\mathbf{u}^\star$ (we remark that this holds for any $\sigma$, and given $C_{2,2} > 2C_2$). Finally, case ✗₂ shows that the condition $C_2 < 0$ is not merely technical, but a fundamental requirement: without it, gradient flow fails to achieve rapid weak recovery of either $\mathbf{u}^\star$ or $\mathbf{v}^\star$. In Table 1, we illustrate the prediction that Theorem 4.2 leads to on several activation functions that are commonly employed in practice. Note that some of the activations in Table 1 are not guaranteed to satisfy the regularity assumptions of Theorem 4.2, so the prediction applies to smooth approximations that share the first four Hermite coefficients as discussed in Appendix A.4. We refer to Appendix E.3 and Appendix E.4 for empirical simulations of the activations listed in Table 1 in the cases $k^\star = 2, 3$.

**Predictions for online SGD sample complexity.** The gradient-flow time scales identified in Theorem 4.2 can be heuristically translated into predictions for the sample complexity of *online SGD* as in the setting of (Ben Arous et al., 2021). For the number of samples $n$, the dimension $d$ and the gradient-flow time to weak recovery $T$, the correspondence is $n^\star \approx Td$. A formal justification of this correspondence would require controlling the stochastic noise in online SGD, which we leave for future work.

Under this correspondence, we predict that cases ✓₁ and ✓₂ yield weak recovery of both $\mathbf{u}^\star$ and $\mathbf{v}^\star$ after $n^\star = \Theta(d \log d)$ samples; case ✗₁ yields weak recovery of $\mathbf{u}^\star$ but not $\mathbf{v}^\star$ at the same scale; and case ✗₂, when $C_2 \geq 0$, requires at least $n \gtrsim d^{3/2}$ samples for weak recovery. These predictions are consistent with recent results on online SGD dynamics (Bardone & Goldt, 2024).

**Remark on the spherical constraint.** Theorem 4.2 and Table 1 describe gradient flow dynamics constrained to the sphere $\|\mathbf{w}\| = \sqrt{d}$. While Theorem 4.2 can be adapted to other radii of the same scale $\|\mathbf{w}\| = r\sqrt{d}$ the relevant Hermite coefficients may differ, potentially altering the classification in Table 1 (see Appendix A.4). For this reason, the ✓ cases in Table 1 are stated specifically for the radius $\|\mathbf{w}\| = \sqrt{d}$: in general, they can be invalidated by constraining the dynamics to sufficiently large radii. Our positive results therefore establish the *existence* of a radius leading to weak recovery (which holds for any $P_{\text{latents}}$ with the specified $k^\star$ value). In contrast, each ✗ case in the Table is driven by more fundamental obstructions and does not achieve weak recovery for any choice of the radius.

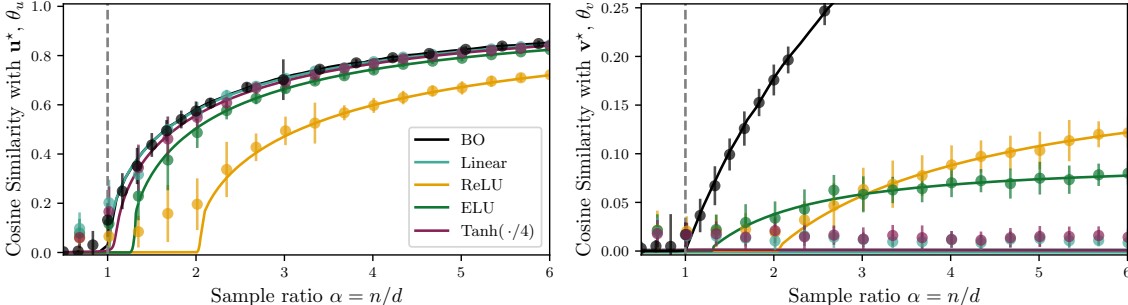

*Figure 1.* Cosine similarity of the BO and ERM estimators with, respectively, $\mathbf{u}^\star$ and $\mathbf{v}^\star$, for different choices of the activation of the autoencoder. Here, we consider a distribution $P_{\text{latents}}$ with correlation exponent $k^\star = 2$, concretely $\lambda \sim \mathcal{N}(0,1)$ and $\nu$ is $-\sqrt{2}$ if $|\lambda| < \Phi^{-1}(0.75)$ and $+\sqrt{2}$ otherwise, where $\Phi$ is the standard Gaussian cdf. Solid lines are analytical predictions in the high-dimensional limit (see Section 5). Colored dots are numerical simulations of minimization with full-batch Adam (Kingma & Ba, 2015), with learning rate $\eta = 0.1$ over 800 epochs and no weight decay ($d = 2000$, averaged over 30 instances, error bars represent one standard deviation). Black dots are runs of AMP for $d = 10000$, averaged over 72 seeds. The vertical dashed line is $\alpha_u^{\text{lin}}$ given in Result 3.2. The theoretical predictions and the simulation match, modulo finite-size effects that are further discussed in Appendix E.1. We see that the linear autoencoder does not achieve weak recovery of the $\mathbf{v}^\star$ spike, nor does the one with $\tanh$ non-linearity, while autoencoders with other depicted non-linearities do. The theoretical prediction of $\tanh$ also holds for the rescaled activation.

**About weak recovery.** Theorem 4.2 characterizes the learning dynamics only up to the time when either $m_u$ or $m_v$ reaches a positive threshold independent of $d$. Beyond this regime, the low-order polynomial approximation of the population loss is no longer valid. This limitation is intrinsic rather than technical: after weak recovery, the dynamics depend on all coefficients in Equation (16), and a general analysis (e.g., of the limiting values of $m_u$ and $m_v$) is impossible without fully specifying $P_{\text{latents}}$ and $\sigma$. Nevertheless, weak recovery already captures the essential phenomenon of dimension reduction, yielding an effective low-dimensional dynamical system independent of $d$.

## 5. Autoencoder: Empirical risk minimization

We study the global minimum of the empirical risk Equation (6) along similar technical lines to (Cui & Zdeborová, 2023).

**Result 5.1** (Asymptotic characterization of the ERM). *Consider the empirical risk Equation (6) under the spiked cumulant model, defined in Section 1, in the high-dimensional limit $n, d \to \infty$ with $\alpha = n/d = \mathcal{O}_d(1)$. Call $\hat{\mathbf{w}} \in \mathbb{R}^d$ any global minimum of Equation (6) and define the order parameters*

$$m_u^{\text{ER}} = \frac{\hat{\mathbf{w}}^\top \mathbf{u}^\star}{d}, \qquad m_v^{\text{ER}} = \frac{\hat{\mathbf{w}}^\top \mathbf{v}^\star}{d}, \qquad q^{\text{ER}} = \frac{\hat{\mathbf{w}}^\top \hat{\mathbf{w}}}{d}. \tag{23}$$

*Then, under the so-called replica symmetric assumption (see Appendix B) the order parameters concentrate onto their mean w.r.t. the randomness of the data distribution, and their limiting values are given by*

$$\{m_u^{\text{ER}}, m_v^{\text{ER}}, q^{\text{ER}}\} = \arg\max_{m_u, m_v, q} \phi_{\text{ER}}(m_u, m_v, q), \tag{24}$$

*where $\phi_{\text{ER}}$ is given by Equation (204) in Appendix D. $\phi_{\text{ER}}$ depends on $\alpha$, $P_{\text{latents}}$ and the activation function $\sigma$. The cosine similarity defined in Equation (8) for the ERM estimator is in the high-dimensional limit given by $\theta_s^{\text{ER}} \to m_s^{\text{ER}}/\sqrt{q^{\text{ER}}}$ for both spikes $s \in \{u, v\}$.*

Detailed derivation of Result 5.1 is given in Appendix D. Here, the empirical loss is a non-convex function of the weights, thus in principle the replica symmetric assumption, detailed in Appendix B, could be violated. If this were the case, the result above would yield a lower bound on the global minimizer of the empirical risk (Guerra, 2003; Franz & Leone, 2003; Talagrand, 2010). Making Result 5.1 mathematically rigorous poses a considerable technical challenge that would require considerable extension of the results in (Vilucchio et al., 2025) to the present model and to show that the so-called replicon condition described therein holds. It is thus left for future work.

Numerical investigations of gradient descent on the ERM objective, see Fig. 1 and 2, show that Result 5.1 is largely predictive of the behavior of GD, despite the non-convexity of the loss and the fact that Result 5.1 describes the global minima (under the replica symmetric assumption). The observed agreement thus supports the validity of the theoretical prediction of Result 5.1. We provide additional results for $\tanh$ activation function without rescaling the argument in Appendix E.2 and simulations with multiple hidden neurons in Appendix E.5.

**Weak-recovery of the hidden spike $\mathbf{v}^\star$.** In Fig. 1, we plot the cosine similarity of the ERM estimator with the data spikes for a given example of $P_{\text{latents}}$ with correlation exponent $k^\star = 2$ as a function of the ratio of sam-

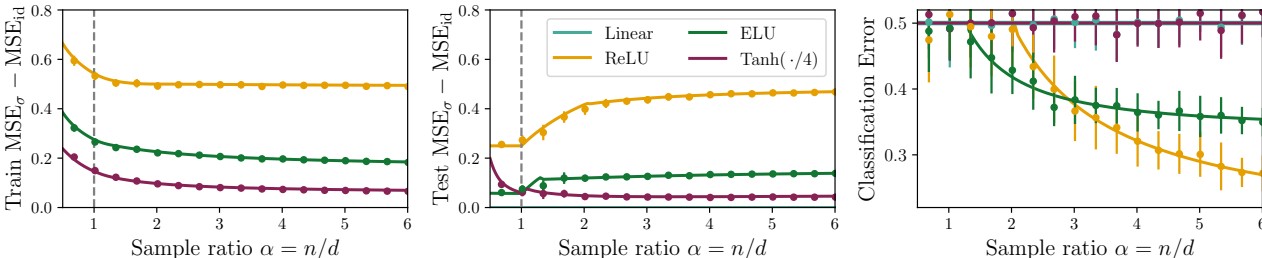

*Figure 2.* Train loss (left panel) and test loss (center panel) of the non-linear autoencoders minus the corresponding loss of the linear autoencoder, $\sigma = \mathrm{id}$, as a function of the sample complexity. The vertical dashed line is $\alpha_u^{\mathrm{lin}}$. The parameters and $P_{\mathrm{latents}}$ are the same as in Fig. 1. We see that both train and test losses are smaller for the linear autoencoder. Right panel: Classification error on the downstream task, where we see the advantage of the non-linearities that recovered correlation with the hidden spike.

ples $\alpha = n/d$ and for a variety of nonlinearities. For the considered non-linearities and ranges of $\alpha$, we observe a phenomenology fully compatible with the results for the gradient flow on the population loss described in Section 4. If a non-linearity should not weakly recover the $\mathbf{v}^\star$ spike in $\Theta(\log d)$ time-steps according to Theorem 4.2, in ERM it does not weakly recover it at considered sample ratio $\alpha$ and vice versa.

The linear autoencoder ($\sigma(x) = x$) aligns only with $\mathbf{u}^\star$ and never weak-recovers $\mathbf{v}^\star$. This is compatible with the fact that the linear autoencoder is performing PCA on the data covariance (see Proposition 2.1) where no information on $\mathbf{v}^\star$ is present. The weak recovery of $\mathbf{u}^\star$ then follows again a Baik-Ben Arous-Péché transition (Baik et al., 2005) at $\alpha_u^{\mathrm{lin}} = 1/\mathbb{E}[\lambda^2]^2$.

For the activation $\sigma = \tanh(\cdot/4)$ and correlation exponent $k^\star = 2$, Theorem 4.2 suggests weak recovery of $\mathbf{v}^\star$ requires at least $\Theta(\sqrt{d})$ time steps. Correspondingly, we see in Fig. 1 that $\tanh$ does not recover $\mathbf{v}^\star$ in the considered range of $\alpha$.

Non-linear autoencoders with other considered activations (ReLU, ELU, but also Swish, Sigmoid and GELU see Appendix E.3) manage to weakly recover both $\mathbf{u}^\star$ and $\mathbf{v}^\star$ at finite sample ratio $\alpha_{\mathrm{weak}}^\sigma$, confirming a clear advantage of nonlinearities in the setting of finite sample complexity. Interestingly, $\alpha_u^{\mathrm{lin}} < \alpha_{\mathrm{weak}}^\sigma$ for all nonlinearities we considered, revealing a tradeoff between how many samples are needed to weakly recover $\mathbf{u}^\star$ and $\mathbf{v}^\star$: the linear autoencoder has an advantage in the $\mathbf{u}^\star$ recovery at the price of not recovering $\mathbf{v}^\star$.

## 6. Lower test loss $\neq$ better generalization

In the left and center plots of Figure 2 we report train and test reconstruction error for different activation functions $\sigma$, observing that the linear autoencoder consistently achieves lower loss than its nonlinear counterparts. Indeed, for any

activation $\sigma$ and any $\mathbf{w} \in \mathbb{R}^d$,

$$\mathbb{E}\left[\left\|\mathbf{x} - \frac{\mathbf{w}}{\sqrt{d}}\sigma\left(\mathbf{x}^\top \frac{\mathbf{w}}{\sqrt{d}}\right)\right\|^2\right] \geq \underbrace{\mathbb{E}\left[\left\|\mathbf{x} - \tilde{\mathbf{w}}^\top \mathbf{x}\tilde{\mathbf{w}}\right\|^2\right]}_{\text{Linear autoencoder with } \|\tilde{\mathbf{w}}\| = 1},$$

(25)

where $\tilde{\mathbf{w}} = \mathbf{w}/\|\mathbf{w}\|$. Interpreting $\mathbb{E}$ as either an empirical or population average yields $\mathcal{L}^\sigma(\mathbf{w}) \geq \mathcal{L}^{\mathrm{linear}}(\tilde{\mathbf{w}})$ for all $\mathbf{w}$. We thus see that reconstruction error is not aligned with representation quality in the present model. Proposition 2.1 shows that linear autoencoders reduce to PCA (Bourlard & Kamp, 1988) and therefore fail to recover the hidden spike $\mathbf{v}^\star$, while Theorem 4.2 and the analysis of Section 5 show that nonlinear autoencoders achieve nontrivial overlap with $\mathbf{v}^\star$ by exploiting higher-order cumulants, even though they incur higher reconstruction loss.

In practice in self-supervised and generative learning, representation quality is rarely assessed through the value of the test loss itself, but instead via downstream tasks or linear probing (Grill et al., 2020; He et al., 2022; Devlin et al., 2019). We thus consider a toy downstream classification task where we assume that we have access to a small labeled dataset. For concreteness and simplicity, we model such labels as being obtained by the data-generative process $y_\mu = \mathrm{sign}(\mathbf{x}_\mu^\top \mathbf{v}^\star)$ for $\mu \in \{1, \ldots, n_{\mathrm{down}}\}$. We thus have access to $n$ unlabeled samples on which we learn the representation $\hat{\mathbf{w}}$ during pre-training (the weight of the autoencoder studied above). The downstream task then consists of a small number $n_{\mathrm{down}} \ll n$ of labeled samples $\{y_\mu, \mathbf{x}_\mu\}_{\mu=1}^{n_{\mathrm{down}}}$ and the goal is to predict the labels. We assume a linear predictor $\hat{y}_{\mathrm{new}} = \mathrm{sign}(\mathbf{x}_{\mathrm{new}}^\top \hat{\mathbf{w}})$ with weights initialized using the pre-training and that can be finetuned on the labeled dataset. We assume the labeled dataset is so small that finetuning does not provide an improvement with respect to initialization.

As shown in the rightmost panel of Figure 2, features learned by the linear autoencoder yield classification error at chance level, whereas nonlinear features achieve substantially lower

error which can be further improved with mild label aggregation, see Appendix D.2. These results identify a clear tractable example in which test reconstruction error and downstream performance are misaligned: linear autoencoders minimize reconstruction loss while discarding identifiable latent structure, whereas nonlinear (with suitable non-linearity) autoencoders recover both latent components and outperform on downstream classification despite higher test loss.

In practice, justifications for when test loss-based evaluation reflects representation quality and its failures remain limited. From this perspective, our model provides a simple and tractable setting in which a failure of such alignment can be explicitly demonstrated.

This issue is particularly relevant because validation loss is still widely used in self-supervised and generative pipelines for early stopping, and to guide the choice of regularization and architectural hyperparameters. Figure 2 illustrates how such loss-based choices can be misleading. In the central panel, the ReLU activation yields a higher test loss than ELU, yet in the right-hand panel, ReLU achieves better performance on the downstream task for sufficiently large sample ratios. This further highlights that loss-based criteria may be misaligned with representation quality in self-supervised learning, motivating further study of when such criteria are justified.

## 7. Conclusion

We introduced a solvable high-dimensional data model in which nonlinear autoencoders provably recover latent structure invisible to PCA. The model also exhibits a concrete misalignment between reconstruction loss and representation quality: linear autoencoders achieve lower test loss while failing to recover identifiable latent factors. We hope this framework will serve as a tractable testbed for further theoretical study of nonlinear representation learning and evaluation in self-supervised settings, including extensions to contrastive and masked-objective architectures. A limitation of this work is its focus on minimal single-neuron autoencoders and a specific class of higher-order dependencies. Extending the analysis to multi-neuron architectures, richer dependence structures, and other self-supervised objectives remains open.

## Acknowledgment

We thank Itay Griniasty for preliminary discussions on the architecture and data model, Hugo Cui and Léo Catteau for discussions related to the analytic solution for the autoencoder architecture, Sebastian Goldt and Florent Krzakala for many insightful discussions about the advantage of non-linearity in autoencoders and on the spiked cumulant model, Marc Mézard and Eric Vanden-Eijnden for discussions about subtleties of validation in self-supervised learning.

We acknowledge funding from the Swiss National Science Foundation grants SNSF SMArtNet (grant number 212049), and the Simons Collaboration on the Physics of Learning and Neural Computation via the Simons Foundation grant (#1257413 (LZ)).

## Impact Statement

This paper presents work whose goal is to advance the field of Machine Learning. There are many potential societal consequences of our work, none which we feel must be specifically highlighted here.

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

# A. Details and proofs for gradient flow on the population loss

## A.1. Linear $\sigma$

*Proof of Proposition 2.1.* With $\sigma = \mathrm{id}$ the reconstruction equals $(\mathbf{w}\mathbf{w}^\top/d)\mathbf{x}$, hence

$$\mathcal{L}(\mathbf{w}) = \mathbb{E}\Big[\big\|\mathbf{x} - \tfrac{\mathbf{w}\mathbf{w}^\top}{d}\mathbf{x}\big\|^2\Big] = \mathbb{E}\|\mathbf{x}\|^2 - \frac{2}{d}\mathbb{E}\big[(\mathbf{w}^\top\mathbf{x})^2\big] + \frac{1}{d^2}\mathbb{E}\big[\|\mathbf{w}\|^2(\mathbf{w}^\top\mathbf{x})^2\big], \tag{26}$$

where $\mathbb{E}$ can denote both population or empirical averages. We now use $\mathbb{E}\|\mathbf{x}\|^2 = \mathrm{Tr}(\Sigma)$ (where we use the notation $\Sigma = \mathbb{E}[\mathbf{x}\mathbf{x}^\top]$) and $\mathbb{E}[(\mathbf{w}^\top\mathbf{x})^2] = \mathbf{w}^\top\Sigma\mathbf{w}$, we get the exact identity

$$\mathcal{L}(\mathbf{w}) = \mathrm{Tr}(\Sigma) - \frac{2}{d}\mathbf{w}^\top\Sigma\mathbf{w} + \frac{\|\mathbf{w}\|^2}{d^2}\mathbf{w}^\top\Sigma\mathbf{w}. \tag{27}$$

**Step 1 (optimal scaling).** Fix any direction $\hat{\mathbf{w}} \neq 0$ and write $\mathbf{w} = r\hat{\mathbf{w}}$. Let $a := \hat{\mathbf{w}}^\top\Sigma\hat{\mathbf{w}} \geq 0$. Then Equation (27) becomes

$$\mathcal{L}(r\hat{\mathbf{w}}) = \mathrm{Tr}(\Sigma) - \frac{2a}{d}r^2 + \frac{a}{d^2}r^4. \tag{28}$$

If $a > 0$, this is a strictly convex quadratic polynomial in $r^2$, minimized at $r^2 = d$. (If $a = 0$ then $\mathbf{w}^\top\Sigma\mathbf{w} = 0$ and $\mathcal{L}(r\hat{\mathbf{w}}) \equiv \mathrm{Tr}(\Sigma)$, so any $r$ is equivalent; this corresponds to $\hat{\mathbf{w}} \in \ker(\Sigma)$.) Thus any non-degenerate minimizer must satisfy $\|\mathbf{w}\|^2 = d$.

**Step 2 (reduction to Rayleigh quotient).** Restricting Equation (27) to $\|\mathbf{w}\|^2 = d$ yields

$$\mathcal{L}(\mathbf{w}) = \mathrm{Tr}(\Sigma) - \frac{1}{d}\mathbf{w}^\top\Sigma\mathbf{w}, \qquad \text{for all } \mathbf{w} \text{ such that } \|\mathbf{w}\|^2 = d. \tag{29}$$

Hence minimizing $\mathcal{L}$ over $\{\|\mathbf{w}\|^2 = d\}$ is equivalent to maximizing $\mathbf{w}^\top\Sigma\mathbf{w}$ over the same set, i.e. maximizing the Rayleigh quotient. Writing $\tilde{\mathbf{w}} := \mathbf{w}/\sqrt{d}$ gives $\|\tilde{\mathbf{w}}\| = 1$ and the optimization becomes

$$\max_{\|\tilde{\mathbf{w}}\|=1} \tilde{\mathbf{w}}^\top\Sigma\tilde{\mathbf{w}}. \tag{30}$$

**Step 3 (PCA).** By the Rayleigh–Ritz variational principle, the maximum equals the top eigenvalue $\lambda_1(\Sigma)$ and is achieved exactly by the top eigenspace; if $\lambda_1(\Sigma) > \lambda_2(\Sigma)$ then the maximizers are $\pm\mathbf{e}_1$, where $\mathbf{e}_1$ is the leading eigenvector, that coincides with $\mathbf{u}^\star/\sqrt{d}$ in our case. Therefore any global minimizer satisfies $\hat{\mathbf{w}}/\sqrt{d} \in \arg\max_{\|\tilde{\mathbf{w}}\|=1} \tilde{\mathbf{w}}^\top\Sigma\tilde{\mathbf{w}}$, proving the claim. Finally, $\hat{\mathbf{w}}\hat{\mathbf{w}}^\top/d = \tilde{\mathbf{w}}\tilde{\mathbf{w}}^\top$ is the orthogonal projector onto the leading principal component.

$\square$

### A.1.1. EMPIRICAL COVARIANCE

**Lemma A.1** (Empirical covariance eigenvectors do not correlate with $\mathbf{v}^\star$). *Let*

$$y_\mu := S\Big(\nu_\mu \frac{v^\star}{\sqrt{d}} + z_\mu\Big) \in \mathbb{R}^d, \qquad \hat{\Sigma}_y := \frac{1}{n}\sum_{\mu=1}^n y_\mu y_\mu^\top, \tag{31}$$

*where $z_\mu \sim \mathcal{N}(0, I_d)$ are i.i.d., $\{\nu_\mu\}_{\mu=1}^n$ are i.i.d. with $\mathbb{E}[\nu] = 0$, $\mathbb{E}[\nu^2] =: \kappa \in (0, \infty)$, and bounded moments of all orders, and $S$ is given by Equation (2). Assume $n/d \to \alpha \in (0, \infty)$. Let $\hat{e}$ be the leading eigenvector of $\hat{\Sigma}_y$. Then for all $\varepsilon > 0$ and $D > 0$*

$$\mathbb{P}\Big(\big|\langle\hat{e}, \, v^\star/\|v^\star\|\rangle\big| \geq d^{-1/2+\varepsilon}\Big) \leq d^{-D} \tag{32}$$

*for $d$ large enough.*

*Proof.* We proceed in three steps.

**Step 1: rotation.** Condition on $v^\star$. Let $O$ be an orthogonal matrix such that $O(v^\star/\|v^\star\|) = e_1$, and define

$$y'_\mu := Oy_\mu, \qquad \hat{\Sigma}_{y'} := O\hat{\Sigma}_y O^\top. \tag{33}$$

If $\hat{e}$ is the leading eigenvector of $\hat{\Sigma}_y$, then $\hat{e}' := O\hat{e}$ is the leading eigenvector of $\hat{\Sigma}_{y'}$, and

$$\langle \hat{e}, \tfrac{v^\star}{\|v^\star\|} \rangle = \langle \hat{e}', e_1 \rangle. \tag{34}$$

Hence it suffices to bound $|\langle \hat{e}', e_1 \rangle|$.

**Step 2: independence and normalization.** Since $z_\mu \sim \mathcal{N}(0, I_d)$, the rotated variables $z'_\mu := Oz_\mu$ have independent coordinates. Because $S$ is a rank-one deformation along $v^\star$, we have $OSO^\top = I - c\,e_1 e_1^\top$ for some scalar $c$, and therefore

$$y'_\mu = \Big( (1-c)\big(\nu_\mu \tfrac{\|v^\star\|}{\sqrt{d}} + z'_{\mu,1}\big),\ z'_{\mu,2}, \ldots, z'_{\mu,d} \Big). \tag{35}$$

Thus the entries of the data matrix $Y' = [y'_1, \ldots, y'_n] \in \mathbb{R}^{d \times n}$ are independent. Moreover, by construction of $S$,

$$\mathbb{E}[y_\mu y_\mu^\top \mid v^\star] = I_d, \tag{36}$$

so each coordinate of $y'_\mu$ has variance one.

Define the normalized matrix

$$X := (dn)^{-1/4} Y' \in \mathbb{R}^{d \times n}. \tag{37}$$

Then the entries of $X$ are independent, centered, satisfy $\mathbb{E}[X_{i\mu}^2] = 1/\sqrt{dn}$, have uniformly bounded moments of all orders, and since $n/d \to \alpha \in (0, \infty)$ the dimensional condition $d \asymp n$ holds. Hence $X$ satisfies Assumptions (2.1)–(2.3) of (Bloemendal et al., 2014).

**Step 3: application of isotropic delocalization.** Let $K := \min\{d, n\}$. By Theorem 2.8 in (Bloemendal et al., 2014), for any deterministic unit vector $q \in \mathbb{R}^d$ and any eigenvector $\tilde{e}^{(\beta)}$ of $XX^\top$ with index $\beta \leq (1-\epsilon)K$,

$$\mathbb{P}\left( |\langle \tilde{e}^{(\beta)}, q \rangle| \geq d^{-1/2+\varepsilon} \right) \leq d^{-D} \tag{38}$$

for any $\varepsilon > 0$ and $D > 0$, for $d$ large enough. Since $K \to \infty$ and the leading eigenvector corresponds to index $\beta = 1$, this condition is satisfied for any fixed $\epsilon \in (0, 1)$ and all sufficiently large $d, n$. By Remark 2.11 in (Bloemendal et al., 2014), the same bound applies to eigenvectors of $XX^\top$. Because $XX^\top$ and $\hat{\Sigma}_{y'} = \frac{1}{n}Y'Y'^\top$ have the same eigenvectors, we obtain

$$\mathbb{P}\left( |\langle \hat{e}, e_1 \rangle| \geq d^{-1/2+\varepsilon} \right) \leq d^{-D} \tag{39}$$

Using Step 1 concludes the proof. $\qquad\qquad\qquad\qquad\qquad\qquad\qquad\qquad\qquad\qquad\qquad\square$

## A.2. Hermite expansion of the loss

**Hermite polynomials and Gaussian $\mathscr{L}^2$ expansions.** Throughout the paper we use the *probabilists' Hermite polynomials* $\{\mathrm{He}_k\}_{k \geq 0}$, defined by

$$\mathrm{He}_k(z) := (-1)^k e^{z^2/2} \frac{\mathrm{d}^k}{\mathrm{d}z^k} e^{-z^2/2}. \tag{40}$$

They form an orthogonal basis of the Hilbert space $\mathscr{L}^2(\gamma_1)$, where $\gamma_1 = \mathcal{N}(0, 1)$ is the standard Gaussian measure, with inner product $\langle f, g \rangle = \mathop{\mathbb{E}}_{Z \sim \gamma_1} [f(Z)g(Z)]$. In particular,

$$\mathop{\mathbb{E}}_{Z \sim \gamma_1} \big[ \mathrm{He}_k(Z) \mathrm{He}_\ell(Z) \big] = k!\, \delta_{k,\ell}. \tag{41}$$

As a consequence, any function $f \in \mathscr{L}^2(\gamma_1)$ admits the expansion

$$f(z) = \sum_{k \geq 0} \frac{1}{k!} \mathop{\mathbb{E}}_{Z \sim \gamma_1} \big[ f(Z) \mathrm{He}_k(Z) \big] \mathrm{He}_k(z), \tag{42}$$

with convergence in $\mathscr{L}^2(\gamma_1)$.

More generally, products of Hermite polynomials provide an orthogonal basis of $\mathscr{L}^2(\gamma_d)$, $\gamma_d = \mathcal{N}(0, \mathbb{I}_d)$, and expectations of products of Hermite polynomials evaluated at correlated Gaussian variables can be computed explicitly.

Standard references on Hermite polynomials and Gaussian Hilbert spaces include (Szegő, 1939; McCullagh, 2018).

**Lemma A.2** (Triple Hermite identity). *Let $(X, Y, Z)$ be centered jointly Gaussian with $\mathbb{E}[X^2] = \mathbb{E}[Y^2] = \mathbb{E}[Z^2] = 1$, $\mathbb{E}[XY] = 0$, and set $\rho_1 := \mathbb{E}[XZ]$, $\rho_2 := \mathbb{E}[YZ]$. Then for all $i, j, k \in \mathbb{N}$,*

$$\mathbb{E}[\mathrm{He}_i(X)\,\mathrm{He}_j(Y)\,\mathrm{He}_k(Z)] = \mathbf{1}_{\{k=i+j\}}\, k!\, \rho_1^i \rho_2^j. \tag{43}$$

*Proof.* Use the generating function $\exp(tz - t^2/2) = \sum_{k \geq 0} \mathrm{He}_k(z)\, t^k/k!$. By Gaussian regression, there exists $\varepsilon \sim \mathcal{N}(0,1)$ independent of $(X, Y)$ such that $Z = \rho_1 X + \rho_2 Y + \sqrt{1 - \rho_1^2 - \rho_2^2}\, \varepsilon$. Hence, conditioning on $(X, Y)$ and using independence of $\varepsilon$,

$$\mathbb{E}\left[e^{tZ - t^2/2} \mid X, Y\right] = \exp\left(t(\rho_1 X + \rho_2 Y) - \tfrac{t^2}{2}(\rho_1^2 + \rho_2^2)\right). \tag{44}$$

Therefore

$$\begin{aligned}
\mathbb{E}\left[\mathrm{He}_i(X)\,\mathrm{He}_j(Y)e^{tZ - t^2/2}\right] &= \mathbb{E}\left[\mathrm{He}_i(X)\,\mathrm{He}_j(Y)e^{t\rho_1 X - (t\rho_1)^2/2}e^{t\rho_2 Y - (t\rho_2)^2/2}\right] \\
&= \mathbb{E}\left[\mathrm{He}_i(X)e^{t\rho_1 X - (t\rho_1)^2/2}\right]\mathbb{E}\left[\mathrm{He}_j(Y)e^{t\rho_2 Y - (t\rho_2)^2/2}\right] \\
&= (t\rho_1)^i (t\rho_2)^j,
\end{aligned}$$

where the factorization uses $X \perp Y$ and the last equality follows by matching coefficients in the generating function (equivalently, expanding $e^{t\rho_1 X - (t\rho_1)^2/2}$ in Hermites and using orthogonality). On the other hand,

$$\mathbb{E}\left[\mathrm{He}_i(X)\,\mathrm{He}_j(Y)e^{tZ - t^2/2}\right] = \sum_{k \geq 0} \frac{t^k}{k!}\, \mathbb{E}[\mathrm{He}_i(X)\,\mathrm{He}_j(Y)\,\mathrm{He}_k(Z)]. \tag{45}$$

Comparing coefficients of $t^k$ yields the claim. $\qquad\square$

*Proof of Lemma 4.1.* Let $\mathbf{s} := \mathbf{w}/\sqrt{d} \in \mathbb{S}^{d-1}$ and set $Z_u := \mathbf{x}^\top \mathbf{u}^\star/\sqrt{d}$, $Z_v := \mathbf{x}^\top \mathbf{v}^\star/\sqrt{d}$, $Z_w := \mathbf{x}^\top \mathbf{w}/\sqrt{d} = \mathbf{s}^\top \mathbf{x}$. Under $\mathbf{x} \sim \mathcal{N}(0, \mathbb{I}_d)$, $(Z_u, Z_v, Z_w)$ is centered Gaussian with $\mathbb{E}[Z_u Z_w] = m_u$, $\mathbb{E}[Z_v Z_w] = m_v$, $\mathbb{E}[Z_u Z_v] = 0$, hence any expectation depending on $\mathbf{x}$ only through $(Z_u, Z_v, Z_w)$ depends on $\mathbf{w}$ only via $(m_u, m_v)$.

Expanding Equation (13) and using $\|\mathbf{w}\|^2 = d$ gives

$$\mathcal{L}(\mathbf{w}) = \mathbb{E}[L(\mathbf{x})\|\mathbf{x}\|^2] + \mathbb{E}[L(\mathbf{x})\sigma(Z_w)^2] - 2\mathbb{E}[L(\mathbf{x})Z_w\sigma(Z_w)], \tag{46}$$

with $C := \mathbb{E}[L(\mathbf{x})\|\mathbf{x}\|^2]$.

By $\tilde{L} \in \mathscr{L}^2(\mathcal{N}(0, \mathbb{I}_2))$ and $\sigma^2, z\sigma \in \mathscr{L}^2(\mathcal{N}(0,1))$, the Hermite expansions hold in $\mathscr{L}^2$:

$$\tilde{L}(z_1, z_2) = \sum_{i,j \geq 0} \frac{c_{i,j}^L}{i!j!}\mathrm{He}_i(z_1)\mathrm{He}_j(z_2), \quad \sigma(z)^2 = \sum_{k \geq 0}\frac{c_k^{\sigma^2}}{k!}\mathrm{He}_k(z), \quad z\sigma(z) = \sum_{k \geq 0}\frac{c_k^{z\sigma}}{k!}\mathrm{He}_k(z). \tag{47}$$

Insert these into $\mathbb{E}[\tilde{L}(Z_u, Z_v)\sigma(Z_w)^2]$ and $\mathbb{E}[\tilde{L}(Z_u, Z_v)Z_w\sigma(Z_w)]$ and apply Lemma A.2 with $(X, Y, Z) = (Z_u, Z_v, Z_w)$, $\rho_1 = m_u$, $\rho_2 = m_v$, to obtain

$$\mathbb{E}[\tilde{L}(Z_u, Z_v)\mathrm{He}_k(Z_w)] = \sum_{i=0}^{k}\frac{c_{i,k-i}^L}{i!(k-i)!}\,k!\, m_u^i m_v^{k-i}. \tag{48}$$

This yields Equation (16) after collecting the $\sigma^2$ and $z\sigma$ contributions.

For uniform absolute convergence on $|m_u|, |m_v| \leq r < 1$, use Parseval: $\sum_{i,j}(c_{i,j}^L)^2/(i!j!) < \infty$, $\sum_k (c_k^{\sigma^2})^2/k! < \infty$, $\sum_k (c_k^{z\sigma})^2/k! < \infty$, and Cauchy–Schwarz together with

$$\sum_{i=0}^{k} \frac{1}{i!(k-i)!} = \frac{1}{k!}\sum_{i=0}^{k}\binom{k}{i} = \frac{2^k}{k!}. \tag{49}$$

$\square$

**Lemma A.3.** *Let $(\lambda, \nu)$ be real-valued random variables, and let $z \sim \mathcal{N}(0,1)$ be independent of $(\lambda, \nu)$. Fix $p \in \mathbb{N}$. Assume $\mathbb{E}\big[|\nu|\,|\lambda|^p\big] < \infty$. Then*

$$\mathbb{E}\big[\operatorname{He}_p(\lambda+z)\,\nu\big] = \mathbb{E}\big[\lambda^p\,\nu\big]. \tag{50}$$

*Proof.* Fix $p \geq 0$. We first show that

$$\mathbb{E}\big[\operatorname{He}_p(\lambda+z) \mid \lambda\big] = \lambda^p \qquad \text{a.s.} \tag{51}$$

Recall the generating function identity

$$G(t,x) := \exp\Big(tx - \tfrac{t^2}{2}\Big) = \sum_{k=0}^{\infty} \operatorname{He}_k(x)\,\frac{t^k}{k!}, \tag{52}$$

and in particular $\operatorname{He}_p(x) = \partial_t^p G(t,x)\big|_{t=0}$.

Fix $\lambda$ and consider the function

$$\phi(t) := \mathbb{E}\big[G(t, \lambda+z) \mid \lambda\big] = \mathbb{E}\Big[\exp\big(t(\lambda+z) - \tfrac{t^2}{2}\big)\Big|\lambda\Big]. \tag{53}$$

Since $z$ is independent of $\lambda$ and $\mathbb{E}[e^{tz}] = e^{t^2/2}$ for $z \sim \mathcal{N}(0,1)$, we have for all $t \in \mathbb{R}$,

$$\phi(t) = \exp\big(t\lambda - \tfrac{t^2}{2}\big)\,\mathbb{E}[e^{tz}] = \exp(t\lambda). \tag{54}$$

We claim that we may differentiate under the conditional expectation at $t = 0$ up to order $p$:

$$\phi^{(p)}(0) = \mathbb{E}\big[\partial_t^p G(t, \lambda+z)\big|_{t=0} \mid \lambda\big] = \mathbb{E}\big[\operatorname{He}_p(\lambda+z) \mid \lambda\big]. \tag{55}$$

To justify Equation (55), note that for $|t| \leq 1$,

$$\big|\partial_t^p G(t, \lambda+z)\big| \leq C_p\,(1 + |\lambda+z|^p)\,\exp(|\lambda+z|), \tag{56}$$

for a constant $C_p$ depending only on $p$ (this follows because $\partial_t^p G(t,x)$ is a degree-$p$ polynomial in $x$ times $e^{tx - t^2/2}$, and $|e^{tx - t^2/2}| \leq e^{|x|}$ when $|t| \leq 1$). The right-hand side is integrable in $z$ conditionally on $\lambda$ because $z$ is Gaussian (all exponential moments exist). Hence dominated convergence (conditionally on $\lambda$) permits differentiation under the conditional expectation at $t = 0$.

Combining Equation (55) with $\phi(t) = e^{t\lambda}$ yields

$$\mathbb{E}\big[\operatorname{He}_p(\lambda+z) \mid \lambda\big] = \phi^{(p)}(0) = \frac{d^p}{dt^p}e^{t\lambda}\Big|_{t=0} = \lambda^p, \tag{57}$$

proving Equation (51).

We now conclude the desired identity. By the tower property and measurability of $\nu$ with respect to $\sigma(\lambda, \nu)$,

$$\mathbb{E}[\operatorname{He}_p(\lambda+z)\,\nu] = \mathbb{E}\big[\mathbb{E}[\operatorname{He}_p(\lambda+z)\,\nu \mid \lambda, \nu]\big] = \mathbb{E}\big[\nu\,\mathbb{E}[\operatorname{He}_p(\lambda+z) \mid \lambda, \nu]\big]. \tag{58}$$

Since $z$ is independent of $(\lambda, \nu)$, the conditional distribution of $\lambda + z$ given $(\lambda, \nu)$ depends only on $\lambda$, hence $\mathbb{E}[\operatorname{He}_p(\lambda+z) \mid \lambda, \nu] = \mathbb{E}[\operatorname{He}_p(\lambda+z) \mid \lambda]$ a.s. Using Equation (51) gives

$$\mathbb{E}[\operatorname{He}_p(\lambda+z)\,\nu] = \mathbb{E}[\nu\,\lambda^p] = \mathbb{E}[\lambda^p\nu], \tag{59}$$

as claimed. $\square$

### A.3. Proof of Theorem 4.2

First a technical lemma.

**Lemma A.4** (Ratio of random spherical projections). *Let $d \geq 3$. Let $u, v$ be independent and uniform on the unit sphere $S^{d-1} \subset \mathbb{R}^d$, and let $w \in \mathbb{R}^d$ be fixed with $\|w\| = 1$. Then for every $\Gamma > 0$,*

$$\mathbb{P}\big(|v \cdot w| > \Gamma \, |u \cdot w|\big) \; \leq \; \frac{2}{\pi} \arctan\Big(\frac{3}{\Gamma}\Big) \; + \; 4e^{-cd}, \tag{60}$$

*for a universal constant $c > 0$. In particular, for any $\eta \in (0,1)$ there exists $d_0 = d_0(\eta)$ such that for all $d \geq d_0$,*

$$\mathbb{P}\big(|v \cdot w| > \Gamma \, |u \cdot w|\big) \; \leq \; f(\Gamma) := \frac{2}{\pi} \arctan\Big(\frac{3}{\Gamma}\Big) + \eta, \tag{61}$$

*where $f$ does not depend on $d$ and $f(\Gamma) \to 0$ as $\Gamma \to \infty$.*

*Proof.* By rotational invariance of the uniform measure on $S^{d-1}$, we may assume $w = e_1$, so that $u \cdot w = u_1$ and $v \cdot w = v_1$. Let $g, h \sim N(0, I_d)$ be independent and set $u = g/\|g\|$, $v = h/\|h\|$. Then $u, v$ are independent and uniform on $S^{d-1}$ and

$$u_1 = \frac{g_1}{\|g\|}, \qquad v_1 = \frac{h_1}{\|h\|}. \tag{62}$$

Define $A = \|g\|/\sqrt{d}$ and $B = \|h\|/\sqrt{d}$, and let

$$E = \Big\{ A \in (1/2, 3/2) \Big\} \cap \Big\{ B \in (1/2, 3/2) \Big\}. \tag{63}$$

Since $\|g\|^2 \sim \chi_d^2$ and $\|h\|^2 \sim \chi_d^2$, the law of large numbers with exponential tails for $\chi_d^2$ variables implies that there exists a universal constant $c > 0$ such that

$$\mathbb{P}(A \notin (1/2, 3/2)) \leq 2e^{-cd}, \qquad \mathbb{P}(B \notin (1/2, 3/2)) \leq 2e^{-cd}, \tag{64}$$

hence by a union bound $\mathbb{P}(E^c) \leq 4e^{-cd}$.

On $E$, we have $A/B \leq 3$, and therefore

$$|v_1| > \Gamma |u_1| \implies \frac{|h_1|}{\sqrt{d}\,B} > \Gamma \frac{|g_1|}{\sqrt{d}\,A} \implies |h_1| > \Gamma \frac{B}{A} |g_1| \implies |h_1| > \frac{\Gamma}{3} |g_1|. \tag{65}$$

Thus,

$$\mathbb{P}(|v_1| > \Gamma |u_1|) \leq \mathbb{P}\big(|h_1| > (\Gamma/3)|g_1|\big) + \mathbb{P}(E^c). \tag{66}$$

Now $g_1, h_1$ are independent $N(0,1)$, say $Z_1, Z_2$. By rotational invariance in $\mathbb{R}^2$, the angle $\Theta$ of $(Z_1, Z_2)$ is uniform on $[0, \pi)$ and $Z_2/Z_1 = \tan\Theta$, hence for $\alpha > 0$,

$$\mathbb{P}(|Z_2| > \alpha|Z_1|) = \mathbb{P}(|\tan\Theta| > \alpha) = \frac{2}{\pi} \arctan\Big(\frac{1}{\alpha}\Big). \tag{67}$$

With $\alpha = \Gamma/3$ we obtain

$$\mathbb{P}(|v_1| > \Gamma|u_1|) \leq \frac{2}{\pi} \arctan\Big(\frac{3}{\Gamma}\Big) + 4e^{-cd}, \tag{68}$$

which is the claimed bound. The final statement follows by choosing $d_0$ so that $4e^{-cd_0} \leq \eta$. $\qquad\square$

We can now proceed with the proof of Theorem 4.2.

**First item.** By Lemma 4.1, the population loss $\mathcal{L}(\mathbf{w})$ depends on $\mathbf{w}$ only through the overlaps

$$m_u = \frac{\mathbf{w}^\top \mathbf{u}^\star}{d}, \qquad m_v = \frac{\mathbf{w}^\top \mathbf{v}^\star}{d}, \tag{69}$$

and admits a convergent Hermite expansion in $(m_u, m_v)$ in a neighborhood of the origin. In particular, since $\mathcal{L}$ is assumed to be $\mathcal{C}^4$, truncating the expansion at total degree four yields

$$\mathcal{L}(\mathbf{w}) = C + C_2 m_u^2 + C_3 m_u^2 m_v + \text{(quartic terms)} + R(m_u, m_v), \tag{70}$$

where $R(m_u, m_v) = \mathcal{O}(\|(m_u, m_v)\|^5)$, and the constants $C_2, C_3$ are exactly the coefficients defined in the statement of the theorem (explicit expressions are given in Lemma 4.1 and the subsequent specialization to the spiked cumulant model).

Consider now the spherical gradient flow dynamics Equation (20). Since $\mathcal{L}$ depends on $\mathbf{w}$ only through $(m_u, m_v)$, projecting the dynamics along $\mathbf{u}^\star$ and $\mathbf{v}^\star$ yields a closed system of ordinary differential equations for the overlaps. More precisely, taking the scalar product of Equation (20) with $\mathbf{u}^\star$ and $\mathbf{v}^\star$, and using $\|\mathbf{w}\| = \|\mathbf{u}^\star\| = \|\mathbf{v}^\star\| = \sqrt{d}$, we obtain that $(m_u(t), m_v(t))$ satisfy the following ODE system:

$$\dot{m}_u(t) = -C_2 m_u + R_u(m_u, m_v) \tag{71}$$

$$\dot{m}_v(t) = -C_3 m_u^2 + R_v(m_u, m_v) \tag{72}$$

with $R_u, R_v$ that satisfy the following bounds for some coefficients $a_{1,2}, b_1, b_2 > 0$:

$$|R_u(m_u, m_v)| \le a_1 m_u^2 + a_2 m_v^2 \tag{73}$$

$$|R_v(m_u, m_v)| \le b_1 |m_u|^3 + b_2 |m_v|^3 \tag{74}$$

in a certain neighborhood of the origin $[-\tilde{\delta}, \tilde{\delta}]$, for some $\tilde{\delta} > 0$. We focus first on the proof of the first point, assuming $C_2 < 0$ and $C_3 \ne 0$. For convenience we now assume $C_3 < 0$ so that $0 < -C_3 = |C_3|$, in the case of flipped sign, we can reason in the same way flipping sign to $v$. Suppose also $m_u(0) > 0$ (otherwise flip sign to $u$), and let $\Gamma > 0$ be a fixed constant. We can assume that $0 < \epsilon < \tilde{\delta}$ is small enough so that:

$$\begin{cases} a_1 \epsilon^2 + a_2 \epsilon^2 \le \frac{|C_2|}{2} \epsilon \\ b_1 \epsilon^3 + \Gamma^3 b_2 \epsilon^3 \le \frac{|C_3|}{2} \epsilon^2 \\ \frac{3|C_3|}{2} \epsilon \le \frac{|C_2|}{2} \end{cases} \tag{75}$$

then consider the set $\mathcal{A}_{\Gamma, \epsilon} = \{(m_u, m_v) \mid 0 < m_u < \epsilon, |m_v| < \Gamma |m_u|\}$. We will prove that if the initialization $(m_u(0), m_v(0)) \in \mathcal{A}_{\Gamma, \epsilon}$, then it exits it in logarithmic time with weak recovery of the spikes. Let

$$T := \inf \{t \mid (m_u(t), m_v(t)) \notin \mathcal{A}_{\Gamma, \epsilon}\} \tag{76}$$

We know that $T > 0$, since we are assuming $(m_u(0), m_v(0)) \in \mathcal{A}_{\Gamma, \epsilon}$. Note that for $t \in [0, T]$, thanks to Equation (75) we have the following inequalities:

$$\begin{cases} \frac{|C_2|}{2} m_u(t) \le \dot{m}_u(t) \le \frac{3|C_2|}{2} m_u(t) \\ \frac{|C_3|}{2} m_u^2(t) \le \dot{m}_v(t) \le \frac{3|C_3|}{2} m_u^2(t) \end{cases} \tag{77}$$

The first line implies $m_u(t) \ge m_u(0) e^{|C_2| t / 2}$, hence $T \le \frac{2}{|C_2|} \log\left(\frac{\epsilon}{m_u(0)}\right)$, since if Equation (77) is satisfied until that point, then $m_u$ reaches $\epsilon$. We will now prove that this is the only scenario that can happen: neither $m_u(T) = 0$ or $|m_v(T)| = \Gamma |m_u(T)|$ can happen. We reason by contradiction, assuming $m_u(T) = 0$, then by the mean value theorem there exists $\hat{t} \in (0, T)$ such that $\dot{m}_u(\hat{t}) = \frac{1}{T}(m_u(T) - m_u(0)) < 0$, which via Equation (77) implies $m_u(\hat{t}) < 0$, contradicting the definition of $T$. Note that this reasoning also implies the slightly stronger consequence $m_u(T) > m_u(0)$. We can proceed analogously for the other condition: assume $|m_v(T)| = \Gamma |m_u(T)|$. We can immediately remove the absolute values since $m_u(T) > 0$ and by Equation (77) $m_v$ is increasing in $(0, T)$, but $-\Gamma m_u(T) < -\Gamma m_u(0) < m_v(0)$. Now we apply the mean value theorem on $g(t) := \Gamma m_u(t) - m_v(t)$, for which we know that $g(0) > 0$ and $g(T) = 0$. This would imply that there exists $\hat{t} \in (0, T)$ such that $0 > \dot{g}(\hat{t}) > \frac{\Gamma |C_2|}{2} m_u(\hat{t}) - \frac{3|C_3|}{2} m_u^2(\hat{t}) \ge m_u(\hat{t})\left(\frac{|C_2|}{2} - \frac{3|C_3|}{2}\epsilon\right)$, which is in contradiction with the third condition in Equation (75). So we proved that $m_u(T) = \epsilon$, and by Equation (77) in $(0, T)$ we have $m_u(t) \le m_u(0) e^{3|C_2| t / 2}$, so, together with the upper bound previously established, we know that

$T_- := \frac{2}{3|C_2|} \log\left(\frac{\epsilon}{m_u(0)}\right) \le T \le \frac{2}{|C_2|} \log\left(\frac{\epsilon}{m_u(0)}\right) =: T_+$. We can also deduce the following lower bound on $m_v(T)$

$$m_v(T) \ge m_v(0) + \frac{|C_3|}{2} \int_0^T m_u(t)^2 dt$$

$$\ge m_v(0) + \frac{|C_3|}{3|C_2|} \int_0^T m_u(t)\dot{m}_u(t) dt$$

$$\ge m_v(0) + \frac{|C_3|}{6|C_2|} \left(\epsilon^2 - m_u^2(0)\right) \tag{78}$$

We completed the proof of the necessary properties of the dynamical system, we now just need to take the high dimensional limit. Since $w(0) \sim \mathrm{Unif}(\mathbb{S}^{d-1})$ then, by Levy's lemma, we can deduce that the probability that $|m_u(0)|, |m_v(0)| \in (d^{-1/2-\hat\epsilon}, d^{-1/2+\hat\epsilon})$ converges to 1 for any $\hat\epsilon > 0$. We can now use lemma A.4 to deduce that given $\delta$, we can choose $\Gamma_\delta$ accordingly so that $|m_v(0)| \le \Gamma_\delta |m_u(0)|$ has probability larger than $1 - \delta/2$. Once we fixed $\Gamma_\delta$ (which is independent of $d$, thanks to lemma A.4) we can choose $\epsilon$ that satisfies Equation (75). So we proved that the event $(m_u(0), m_v(0)) \in \mathcal{A}_{\epsilon,\Gamma_\delta}$ has probability larger than $1 - \delta$, for $d$ large enough. Conditioning on this event we can then deduce from $\frac{2}{3|C_2|} \log\left(\frac{\epsilon}{m_u(0)}\right) \le T \le \frac{2}{|C_2|} \log\left(\frac{\epsilon}{m_u(0)}\right)$ that $T = \Theta(\log d)$ and that time achieves weak recovery: $m_u(T) = \epsilon$ and Equation (78) is lower bounded by $\frac{|C_3|}{6|C_2|}\epsilon^2 + o\left(\frac{1}{\sqrt{d}}\right)$ that leads to a positive lower bound independent of $d$. This concludes the proof of the first item of the statement.

**Proof of second item.** We now consider the case where $C_2 < 0$ but $C_3 = 0$. By Lemma 4.1, the population loss admits a Hermite expansion in $(m_u, m_v)$, and the gradient flow dynamics for the overlaps are obtained by differentiating this expansion.

Since $C_3 = 0$, all cubic terms in the expansion of $\mathcal{L}$ vanish identically. Consequently, the leading contribution to the $m_v$-dynamics arises from quartic terms. More precisely, retaining the relevant degree four part of the expansion (the term in the equation for $m_v$ that depends only on $m_u$ thanks to $C_{(3,1)}$) and absorbing all higher-order terms into a remainder yields

$$\dot{m}_v(t) = -C_{(3,1)} m_u^3(t) + R(m_u(t), m_v(t)), \tag{79}$$
$$|R(m_u, m_v)| \le b\left(|m_u|^2|m_v| + |m_u||m_v|^2 + |m_v|^3 + m_u^4\right). \tag{80}$$

for some $b > 0$. Without loss of generality we can assume $-C_{(3,1)} > 0$, otherwise we can just change the sign of $v$, in this way we have the inequalities:

$$\dot{m}_v(t) \le |C_{(3,1)}|m_u^3(t) + b\left(|m_u|^2|m_v| + |m_u||m_v|^2 + |m_v|^3 + m_u^4\right) \tag{81}$$
$$\dot{m}_v(t) \ge |C_{(3,1)}|m_u^3(t) - b\left(|m_u|^2|m_v| + |m_u||m_v|^2 + |m_v|^3 + m_u^4\right) \tag{82}$$

So the idea of the proof is the same as before, we define assume that $(m_u(0), m_v(0)) \in \mathcal{A}_{\epsilon,\Gamma}$ and prove that the time $T$ of first exit will be logarithmic in d and lead to weak recovery of $u$ and $v$. Assuming that $\epsilon > 0$ is small enough so that

$$\begin{cases} (a_1 + a_2\Gamma^2)\epsilon \le \frac{|C_2|}{2}, \\ \frac{\Gamma|C_2|}{2} \ge (|C_{(3,1)}| + b(2 + \Gamma^3))\epsilon^2. \end{cases} \tag{83}$$

we can repeat all the derivation for $m_u$ (assuming again $m_u(0) \ge 0$, otherwise we can flip the sign of $u$), reaching that neither the constraint $m_u(t) \ge 0$ nor $m_v(t) \le \Gamma m_u(t)$ will be violated for $t \le T$. Hence it must be that $m_u(T) = \epsilon$, and $T \le \frac{2}{|C_2|} \log\left(\frac{\epsilon}{m_u(0)}\right)$, since we also have the lower bound $m_u(t) \ge m_u(0)e^{|C_2|t/2}$.

The part of the argument that changes is related to showing that also $m_v$ needs to grow and reach weak recovery at time $T$. To do this we will show that for some $\kappa > 0, \theta \in \mathbb{R}$ we have that $m_v(t) > \kappa m_u^3(t) + \theta$, in some interval $[T_0, T]$. First note that in $\mathcal{A}_{\epsilon,\Gamma}$ we have that $|\dot{m}_v(t)| \le \epsilon|\dot{m}_u(t)|$ (assuming $\epsilon$ smaller than a constant that depends monotonically only on ODEs coefficients and $\Gamma$). So we have that

$$m_v(t)| \le |m_v(0)| + \epsilon(m_u(t) - m_u(0)) \le (\Gamma - \epsilon)m_u(0) + \epsilon m_u(t) \tag{84}$$

Hence (recalling $m_u(t) \geq m_u(0)e^{|C_2|t/2}$) we have that, for any $\kappa > \epsilon$, and $t \geq T_0 := \frac{2}{|C_2|}\log\frac{\Gamma-\epsilon}{\kappa-\epsilon}$, then $|m_v(t)| \leq \kappa m_u(t)$. So take $\theta = -2\kappa m_u(T_0)$ so that $m_v(T_0) > \kappa m_u^3(T_0) + \theta$ (we are using $m_u^3 < m_u$ in $\mathcal{A}_{\epsilon,\Gamma}$ so that the distance between the lower bound on $m_v(T_0)$ and $\theta$ is larger than $\kappa m_u^3(T_0)$). Then, consider the function $g(t) = m_v(t) - \kappa m_u^3(t) - \theta$. Assume by contradiction that it is not always positive in $(T_0, T)$, and call $T_g$ its smallest 0. $T_g > T_0$ and so, by the mean value theorem we have that there must be $\tilde{t} \in (T_0, T_g)$ such that the derivative of $g$ is negative (since $g(T_0) > 0$ and $g(T_g) = 0$ by construction), which implies:

$$0 > \dot{g}(\tilde{t}) \tag{85}$$

$$= \dot{m}_v(\tilde{t}) - 3\kappa m_u^2(\tilde{t})\dot{m}_u(\tilde{t}) \tag{86}$$

$$> |C_{(3,1)}|m_u^3(\tilde{t}) - b\left(m_u^2(\tilde{t})m_v(\tilde{t}) + m_u(\tilde{t})m_v(\tilde{t})^2 + m_v(\tilde{t})^3 + m_u^4(\tilde{t})\right) - \kappa m_u^2(\tilde{t})\left(\frac{9|C_2|}{2}m_u(\tilde{t})\right) \tag{87}$$

but now we can use that $\tilde{t} > T_0$, so $\kappa m_u(\tilde{t}) \geq |m_v(\tilde{t})|$, hence

$$0 > m_u^3(\tilde{t})\left(|C_{(3,1)}| - b\left(\kappa + \kappa^2 + \kappa^3 + \epsilon\right) - \kappa\frac{9|C_2|}{2}\right) \tag{88}$$

Hence we reach a contradiction by taking $\kappa$ smaller than an expression containing only the constants $|C_{(3,1)}|$, $b$ and $C_2$ and any $\epsilon < \kappa$.

So we proved that in $\mathcal{A}_{\epsilon,\Gamma}$ we must have $m_v(t) > \kappa m_u^3(t) - 2\kappa m_u(T_0)$. So we have that $m_v(t) \geq \kappa\epsilon^3 - m_u(0)e^{3\frac{\Gamma-\epsilon}{\kappa-\epsilon}}$, that we got using the upper bound $m_u(T_0) \leq m_u(0)e^{\frac{3}{2}|C_2|T_0}$ deduced from Equation (77).

As we did in the previous part of the proof we can now conclude by invoking Levy's lemma, deducing that the probability that $|m_u(0)|, |m_v(0)| \in (d^{-1/2-\hat{\epsilon}}, d^{-1/2+\hat{\epsilon}})$ converges to 1 for any $\hat{\epsilon} > 0$. Moreover, we use again lemma A.4 and re-apply the choices on $\Gamma_\delta$ as done in the first item, to conclude that for $d$ large enough and with probability larger than $1 - \delta$, $m_v(T) \geq \frac{\epsilon^4}{2}$ and as before $T = \Theta(\log d)$ concluding the proof of this item of the statement.

**Proof of third item.** Using the assumptions we have that the effective coefficient leading coefficient for $m_v$ dynamics $-C_e := 2C_2 - C_{(2,2)} < 0$, moreover all the terms that do not contain $m_v$ (i.e. are of the form $Cm_u^p$ for some $p \in \mathbb{N}$) are also zero by assumption. So we have that:

$$\dot{m}_v(t) = -C_e m_u^2(t)m_v(t) + R(m_u, m_v) \tag{89}$$

$$|R(m_u, m_v)| \leq |m_v|(a_1|m_u|^3 + a_2|m_v|) \tag{90}$$

We will again consider the set $\mathcal{A}_{\Gamma,\epsilon} = \{(m_u, m_v)|\, 0 < m_u < \epsilon, |m_v| < \Gamma|m_u|\}$ and repeat the steps of items one and two (assuming again $m_u(0) \geq 0$, otherwise we can flip the sign of $u$), reaching that neither the constraint $m_u(t) \geq 0$ nor $m_v(t) \leq \Gamma m_u(t)$ will be violated for $t \leq T$. Hence it must be that $m_u(T) = \epsilon$, and $T \leq \frac{2}{|C_2|}\log\left(\frac{\epsilon}{m_u(0)}\right)$, since we also have the lower bound $m_u(t) \geq m_u(0)e^{|C_2|t/2}$.

Now we turn to upper bound $|m_v|$, for $\epsilon$ small enough we have the bound

$$\frac{3}{2}C_e m_u^2(t) \leq \frac{\dot{m}_v(t)}{m_v(t)} \leq \frac{1}{2}C_e m_u^2(t) \tag{91}$$

and we can easily see that the dynamics will be bounded between $-m_v(0)$ and $m_v(0)$ for all times, otherwise an application of the mean value theorem would lead to the presence of a time $\tilde{t}$ in which $m_v(\tilde{t}) > 0$ and $\dot{m}_v(\tilde{t}) > 0$ or $m_v(\tilde{t}) < 0$ and $\dot{m}_v(\tilde{t}) < 0$, and both contradict Equation (91).

Then we can conclude the proof repeating the argument from the previous items: since $w(0) \sim \text{Unif}(\mathbb{S}^{d-1})$ then, by Levy's lemma, we can deduce that the probability that $|m_u(0)|, |m_v(0)| \in (d^{-1/2-\hat{\epsilon}}, d^{-1/2+\hat{\epsilon}})$ converges to 1 for any $\hat{\epsilon} > 0$. We can now use lemma A.4 to deduce that given $\delta$, we can choose $\Gamma_\delta$ accordingly so that $|m_v(0)| \leq \Gamma_\delta|m_u(0)|$ has probability larger than $1 - \delta/2$. Once we fixed $\Gamma_\delta$ (which is independent of $d$, thanks to lemma A.4) we can choose $\epsilon$ that satisfies Equation (75). So we proved that the event $(m_u(0), m_v(0)) \in \mathcal{A}_{\epsilon,\Gamma_\delta}$ has probability larger than $1 - \delta$, for $d$ large enough. Conditioning on this event we can then deduce from that $T = \Theta(\log d)$ and that time achieves weak recovery for u $m_u(T) = \epsilon$ while $|m_v(t)| \leq m_v(0) < d^{-1/2+\hat{\epsilon}}$

**Proof of fourth item.** We now turn to the proof of the fourth point: let $(A_d)_{d\in\mathbb{N}}$ be a sequence that converges to 0 as $d \to \infty$ and consider $\mathcal{A}_d = \{(m_u, m_v) \mid |m_u| < A_d, |m_v| < A_d\}$. Suppose $(m_u(0), m_v(0)) \in \mathcal{A}_d$ and we want to derive a lower bound on $T_d := \inf \{t \mid (m_u(t), m_v(t)) \notin \mathcal{A}_d\}$. Note that, since $A_d \to 0$, if $d$ is large enough, $A_d \leq \epsilon$, hence Equation (73) and Equation (74) (or Equation (82)-Equation (81)) are satisfied, so for $t \in (0, T_d)$:

$$\begin{cases} -C_2 m_u(t) - a_1 m_u^2(t) - a_2 m_v^2(t) \leq \dot{m}_u(t) \leq -C_2 m_u(t) + a_1 m_u^2(t) + a_2 m_v^2(t) \\ -C_3 m_u^2(t) - b_1 |m_u(t)|^3 - b_2 |m_v(t)|^3 \leq \dot{m}_v(t) \leq -C_3 m_u^2(t) + b_1 |m_u(t)|^3 + b_2 |m_v(t)|^3 \end{cases} \tag{92}$$

Note that this also holds in case of $C_3 = 0$ for appropriate constants $b_1, b_2$ that depend on $C_{(3,1)}$ and $b$. We now consider the auxiliary function $g$ defined to be the unique solution of the following Cauchy problem:

$$\begin{cases} \dot{g}(t) = \max\big((a_1 + a_2)g^2(t), |C_3|g^2(t) + (b_1 + b_2)|g^3(t)|\big) \\ g(0) = g_0 > \max(|m_u(0)|, |m_v(0)|) \end{cases} \tag{93}$$

Now we state that $g(t) > |m_u(t)|$ and $g(t) > |m_v(t)|$ for all times $t$ in $[0, T_d]$. To show it, let $T_g$ the smallest time such that $|m_u(T_g)| \geq g(T_g)$ or $|m_v(T_g)| \geq g(T_g)$. Suppose by contradiction that $T_g \leq T_d$, then, let us consider separately the two cases:

- if $|m_v(T_g)| \geq g(T_g)$ then consider $T_v := \sup \{0 \leq t < T_g \mid m_v(t) = 0\}$. We have that $|m_v|$ never touches 0 in the interval $(T_v, T_g)$, hence it is differentiable in that interval and we can apply the mean value theorem on $G_v = g - |m_v|$. Note that $G_v(T_v) > 0$ since $g$ is always strictly positive by construction, whereas $G_v(T_g) = 0$. So there must be an intermediate point $\hat{t} \in (T_v, T_g)$ with strictly negative derivative. If $m_v$ is positive in $(T_v, T_g)$ we have that (using Equation (92)) :

$$\begin{aligned} 0 > \dot{G}_v(\hat{t}) &= \dot{g}(\hat{t}) - \dot{m}_v(\hat{t}) \\ &\geq |C_3|g^2(\hat{t}) + (b_1 + b_2)|g^3(\hat{t})| - |C_3|m_u^2(\hat{t}) - (b_1|m_u^3|(\hat{t}) + b_2|m_v^3(\hat{t})|) \\ &= |C_3|(g^2(\hat{t}) - m_u^2(\hat{t})) + b_1(|g^3(\hat{t})| - |m_u^3|(\hat{t})|) + b_2(g^3(\hat{t}) - |m_v^3(\hat{t})|) \end{aligned}$$

  And now since $b_1, b_2 > 0$, this condition violates $|m_u(\hat{t})| < g(\hat{t})$ or $|m_v(\hat{t})| < g(\hat{t})$, so this case can be excluded. A practically identical computations excludes also the case $m_v$ negative.

- if $|m_u(T_g)| \geq g(T_g)$ then consider $T_u := \sup \{0 \leq t < T_g \mid m_u(t) = 0\}$. We have that $|m_u|$ never touches 0 in the interval $(T_u, T_g)$, hence it is differentiable in that interval and we can apply the mean value theorem on $G_u = g - |m_u|$. Note that $G_u(T_u) > 0$ since $g$ is always strictly positive by construction, whereas $G_u(T_g) = 0$. So there must be an intermediate point $\hat{t} \in (T_u, T_g)$ with strictly negative derivative. Now we have two further sub-cases. If $m_u$ is positive in $(T_u, T_g)$, then $|m_u(\hat{t})| = m_u(\hat{t})$ and, using Equation (92)

$$\begin{aligned} 0 > \dot{G}_u(\hat{t}) &= \dot{g}(\hat{t}) - \dot{m}_u(\hat{t}) \\ &\geq (a_1 + a_2)g^2(\hat{t}) - \left(-C_2 m_u(\hat{t}) + a_1 m_u^2(\hat{t}) + a_2 m_v^2(\hat{t})\right) \\ &\geq C_2 m_u(\hat{t}) \end{aligned}$$

  which is a contradiction due to the fact that $C_2 \geq 0$ and $m_u(\hat{t}) \geq 0$. Conversely if $m_u$ is negative in $(T_u, T_g)$, then $|m_u(\hat{t})| = -m_u(\hat{t})$ and

$$\begin{aligned} 0 > \dot{G}_u(\hat{t}) &= \dot{g}(\hat{t}) + \dot{m}_u(\hat{t}) \\ &\geq (a_1 + a_2)g^2(\hat{t}) + \left(-C_2 m_u(\hat{t}) - a_1 m_u^2(\hat{t}) - a_2 m_v^2(\hat{t})\right) \\ &\geq -C_2 m_u(\hat{t}) \end{aligned}$$

  leading again to a contradiction since we are assuming $m_u(\hat{t}) \leq 0$.

So we concluded the proof that $g$ upper bounds the norms of $m_u$ and $m_v$ in $\mathcal{A}_d$. Now note that as long as $g(t) \leq A_d$ then $\dot{g}(t) \leq ag^2(t)$ for some $a > 0$, hence $g(t) \leq \frac{g_0}{1 - ag_0 t}$ which means that then $T_d \geq \frac{1}{ag_0} - \frac{1}{aA_d}$, for all $\max(|m_u(0)|, |m_v(0)|) < g_0 < A_d$.

Since the probability that $\max(|m_u(0)|, |m_v(0)|) \geq \frac{B_d}{\sqrt{d}}$ converges to 1 for any sequence $B_d$ that diverges at $+\infty$ as $d \to \infty$, we can state that with high probability $T_d \geq \frac{d^{1/2}}{aB_d} - \frac{1}{aA_d}$. So given any sequence $t_d = o(\sqrt{d})$ we can tune $B_d$ to diverge slow enough and $A_d$ to converge to 0 slow enough so that $T_d \geq \frac{d^{1/2}}{aB_d} - \frac{1}{aA_d} \geq t_d$, which is what we needed to prove.

### A.4. Details for Table 1 and Theorem 4.2 for different radii

We note that a few columns of Table 1 show activation functions $\sigma$ (such as ReLU and ELU) that are not regular enough to ensure the application of Theorem 4.2. In this case the prediction should be applied to any smooth approximations $\tilde{\sigma}$ that has matching Hermite coefficients: $c_k^{\tilde{\sigma}^2} = c_k^{\sigma^2}$ and $c_k^{z\tilde{\sigma}} = c_k^{z\sigma}$ for all $k \leq 4$.

We emphasize that each entry in Table 1 is to be interpreted as holding for *any latent distribution* $P_{\text{latents}}$ with the specified correlation exponent $k^\star$, with the exception of the entries corresponding to ✗₁. The latter should be understood as exhibiting the existence of counterexamples: there exist explicit constructions of $P_{\text{latents}}$ satisfying the assumption $C_{p,1} = 0$ for all $p \in \mathbb{N}$ for which weak recovery of $\mathbf{v}^\star$ does not occur.

For $\sigma = \tanh$, it is sufficient to impose $\mathbb{E}[\lambda^p \nu] = 0$ for all odd $p$; for instance, choosing $\nu = F(\lambda)$ with $F$ an even function and $\lambda \sim \mathcal{N}(0,1)$ satisfies these conditions. For $\sigma = \text{ReLU}$, we consider the explicit example $\nu = \text{He}_3(\lambda)$, which yields $C_{p,1} = 0$ for all $p$, since $\mathbb{E}_{Z \sim \mathcal{N}(0,1)}[\text{ReLU}^2(Z)\,\text{He}_4(Z)] = 0$.

We also comment on how the predictions of Theorem 4.2 depend on the choice of radius via the rescaled activation $\sigma_r(\xi) = r\sigma(r\xi)$. In general, rescaling can substantially alter the relevant Hermite coefficients and, consequently, the qualitative behavior of the dynamics. As a simple illustration, the activation $\text{He}_1 + \text{He}_2$ falls into case ✗₂ for $r = 1$, but belongs to cases ✓₁ or ✓₂ (depending on $k^\star$) when $r = 1/\sqrt{2}$ (see also Figure 7).

## B. Derivation of the Results 3.1, 3.2, 5.1 from the main text

Our goal in this section is to provide the detailed derivation behind the results shown in sections 3 and 5. We use the replica method (Mézard & Montanari, 2009; Zdeborová & Krzakala, 2016), a heuristic tool from statistical physics. In this section, we perform the computation for a richer class of data settings and learning algorithms than those described in the main text, which include the spiked cumulant model and learning with an autoencoder. We then specify to the setting of the main text and present the associated results in:

- Result 3.1 (characterization of the BO estimator) in Appendix C.1.

- Statement of Approximate Message Passing and its State Evolution in Appendix C.2.1.

- Result 3.2 (weak recovery of AMP) in Appendix C.2, with additional considerations in the case of independent latents.

- Result 5.1 (characterization of the ERM estimator) in Appendix D.

### B.1. Data Model

In this section, we will discuss our data model. We consider a dataset with $n$ samples $\mathbf{x}_\mu \in \mathbb{R}^d$ of dimension $d$, and we assume that $d, n$ grow to infinity proportionally with sample ratio $\alpha = n/d$. Our goal will be to recover $k = \mathcal{O}_d(1)$ hidden high-dimensional vectors in the input, which are stacked in a matrix $\mathbf{w}^\star \in \mathbb{R}^{d \times k}$. We additionally allow for some low-dimensional, sample dependent latent variables $\Lambda_\mu \in \mathbb{R}^p$ with $p = \mathcal{O}_d(1)$, which we are not interested in recovering. The data $\mathbf{x}_\mu$ is then defined as

$$\mathbf{x}_\mu = \left(\mathbb{I} + \frac{\mathbf{w}^\star F_1(\Lambda_\mu)\mathbf{w}^{\star\top}}{d}\right)\mathbf{z}_\mu + \frac{\mathbf{w}^\star F_2(\Lambda_\mu)}{\sqrt{d}}, \tag{94}$$

where $F_1 : \mathbb{R}^p \to \mathbb{R}^{k \times k}$, $F_2 : \mathbb{R}^p \to \mathbb{R}^d$ are two generic functions of the latent variables and $\mathbf{z}_\mu \sim \mathcal{N}(0, \mathbb{I})$ is white Gaussian noise. For $F_1 = F_2 = 0$, the data is just an isotropic Gaussian. $F_2$ introduces a low-rank signal in the directions spanned by the columns of $\mathbf{w}^\star$, and $F_1$ induces a low-rank deformation to the covariance of the noise. The latent variables $\Lambda_\mu$ couple these two effects. We assume that the latents are sampled from a prior $P_{\text{latents}}(\Lambda)$ and $\mathbf{w}^\star \sim P_w^\star(\mathbf{w})$.

The data model we consider in the present work, equation Equation (1), reduces to the one above with $\Lambda_\mu = (\lambda_\mu, \nu_\mu)$ as latent variables, $\mathbf{w}^\star$ standard Gaussian in each component, and:

$$F_1(\lambda, \nu) = \begin{pmatrix} 0 & 0 \\ 0 & -\frac{\mathbb{E}[\nu^2]}{1+\mathbb{E}[\nu^2]+\sqrt{1+\mathbb{E}[\nu^2]}} \end{pmatrix} \quad \text{and} \quad F_2(\lambda, \nu) = \begin{pmatrix} \lambda \\ \frac{\nu}{\sqrt{1+\mathbb{E}[\nu^2]}} \end{pmatrix}. \tag{95}$$

## B.2. High-level description of the technique

In full generality, the tools we will describe characterize the properties of the mean of the distribution

$$P_{\text{likelihood}}(\mathbf{w}) \propto P_w(\mathbf{w}) \prod_{\mu=1}^{n} P_{\text{out}}\left(\frac{\mathbf{w}^\top \mathbf{x}_\mu}{\sqrt{d}}, \frac{\mathbf{w}^\top \mathbf{w}}{d}\right), \tag{96}$$

where $\mathbf{w} \in \mathbb{R}^{d \times p}$.

**Bayesian setting.** If $p = k$, $P_w = P_w^\star$, $P_{\text{out}} = P_{\text{out}}^\star$, where $P_{\text{out}}^\star$ is defined through the identity

$$P^\star(\mathbf{x}|\mathbf{w}) = \mathcal{N}(0, \mathbb{I})[\mathbf{x}]\, P_{\text{out}}^\star\left(\frac{\mathbf{w}^\top \mathbf{x}}{\sqrt{d}}, \frac{\mathbf{w}^\top \mathbf{w}}{d}\right) \tag{97}$$

where $P^\star(\mathbf{x}|\mathbf{w})$ is the conditional probability of generating one sample $\mathbf{x}$ from spikes $\mathbf{w}$, then $P_{\text{likelihood}}$ is the actual posterior distribution, i.e. the probability that a matrix $\mathbf{w}^\star$ generated the given data, and averaging it would describe the Bayes-Optimal estimator w.r.t. the mean-squared-error. Notice that Equation (97) is non-trivial, i.e. the fact that $P_{\text{out}}^\star$ has the declared dependencies is a direct consequence of Equation (94). For our data model, $P_w^\star$ is a standard isotropic Gaussian, and letting $B \equiv \mathbb{I} + 2QF_1(\Lambda) + QF_1(\Lambda)QF_1(\Lambda)$

$$P_{\text{out}}^\star(h, Q) = \mathbb{E}_{\text{latents}}\left[\frac{1}{|\det(\mathbb{I}+QF_1)|} e^{F_2^\top B^{-1}h - \frac{1}{2}F_2^\top B^{-1}QF_2 + \frac{1}{2}h^\top Q^{-1}(\mathbb{I}-B^{-1})h}\right] \tag{98}$$

**Empirical risk minimization setting.** Another possible choice we will make is

$$P_w(\mathbf{w}) = e^{-\beta\, r(\mathbf{w})}, \qquad P_{\text{out}}\left(\frac{\mathbf{w}^\top \mathbf{x}_\mu}{\sqrt{d}}, \frac{\mathbf{w}^\top \mathbf{w}}{d}\right) = e^{-\beta\, \ell\left(\frac{\mathbf{w}^\top \mathbf{x}_\mu}{\sqrt{d}}, \frac{\mathbf{w}^\top \mathbf{w}}{d}\right)}. \tag{99}$$

In the limit of $\beta \to \infty$ the average of $P_{\text{likelihood}}$ will minimize the risk $\mathcal{R}(\mathbf{w})$

$$\mathcal{R}(\mathbf{w}) = \sum_{\mu=1}^{n} \ell\left(\frac{\mathbf{w}^\top \mathbf{x}_\mu}{\sqrt{d}}, \frac{\mathbf{w}^\top \mathbf{w}}{d}\right) + r(\mathbf{w}). \tag{100}$$

This choice will then allow us to study the global minima of the empirical risk Equation (6) for the autoencoder architecture, by setting

$$r(\mathbf{w}) = 0 \quad \text{and} \quad \ell(h, q) = \sigma(h)^\top q\, \sigma(h) - 2h^\top \sigma(h) \tag{101}$$

## B.3. Details of the computation

As it's customary in this kind of analysis, one wishes to analyze the averaged intensive free entropy $\Phi \equiv \mathbb{E}[\log \mathcal{Z}]/d$, where $\mathcal{Z}$ is the normalization of $P_{\text{post}}$

$$\mathcal{Z} = \int d\mathbf{w}\, P_w(\mathbf{w}) \prod_{\mu=1}^{n} P_{\text{out}}\left(\frac{\mathbf{w}^\top \mathbf{x}_\mu}{\sqrt{d}}, \frac{\mathbf{w}^\top \mathbf{w}}{d}\right), \tag{102}$$

and the average is taken over all sources of randomness. We will compute $\Phi$ through the replica *trick*:

$$\mathbb{E}[\log \mathcal{Z}] = \lim_{s \to 0} \frac{\mathbb{E}[\mathcal{Z}^s] - 1}{s}. \tag{103}$$

Indeed, computing the expected moments of $\mathcal{Z}$ is a much easier task than its average logarithm. The $s$-th moment of $\mathcal{Z}$ is

$$\mathcal{Z}^s = \int \prod_{a=1}^{s} d\mathbf{w}^a P_w(\mathbf{w}^a) \prod_{\mu=1}^{n} \prod_{a=1}^{s} P_{\text{out}}\left(\frac{\mathbf{w}^{a\top} \mathbf{x}_\mu}{\sqrt{d}}, \frac{\mathbf{w}^{a\top} \mathbf{w}^a}{d}\right), \tag{104}$$

and its expectation will be:

$$\mathbb{E}[\mathcal{Z}^s] = \mathbb{E}_{\mathbf{w}^\star} \mathbb{E}_{\mathbf{x}|\mathbf{w}^\star}[\mathcal{Z}^s] = \mathbb{E}_{\mathbf{w}^\star} \int \prod_{a=1}^{s} d\mathbf{w}^a P_w(\mathbf{w}^a) \left[\mathbb{E}_{\mathbf{x}|\mathbf{w}^\star}\left[\prod_{a=1}^{s} P_{\text{out}}\left(\frac{\mathbf{w}^{a\top} \mathbf{x}}{\sqrt{d}}, \frac{\mathbf{w}^{a\top} \mathbf{w}^a}{d}\right)\right]\right]^n. \tag{105}$$

where we used the law of total expectation and the assumption that the samples are taken i.i.d..

Consider the inner expectation: even though it is an expectation over a high-dimensional $\mathbf{x}$, the function only depends on the low dimensional projection $\mathbf{w}^{a\top}\mathbf{x} \in \mathbb{R}^p$. We can use this to write this as a low-dimensional expectation, which will decouple over the index $a$.

Using the identity Equation (97) we have

$$\mathbb{E}_{\mathbf{x}|\mathbf{w}^\star}\left[\prod_{a=1}^{s} P_{\text{out}}\left(\frac{\mathbf{w}^{a\top}\mathbf{x}}{\sqrt{d}}, \frac{\mathbf{w}^{a\top}\mathbf{w}^a}{d}\right)\right] = \mathbb{E}_{\mathbf{x}\sim\mathcal{N}(0,\mathbb{I})}\left[P_{\text{out}}^\star\left(\frac{\mathbf{w}^{\star\top}\mathbf{x}}{\sqrt{d}}, \frac{\mathbf{w}^{\star\top}\mathbf{w}^\star}{d}\right)\prod_{a=1}^{s} P_{\text{out}}\left(\frac{\mathbf{w}^{a\top}\mathbf{x}}{\sqrt{d}}, \frac{\mathbf{w}^{a\top}\mathbf{w}^a}{d}\right)\right]. \quad (106)$$

We now notice that the random variables $(h^\star, h^a)$ defined by

$$(h^\star, h^a) \equiv \left(\frac{\mathbf{w}^{\star\top}\mathbf{x}}{\sqrt{d}}, \frac{\mathbf{w}^{a\top}\mathbf{x}}{\sqrt{d}}\right) \in \mathbb{R}^{k\times(p\times s)}, \quad (107)$$

are Gaussian with mean zero and covariances that depend on the matrices $Q^\star \in \mathbb{R}^{k\times k}$, $Q^{ab} \in \mathbb{R}^{p\times p}$, $M^a \in \mathbb{R}^{p\times k}$ for $1 \le a, b \le s$:

$$\mathbb{E}[h^\star h^{\star\top}] = Q^\star, \qquad \mathbb{E}[h^a h^{b\top}] = Q^{ab}, \qquad \mathbb{E}[h^a h^{\star\top}] = M^a. \quad (108)$$

All this allows us to do following decomposition of the integral in Equation (105):

$$\mathbb{E}[\mathcal{Z}^s] = \mathbb{E}_{\mathbf{w}^\star}\int\prod_{a=1}^{s} d\mathbf{w}^a P_w(\mathbf{w}^a)\left[\mathbb{E}_{\mathbf{x}|\mathbf{w}^\star}\left[\prod_{a=1}^{s} P_{\text{out}}\left(\frac{\mathbf{w}^{a\top}\mathbf{x}}{\sqrt{d}}, \frac{\mathbf{w}^{a\top}\mathbf{w}^a}{d}\right)\right]\right]^n$$

$$= \mathbb{E}_{\mathbf{w}^\star}\int\prod_{a=1}^{s} d\mathbf{w}^a P_w(\mathbf{w}^a)\left[\mathbb{E}_{\mathbf{x}\sim\mathcal{N}(0,\mathbb{I})}\left[P_{\text{out}}^\star\left(\frac{\mathbf{w}^{\star\top}\mathbf{x}}{\sqrt{d}}, \frac{\mathbf{w}^{\star\top}\mathbf{w}^\star}{d}\right)\prod_{a=1}^{s} P_{\text{out}}\left(\frac{\mathbf{w}^{a\top}\mathbf{x}}{\sqrt{d}}, \frac{\mathbf{w}^{a\top}\mathbf{w}^a}{d}\right)\right]\right]^n = \quad (109)$$

$$= \mathbb{E}_{\mathbf{w}^\star}\int\prod_{a=1}^{s} d\mathbf{w}^a P_w(\mathbf{w}^a)\left[\mathbb{E}_{(h^\star, h^a)}\left[P_{\text{out}}^\star(h^\star, Q^\star(\mathbf{w}^\star))\prod_{a=1}^{s} P_{\text{out}}(h^a, Q^{aa}(\mathbf{w}^a))\right]\right]^n$$

where the last expectation is over the random variables $(h^a, h^\star)$. Notice how the matrices $Q^{ab}$, $M^a$ still depend on $\mathbf{w}\star$, $\mathbf{w}^a$. We can decouple them by introducing some Dirac deltas, giving us

$$\mathbb{E}[\mathcal{Z}^s] = \int dQ^{ab}\, dM^a\, dQ^\star\left[\mathbb{E}_{\mathbf{w}^\star}\int\prod_{a=1}^{s} d\mathbf{w}^a P_w(\mathbf{w}^a)\prod_{a\le b}\delta(dQ^{ab} - \mathbf{w}^{a\top}\mathbf{w}^b)\times\right.$$

$$\times\prod_{a=1}^{s}\delta(dM^a - \mathbf{w}^{a\top}\mathbf{w}^\star)\,\delta(dQ^\star - \mathbf{w}^{\star\top}\mathbf{w}^{\star\top})\right] \quad (110)$$

$$\left[\mathbb{E}_{(h^\star, h^a)}\left[P_{\text{out}}^\star(h^\star, Q^\star(\mathbf{w}^\star))\prod_{a=1}^{s} P_{\text{out}}(h^a, Q^{aa}(\mathbf{w}^a))\right]\right]^n$$

where by symmetry we imposed Dirac deltas only for $a \le b$. Similarly, the integration is intended over all $(a, b)$ for $a \le b$. We will now mostly focus on simplifying the first square bracket. Introducing the Lagrange multipliers $\hat{Q}^\star, \hat{M}^a, \hat{Q}^{ab}$, we write

$$\prod_{a\le b}\delta(dQ^{ab} - \mathbf{w}^{a\top}\mathbf{w}^b)\prod_{a=1}^{s}\delta(dM^a - \mathbf{w}^{a\top}\mathbf{w}^\star)\,\delta(dQ^\star - \mathbf{w}^{\star\top}\mathbf{w}^\star) = \int d\hat{Q}^{ab}\, d\hat{M}^a\, d\hat{Q}^\star.$$

$$\exp\left\{d\sum_{a\le b}\text{Tr}(\hat{Q}^{ab}Q^{ab\top}) + d\sum_{a=1}^{s}\text{Tr}(\hat{M}^a M^{a\top}) + d\text{Tr}(\hat{Q}^\star Q^{\star\top})\right. \quad (111)$$

$$\left. -\text{Tr}(\hat{Q}^{\star\top}\mathbf{w}^{\star\top}\mathbf{w}^\star) - \sum_{a\le b}\text{Tr}(\hat{Q}^{ab\top}\mathbf{w}^{a\top}\mathbf{w}^b) - \sum_{a=1}^{s}\text{Tr}(\hat{M}^{a\top}\mathbf{w}^{a\top}\mathbf{w}^\star)\right\}.$$

The integral over $\hat{Q}^\star$, $Q^\star$ is actually not needed. Let's compute the expected zeroth moment of $\mathcal{Z}$, which is one. Notice that here we literally ask for $s = 0$, and not for the limit $s \to 0$, as we will do later. From the derivation above we get

$$\mathbb{E}[\mathcal{Z}^0] = \int d\hat{Q}^\star \, dQ^\star \exp\left\{ d\mathrm{Tr}(\hat{Q}^\star Q^{\star\top}) - \mathrm{Tr}(\hat{Q}^{\star\top} \mathbf{w}^{\star\top} \mathbf{w}^\star) \right\} = 1 \,. \tag{112}$$

In the large $d$ limit we can use the saddle point method to see that the integral is dominated by $\hat{Q}^\star = \mathbb{O}$ (zero matrix) and

$$Q^\star = \mathbb{E}_{\mathbf{w}^\star}\left[ \frac{\mathbf{w}^{\star\top} \mathbf{w}^\star}{d} \right] \,. \tag{113}$$

We will thus fix these two variables. Gathering all our progress, we can write

$$\mathbb{E}[\mathcal{Z}^s] = \int dM^a \, d\hat{M}^a \, dQ^{ab} \, d\hat{Q}^{ab} \exp\left\{ d(\Phi_{\text{trace}} + \Phi_{\text{prior}} + \alpha \Phi_{\text{output}}) \right\} \tag{114}$$

where

$$\Phi_{\text{trace}} = \sum_{a \leq b} \mathrm{Tr}(\hat{Q}^{ab} Q^{ab\top}) + \sum_{a=1}^s \mathrm{Tr}(\hat{M}^a M^{a\top})$$

$$\Phi_{\text{prior}} = \frac{1}{d} \log \mathbb{E}_{\mathbf{w}^\star}\left[ \int \prod_{a=1}^s d\mathbf{w}^a P_w(\mathbf{w}^a) \exp\left\{ -\sum_{a \leq b} \mathrm{Tr}(\hat{Q}^{ab\top} \mathbf{w}^{a\top} \mathbf{w}^b) - \sum_{a=1}^s \mathrm{Tr}(\hat{M}^{a\top} \mathbf{w}^{a\top} \mathbf{w}^\star) \right\} \right] \tag{115}$$

$$\Phi_{\text{output}} = \log \mathbb{E}_{(h^\star, h^a)}\left[ P_{\text{out}}^\star(h^\star, Q^\star) \prod_{a=1}^s P_{\text{out}}(h^a, Q^{aa}) \right]$$

We will eventually need to take the limit $s \to 0$. On the other hand, $s$ is an integer, so to take the limit we will first assume a specific structure on $Q^{ab}$, $M^a$, $\hat{Q}^{ab}$, $\hat{M}^a$, and then do an analytic continuation for real valued $s$. In particular we assume the so-called *replica symmetric ansatz*

$$\begin{aligned} M^a &= m, \\ Q^{ab} &= (r - q)\delta_{ab} + q, \\ \hat{M}^a &= -\hat{m}, \\ \hat{Q}^{ab} &= \left( \frac{\hat{r}}{2} + \hat{q} \right) \delta_{ab} - \hat{q}. \end{aligned} \tag{116}$$

The trace piece is the simplest one to simplify in the $s \to 0$ limit

$$\Phi_{\text{trace}} = s\left[ -\mathrm{Tr}(m^\top \hat{m}) + \frac{1}{2}\mathrm{Tr}(r^\top \hat{r}) + \frac{1}{2}\mathrm{Tr}(q^\top \hat{q}) \right] + \mathcal{O}(s^2) \,. \tag{117}$$

Let's move to the prior piece. First, we use our ansatz to write

$$\Phi_{\text{prior}} = \frac{1}{d} \log \mathbb{E}_{\mathbf{w}^\star}\left[ \int \prod_{a=1}^s d\mathbf{w}^a P_w(\mathbf{w}^a) \exp\left\{ \sum_{a \leq b} \mathrm{Tr}(\hat{q}^\top \mathbf{w}^{a\top} \mathbf{w}^b) - \frac{1}{2}\sum_{a=1}^s \mathrm{Tr}(\hat{r}^\top \mathbf{w}^{a\top} \mathbf{w}^b) + \sum_{a=1}^s \mathrm{Tr}(\hat{m}^\top \mathbf{w}^{a\top} \mathbf{w}^\star) \right\} \right] \tag{118}$$

We will now do a number of manipulations in order to decouple the first piece in the exponent over the index $a$. Let's call $\mathbf{w}_i$ for $i = 1, ..., d$ the rows of $\mathbf{w}$. We then have

$$\exp\left\{ -\sum_{a \leq b} \mathrm{Tr}(\hat{q} \, \mathbf{w}^{a\top} \mathbf{w}^b) \right\} = \prod_{i=1}^d \exp\left\{ -\sum_{a \leq b} \mathbf{w}_i^{a\top} \hat{q} \, \mathbf{w}_i^b \right\} = \prod_{i=1}^d \exp\left\{ -\sum_{a \leq b} (\mathbf{w}_i^a \hat{q}^{1/2}) \, (\mathbf{w}_i^b \hat{q}^{1/2})^\top \right\} \,. \tag{119}$$

Using the identity for any set of vectors $\mathbf{v}_a$

$$\sum_{a \leq b} \mathbf{v}_i^{a\top} \mathbf{v}_i^b = \frac{1}{2} \left\| \sum_{a=1}^s \mathbf{v}_i^a \right\|^2 - \frac{1}{2} \sum_{a=1}^s \|\mathbf{v}_i^a\|^2 , \tag{120}$$

as well as the following Gaussian identity true for every vector $v$

$$\exp\left\{ \frac{v^\top v}{2} \right\} = \mathbb{E}_{\xi \sim \mathcal{N}(0, \mathbb{I})} \left[ \exp\left\{ v^\top \xi \right\} \right] , \tag{121}$$

one can write

$$
\begin{aligned}
\exp\left\{ -\sum_{a \leq b} \text{Tr}(\hat{q}\,\mathbf{w}^{a\top} \mathbf{w}^b) \right\} &= \prod_{i=1}^d \exp\left\{ \frac{1}{2} \left\| \sum_{a=1}^s \hat{q}^{1/2} \mathbf{w}_i^a \right\|^2 - \frac{1}{2} \sum_{a=1}^s \|\hat{q}^{1/2} \mathbf{w}_i^a\|^2 \right\} \\
&= \prod_{i=1}^d \mathbb{E}_\xi \left[ \prod_{a=1}^s \exp\left\{ \mathbf{w}_i^{a\top} \hat{q}^{1/2} \xi_i - \frac{1}{2} \mathbf{w}_i^{a\top} \hat{q}\, \mathbf{w}_i^a \right\} \right] \\
&= \mathbb{E}_\Xi \left[ \prod_{a=1}^s \exp\left\{ \text{Tr}(\mathbf{w}^{a\top} \hat{q}^{1/2} \Xi) - \frac{1}{2} \text{Tr}(\mathbf{w}^{a\top} \hat{q}\, \mathbf{w}^a) \right\} \right] ,
\end{aligned}
\tag{122}
$$

where $\Xi \in \mathbb{R}^{d \times p}$ is Gaussian in every entry. The whole expression will then be, in the limit $s \to 0$

$$
\begin{aligned}
\Phi_{\text{prior}} &= \frac{1}{d} \log \mathbb{E}_{\Xi, \mathbf{w}^\star} \left[ \int \prod_{a=1}^s d\mathbf{w}^a P_w(\mathbf{w}^a) \exp\left\{ \text{Tr}(\mathbf{w}^{a\top} \hat{q}^{1/2} \Xi) - \frac{1}{2} \text{Tr}(\mathbf{w}^{a\top} (\hat{q} + \hat{r})\, \mathbf{w}^a) + \text{Tr}(\hat{m}^\top \mathbf{w}^{a\top} \mathbf{w}^\star) \right\} \right] \\
&= \frac{s}{d} \log \mathbb{E}_{\Xi, \mathbf{w}^\star} \left[ \int d\mathbf{w} P_w(\mathbf{w}) \exp\left\{ \text{Tr}(\mathbf{w}^\top \hat{q}^{1/2} \Xi) - \frac{1}{2} \text{Tr}(\mathbf{w}^\top (\hat{q} + \hat{r})\, \mathbf{w}) + \text{Tr}(\hat{m}^\top \mathbf{w}^\top \mathbf{w}^\star) \right\} \right] \\
&= \frac{s}{d} \mathbb{E}_{\Xi, \mathbf{w}^\star} \left[ \log \int d\mathbf{w} P_w(\mathbf{w}) \exp\left\{ \text{Tr}(\mathbf{w}^\top \hat{q}^{1/2} \Xi) - \frac{1}{2} \text{Tr}(\mathbf{w}^\top (\hat{q} + \hat{r})\, \mathbf{w}) + \text{Tr}(\hat{m}^\top \mathbf{w}^\top \mathbf{w}^\star) \right\} \right] + \mathcal{O}(s^2) .
\end{aligned}
\tag{123}
$$

The last piece, $\Phi_{\text{output}}$, is the most challenging. With our ansatz, it will be

$$\Phi_{\text{output}} = \log \mathbb{E}_{(h^\star, h^a)} \left[ P_{\text{out}}^\star (h^\star, Q^\star) \prod_{a=1}^s P_{\text{out}} (h^a, r) \right] . \tag{124}$$

To simplify it further, we need to look at the covariance of $(h^\star, h^a)$ across all $a$. Recall that $(h^\star, h^a) \sim \mathcal{N}(0, \Sigma)$, with

$$\Sigma = \begin{pmatrix} Q^\star & M^T \\ M & Q \end{pmatrix} = \begin{pmatrix} Q^\star & m^\top & \dots & m^\top \\ m & r & \dots & q \\ \vdots & \vdots & \ddots & \vdots \\ m & q & \dots & r \end{pmatrix} \in \mathbb{R}^{(k+sp) \times (k+sp)} \tag{125}$$

Its inverse is

$$\Sigma^{-1} = \begin{pmatrix} \tilde{Q}^\star & \tilde{m}^\top & \dots & \tilde{m}^\top \\ \tilde{m} & \tilde{r} & \dots & \tilde{q} \\ \vdots & \vdots & \ddots & \vdots \\ \tilde{m} & \tilde{q} & \dots & \tilde{r} \end{pmatrix} \in \mathbb{R}^{(K+sp) \times (K+sp)} \tag{126}$$

where $V \equiv r - q$ and

$$
\begin{aligned}
\tilde{Q}^\star &= (Q^\star - sm^\top (V + sq)^{-1} m)^{-1} , \\
\tilde{r} &= V^{-1} + (V + sq)^{-1} (m\tilde{Q}^\star m^\top (V + sq)^{-1} - qV^{-1}) , \\
\tilde{q} &= \tilde{r} - V^{-1} , \\
\tilde{m} &= -(V + sq)^{-1} m\tilde{Q}^\star .
\end{aligned}
\tag{127}
$$

The determinant of $\Sigma$ is:

$$\log \det \Sigma = (s-1) \log \det V + \log \det(V + sq) + \log \det(Q^\star - sm^\top (V + sq)^{-1} m). \tag{128}$$

We are now ready to explicitly write the distribution of $h^\star$, $h^a$:

$$\exp \left\{ -\frac{1}{2} \log \det(2\pi\Sigma) - \frac{1}{2} h^{\star\top} \tilde{Q}^\star h^\star - \sum_{a=1}^{s} h^{a\top} \tilde{m} \, h^\star - \frac{1}{2} \sum_{a=1}^{s} h^{a\top} (\tilde{r} - \tilde{q}) h^a - \frac{1}{2} \sum_{a,b=1}^{s} h^{a\top} \tilde{q} \, h^b \right\}. \tag{129}$$

The last piece gets decoupled using the same Gaussian trick as before

$$\exp \left\{ -\frac{1}{2} \sum_{a,b=1}^{s} h^{a\top} \tilde{q} \, h^b \right\} = \exp \left\{ \frac{1}{2} \left\| \sum_{a=1}^{s} (-\tilde{q})^{1/2} h^a \right\|^2 \right\} = \mathbb{E}_{\xi \sim \mathcal{N}(0,\mathbb{I})} \left[ \exp \left\{ \sum_{a=1}^{s} \xi^\top (-\tilde{q})^{1/2} h^a \right\} \right], \tag{130}$$

where we assumed $-\tilde{q}$ is positive semi-definite. We thus have

$$\Phi_{\text{output}} = \log \mathbb{E}_\xi \int dh^\star e^{-\frac{1}{2} h^{\star\top} \tilde{Q}^\star h^\star - \frac{1}{2} \log \det (2\pi\Sigma)} P_{\text{out}}^\star (h^\star, Q^\star) \times$$
$$\times \left[ \int dh P_{\text{out}} (h, r) \exp \left\{ -h^\top \tilde{m} \, h^\star - \frac{1}{2} h^\top V^{-1} h + \xi^\top (-\tilde{q})^{1/2} h \right\} \right]^s. \tag{131}$$

Now, we should take the $s \to 0$ limit, but this is particularly challenging, as all the tilde variables depend on $s$. Let's complete the square in the exponential

$$\exp \left\{ -\frac{1}{2} h^\top V^{-1} h + \left[ \tilde{m} h^\star + \xi^\top (-\tilde{q})^{1/2} \right] h \right\} = \exp \left\{ -\frac{1}{2} \left[ h - V \left( (-\tilde{q})^{1/2} \xi - \tilde{m} h^\star \right) \right]^\top V^{-1} \times \left[ h - V \left( (-\tilde{q})^{1/2} \xi - \tilde{m} h^\star \right) \right] \right\}$$
$$\times \exp \left\{ \left[ (-\tilde{q})^{1/2} \xi - \tilde{m} h^\star \right]^\top V \left[ (-\tilde{q})^{1/2} \xi - \tilde{m} h^\star \right] \right\}. \tag{132}$$

Using the following limits

$$\lim_{s \to 0} \tilde{Q}^\star = Q^{\star-1},$$
$$\lim_{s \to 0} -\tilde{q} = V^{-1} (q - m Q^{\star-1} m^\top) V^{-1} = -\tilde{q}_0, \tag{133}$$
$$\lim_{s \to 0} \tilde{m} = -V^{-1} m \tilde{Q}^{\star-1} = \tilde{m}_0,$$

and

$$\log \det \Sigma = \log \det Q^\star + s \left[ \log \det V + \text{Tr}(V^{-1} q) - \text{Tr}(Q^{\star-1} m^\top V^{-1} m) \right] + \mathcal{O}(s^2), \tag{134}$$

one can show that

$$\Phi_{\text{output}} = \log \mathbb{E}_\xi \int dh^\star e^{-\frac{1}{2} h^{\star\top} Q^{\star-1} h^\star - \frac{1}{2} \log \det (2\pi Q^\star)} P_{\text{out}}^\star (h^\star, Q^\star)$$
$$\left[ \int dh P_{\text{out}} (h, r) \exp \left\{ -\frac{1}{2} \left[ h - V \left( (-\tilde{q}_0)^{1/2} \xi - \tilde{m}_0 h^\star \right) \right]^\top \times \right. \right.$$
$$\left. \left. \times V^{-1} \left[ h - V \left( (-\tilde{q}_0)^{1/2} \xi - \tilde{m}_0 h^\star \right) \right] - \frac{1}{2} \log \det(2\pi V) \right\} \right]^s \tag{135}$$

There is a nicer way of writing this result after the substitution

$$V \left( (-\tilde{q}_0)^{1/2} \xi - \tilde{m}_0 h^\star \right) \to q^{1/2} \xi, \tag{136}$$

which after some manipulations gives

$$\Phi_{\text{output}} = \log \mathbb{E}_\xi \int dh^\star e^{-\frac{1}{2}(h^\star - m^\top q^{-1/2}\xi)^\top (Q^\star - m^\top q^{-1}m)^{-1}(h^\star - m^\top q^{-1/2}\xi) - \frac{1}{2}\log\det(2\pi(Q^\star - m^\top q^{-1}m))} P_{\text{out}}^\star(h^\star, Q^\star)$$
$$\left[\int dh P_{\text{out}}(h, r) \exp\left\{-\frac{1}{2}(h - q^{1/2}\xi)^\top V^{-1}(h - q^{1/2}\xi) - \frac{1}{2}\log\det(2\pi V)\right\}\right]^s. \tag{137}$$

For $s \to 0$ we have

$$\Phi_{\text{output}} = \mathbb{E}_\xi \int dh^\star e^{-\frac{1}{2}(h^\star - m^\top q^{-1/2}\xi)^\top (Q^\star - m^\top q^{-1}m)^{-1}(h^\star - m^\top q^{-1/2}\xi) - \frac{1}{2}\log\det(2\pi(Q^\star - m^\top q^{-1}m))} P_{\text{out}}^\star(h^\star, Q^\star)$$
$$\log \int dh P_{\text{out}}(h, r) \exp\left\{-\frac{1}{2}(h - q^{1/2}\xi)^\top V^{-1}(h - q^{1/2}\xi) - \frac{1}{2}\log\det(2\pi V)\right\}. \tag{138}$$

We are at the end of the computation. The averaged intensive free entropy can be written in terms of $m$, $q$, $V$, $\hat{m}$, $\hat{q}$, $\hat{V}$ using the saddle point method, where $\hat{V} = \hat{r} + \hat{q}$:

$$\max_{m, q, V, \hat{m}, \hat{q}, \hat{V}} \Phi(m, q, V, \hat{m}, \hat{q}, \hat{V}), \tag{139}$$

$$\Phi(m, q, V, \hat{m}, \hat{q}, \hat{V}) = \Phi_{\text{trace}}(m, q, V, \hat{m}, \hat{q}, \hat{V}) + \Phi_{\text{prior}}(\hat{m}, \hat{q}, \hat{V}) + \alpha\Phi_{\text{output}}(m, q, V)$$

with

$$\Phi_{\text{trace}} = -\text{Tr}(m^\top \hat{m}) + \frac{1}{2}\text{Tr}(q^\top \hat{V}) + \frac{1}{2}\text{Tr}(V^\top(\hat{V} - \hat{q})),$$

$$\Phi_{\text{prior}} = \frac{1}{d}\mathbb{E}_{\Xi, \mathbf{w}^\star}\left[\log \int d\mathbf{w} P_w(\mathbf{w}) \exp\left\{\text{Tr}(\mathbf{w}^\top \hat{q}^{1/2}\Xi) - \frac{1}{2}\text{Tr}(\mathbf{w}^\top \hat{V} \mathbf{w}) + \text{Tr}(\hat{m}^\top \mathbf{w}^\top \mathbf{w}^\star)\right\}\right],$$

$$\Phi_{\text{output}} = \mathbb{E}_\xi \int dh^\star e^{-\frac{1}{2}(h^\star - m^\top q^{-1/2}\xi)^\top (Q^\star - m^\top q^{-1}m)^{-1}(h^\star - m^\top q^{-1/2}\xi) - \frac{1}{2}\log\det(2\pi(Q^\star - m^\top q^{-1}m))} P_{\text{out}}^\star(h^\star, Q^\star)$$
$$\log \int dh P_{\text{out}}(h, V + q) \exp\left\{-\frac{1}{2}(h - q^{1/2}\xi)^\top V^{-1}(h - q^{1/2}\xi) - \frac{1}{2}\log\det(2\pi V)\right\}. \tag{140}$$

### B.4. Specialization of Equation (140) for Bayes-Optimal estimators

We start by imposing that $P_w = P_w^\star = \mathcal{N}(0, \mathbb{I})$. This enables us to simplify $\Phi_{\text{prior}}$ drastically, as the integrals over $\Xi$, $\mathbf{w}$, $\mathbf{w}^\star$ are all Gaussian. After a tedious computation one gets

$$\Phi_{\text{prior}} = -\frac{1}{2}\log\det(\mathbb{I} + \hat{V}) + \frac{1}{2}\text{Tr}[(\mathbb{I} + \hat{V})^{-1}(\hat{q} + \hat{m}\hat{m}^\top)] \tag{141}$$

Next, we impose the Nishimori conditions $m = q$, $r = Q^\star = \mathbb{I}$, which implies $\hat{m} = \hat{q}$, $V = \mathbb{I} - q$ and $\hat{V} = \hat{q}$, which gives

$$\Phi_{\text{trace}} = -\frac{1}{2}\text{Tr}(q^\top \hat{q}),$$

$$\Phi_{\text{prior}} = -\frac{1}{2}\log\det(\mathbb{I} + \hat{q}) + \frac{1}{2}\text{Tr}(\mathbb{I} + \hat{q}),$$

$$\Phi_{\text{output}} = \mathbb{E}_\xi \int dh^\star e^{-\frac{1}{2}(h^\star - q^{1/2}\xi)^\top (\mathbb{I} - q)^{-1}(h^\star - q^{1/2}\xi) - \frac{1}{2}\log\det(2\pi(\mathbb{I} - q))} P_{\text{out}}^\star(h^\star, \mathbb{I})$$
$$\log \int dh P_{\text{out}}^\star(h, \mathbb{I}) \exp\left\{-\frac{1}{2}(h - q^{1/2}\xi)^\top (\mathbb{I} - q)^{-1}(h - q^{1/2}\xi) - \frac{1}{2}\log\det(2\pi(\mathbb{I} - q))\right\}. \tag{142}$$

There is an alternative writing of $\Phi_{\text{output}}$ using some auxiliary functions. Define $\mathcal{Z}_{\text{out}}$, $f_{\text{out}}$ as

$$\mathcal{Z}_{\text{out}}^\star(\omega, V, Q) = \mathbb{E}_{h \sim \mathcal{N}(\omega, V)}[P_{\text{out}}^\star(h, Q)], \qquad f_{\text{out}}^\star(\omega, V, Q) = \mathbb{E}_{h \sim \mathcal{N}(\omega, V)}[P_{\text{out}}^\star(h, Q)V^{-1}(h - \omega)] \tag{143}$$

which gives us

$$\Phi_{\text{output}} = \mathbb{E}_\xi\left[\mathcal{Z}_{\text{out}}^\star(q^{1/2}\xi, \mathbb{I} - q, \mathbb{I}) \log \mathcal{Z}_{\text{out}}^\star(q^{1/2}\xi, \mathbb{I} - q, \mathbb{I})\right]. \tag{144}$$

The saddle point method gives us two equations that $q$, $\hat{q}$ have to satisfy

$$\begin{cases} q = \hat{q}(\mathbb{I}+\hat{q})^{-1}\,, \\ \hat{q} = \alpha\,\mathbb{E}_\xi\left[ Z^\star_{\text{out}}(q^{1/2}\xi, \mathbb{I}-q, \mathbb{I})f^\star_{\text{out}}(q^{1/2}\xi, \mathbb{I}-q, \mathbb{I})\,f^\star_{\text{out}}(q^{1/2}\xi, \mathbb{I}-q, \mathbb{I})^\top\right]\,. \end{cases} \tag{145}$$

## B.5. Specialization of Equation (140) for minimizers of the empirical risk

We start from Equation (140) and substitute in $P_w$ and $P_{\text{out}}$ from Equation (99). Starting with $\Phi_{\text{prior}}$, we have

$$\Phi_{\text{prior}} = \frac{1}{d}\mathbb{E}_{\Xi,\mathbf{w}^\star}\left[\log\int d\mathbf{w}\,\exp\left\{-\beta\,r(\mathbf{w}) + \text{Tr}(\mathbf{w}^\top\hat{q}^{1/2}\Xi) - \frac{1}{2}\text{Tr}(\mathbf{w}^\top\hat{V}\,\mathbf{w}) + \text{Tr}(\hat{m}^\top\mathbf{w}^\top\mathbf{w}^\star)\right\}\right]. \tag{146}$$

This prompts us to rescale the order parameters

$$\hat{m}\to\beta\,\hat{m}, \qquad \hat{V}\to\beta\,\hat{V}, \qquad \hat{q}\to\beta^2\,\hat{q}\,, \tag{147}$$

and take the limit $\beta\to\infty$. The integral over $\mathbf{w}$ will be dominated by the minimum of the exponential:

$$\mathbf{w} = \arg\min_{\mathbf{w}}\left\{r(\mathbf{w}) - \text{Tr}(\mathbf{w}^\top\hat{q}^{1/2}\Xi) + \frac{1}{2}\text{Tr}(\mathbf{w}^\top\hat{V}\,\mathbf{w}) - \text{Tr}(\hat{m}^\top\mathbf{w}^\top\mathbf{w}^\star)\right\}. \tag{148}$$

Our choice of regularization will be $r(\mathbf{w}) = \tilde{\lambda}\|\mathbf{w}\|^2/2$, for which we have

$$\Phi_{\text{prior}} = \frac{\beta}{2}\text{Tr}[(\hat{V} + \tilde{\lambda}\,\mathbb{I})(\hat{q} + \hat{m}\hat{m}^\top)]. \tag{149}$$

The output piece will be

$$\Phi_{\text{output}} = \mathbb{E}_\xi\int dh^\star e^{-\frac{1}{2}(h^\star - m^\top q^{-1/2}\xi)^\top(Q^\star - m^\top q^{-1}m)^{-1}(h^\star - m^\top q^{-1/2}\xi) - \frac{1}{2}\log\det\,(2\pi(Q^\star - m^\top q^{-1}m))}P^\star_{\text{out}}(h^\star, Q^\star)$$
$$\log\int dh\,\exp\left\{-\beta\ell(h, V+q) - \frac{1}{2}(h - q^{1/2}\xi)^\top V^{-1}(h - q^{1/2}\xi) - \frac{1}{2}\log\det(2\pi V)\right\} \tag{150}$$

which prompts us to take $V\to V/\beta$. As before the integral over $h$ is dominated by the minimum, which we can write using the proximal operator $\text{prox}_\ell(\omega, V, Q)$

$$\text{prox}_\ell(\omega, V, q) = \arg\min_h\left\{\ell(h, q) + \frac{1}{2}(h - \omega)^\top V^{-1}(h - \omega)\right\}. \tag{151}$$

The whole integral will then depend on the Moreau envelope $\mathcal{M}_\ell(\omega, V, Q)$

$$\mathcal{M}_\ell(\omega, V, q) = \min_h\left\{\ell(h, q) + \frac{1}{2}(h - \omega)^\top V^{-1}(h - \omega)\right\} \tag{152}$$

which gives

$$\Phi_{\text{output}} = -\beta\mathbb{E}_\xi\left[\mathcal{Z}^\star_{\text{out}}(m^\top q^{-1/2}\xi, Q^\star - m^\top q^{-1}m, q)\,\mathcal{M}_\ell(q^{1/2}\xi, V, q)\right]. \tag{153}$$

We can then obtain after a tedious computation the saddle point equations

$$\begin{cases} m = (\hat{V} + \tilde{\lambda}\,\mathbb{I})^{-1}\hat{m}, \\ q = (\hat{V} + \tilde{\lambda}\,\mathbb{I})^{-1}(\hat{q} + \hat{m}\hat{m}^\top)(\hat{V} + \tilde{\lambda}\,\mathbb{I})^{-1}, \\ V = (\hat{V} + \tilde{\lambda}\,\mathbb{I})^{-1}, \\ \hat{m} = \alpha\mathbb{E}_\xi[\mathcal{Z}^\star_{\text{out}}(m^\top q^{-1/2}\xi, Q^\star - m^\top q^{-1}m, Q^\star)\,f_{\text{out}}(\sqrt{q}\xi, V, q)\,f^{\star\top}_{\text{out}}(m^\top q^{-1/2}\xi, Q^\star - m^\top q^{-1}m, Q^\star)], \\ \hat{q} = \alpha\mathbb{E}_\xi\left[\mathcal{Z}^\star_{\text{out}}(m^\top q^{-1/2}\xi, Q^\star - m^\top q^{-1}m, Q^\star)\,\left(f_{\text{out}}(\sqrt{q}\xi, V, q)^{\otimes2}\right)\right], \\ \hat{V} = \alpha\mathbb{E}_\xi[\mathcal{Z}^\star_{\text{out}}(m^\top q^{-1/2}\xi, Q^\star - m^\top q^{-1}m, Q^\star)(2\partial_q l(\text{prox}_\ell(\omega, V, q), q) - f_{\text{out}}(\sqrt{q}\xi, V, q)\xi^\top q^{-1/2}))] + \hat{m}m^\top q^{-1}, \end{cases} \tag{154}$$

where we defined

$$f_{\text{out}}(\omega, V, q) = V^{-1}(\text{prox}_\ell(\omega, V, q) - \omega). \tag{155}$$

### B.6. Test and Train Loss

The test loss or generalization error is defined in full generality as:

$$\epsilon_g = \mathbb{E}_x \left[ \ell \left( \frac{\mathbf{w}^{\star\top}\mathbf{x}}{\sqrt{d}}, \frac{\mathbf{w}^\top\mathbf{x}}{\sqrt{d}}, \frac{\mathbf{w}^\top\mathbf{w}}{d}, \frac{\mathbf{w}^{\star\top}\mathbf{w}^\star}{d} \right) \right] \tag{156}$$

where $x$ is intended to be independent from the train set. This is a quantity that will depend on the solutions $m, q, V$ of the saddle point equations. This dependence appears through its arguments, that we assume concentrate to the following quantities in the $d \to \infty, \beta \to \infty$ limit:

$$\frac{\mathbf{w}^{\star\top}\mathbf{w}^\star}{d} \xrightarrow[d\to\infty]{} Q^\star, \qquad \frac{\mathbf{w}^\top\mathbf{w}}{d} \xrightarrow[d\to\infty]{} Q \xrightarrow[\beta\to\infty]{} q, \tag{157}$$

as well as

$$\frac{\mathbf{w}^\top\mathbf{x}}{\sqrt{d}} = \frac{\mathbf{w}^\top \left( \left( \mathbb{I} + \frac{\mathbf{W}^\star F_1(\Lambda, Q^\star)\mathbf{W}^{\star\top}}{d} \right) \mathbf{z}_\mu + \frac{\mathbf{W}^\star F_2(\Lambda, Q^\star)}{\sqrt{d}} \right)}{\sqrt{d}} \xrightarrow[d\to\infty]{} h_0 + m(F_1(\Lambda, Q^\star)h_0^\star + F_2(\Lambda, Q^\star)) \tag{158}$$

$$\frac{\mathbf{w}^{\star\top}\mathbf{x}}{\sqrt{d}} \xrightarrow[d\to\infty]{} h_0^\star + Q^\star(F_1(\Lambda, Q^\star)h_0^\star + F_2(\Lambda, Q^\star))$$

with $h_0 \equiv \frac{\mathbf{w}^\top\mathbf{z}}{\sqrt{d}}$ and $h_0^\star \equiv \frac{\mathbf{w}^{\star\top}\mathbf{z}}{\sqrt{d}}$.

Therefore, the randomness in equation Equation (156) reduces to a low-dimensional integral over $P_{\text{latents}}$ and the random variables $(h_0^\star, h_0)$, which follow the distribution $\begin{pmatrix} h_0^\star \\ h_0 \end{pmatrix} \sim \mathcal{N}\left(0, \begin{pmatrix} \mathbb{I} & m^\top \\ m & q \end{pmatrix}\right)$.

In Appendix D we will specify the test loss form for the specific setting of interest in the main text, as well as the numerical procedure used to compute it.

Another metric of interest is the training loss $\epsilon_t$ (at the global minimizer of the loss). While it is not straightforward to express it as a function of the summary statistics $m, q, V$, it is in fact easily obtained from our analysis, as it's proportional to the intensive free entropy. This is quite natural, as we choose $P_{\text{likelihood}}$ such that for $\beta \to \infty$ it will describe the minima of the empirical risk:

$$\epsilon_t = -\lim_{\beta,d\to\infty} \frac{\mathbb{E}[\log \mathcal{Z}]}{\beta d}. \tag{159}$$

In Appendix D we will specify the free entropy form for the specific setting of interest in the main text, as well as the numerical procedure used to compute it.

## C. Derivation of results 3.1 and 3.2 from the main text

### C.1. Derivation of result 3.1

In order to obtain Result 3.1 from the main text, we follow the computation for the more generic model detailed in Appendix B. We consider the Bayes optimal free entropy $\Phi^{BO}$ in Equation (139) and the corresponding maximization conditions in Equation (145) and specialize them for the data model Equation (1) by setting $K = 2$, $\Lambda_\mu = (\lambda_\mu, \nu_\mu)$, $\mathbf{w}^\star$ standard Gaussian in each component, and

$$F_1(\lambda, \nu) = \begin{pmatrix} 0 & 0 \\ 0 & -\frac{\mathbb{E}[\nu^2]}{1+\mathbb{E}[\nu^2]+\sqrt{1+\mathbb{E}[\nu^2]}} \end{pmatrix} \quad \text{and} \quad F_2(\lambda, \nu) = \begin{pmatrix} \lambda \\ \frac{\nu}{\sqrt{1+\mathbb{E}[\nu^2]}} \end{pmatrix}. \tag{160}$$

Additionally, we remark that the order parameters will be $q, \hat{q} \in \mathbb{R}^{2\times2}$ (see Appendix B.1 for the notations).

Starting from equations Equation (139), Equation (142) and Equation (144), the free entropy becomes

$$\Phi^{\mathrm{BO}} = \arg\max_{q} \phi^{\mathrm{BO}}(q)$$

$$= \max_{q,\hat{q}} \left\{ -\frac{1}{2}\operatorname{Tr}\left(q\hat{q}^{\top}\right) + \frac{1}{2}\operatorname{Tr}(\mathbb{I}+\hat{q}) - \frac{1}{2}\log\det(\mathbb{I}+\hat{q}) + \alpha\,\mathbb{E}_{\xi}[Z_{\mathrm{out}}^{\star}(\sqrt{q}\,\xi,\,\mathbb{I}-q,\mathbb{I})\,\log Z_{\mathrm{out}}^{\star}(\sqrt{q}\,\xi,\,\mathbb{I}-q,\mathbb{I})] \right\}. \tag{161}$$

where we recall the definition $\mathcal{Z}_{\mathrm{out}}^{\star}(\omega,V,Q) = \mathbb{E}_{h\sim\mathcal{N}(\omega,V)}[P_{\mathrm{out}}^{\star}(h,Q)]$ from Equation (143). We specify our data model Equation (1) following Equation (95) and Equation (98) to get its component $P_{\mathrm{out}}^{\star}(h,Q=\mathbb{I})$:

$$P_{\mathrm{out}}^{\star}(h,\mathbb{I}) = \sqrt{1+\mathbb{E}[\nu^2]}\,\mathbb{E}_{\lambda,\nu}\left[e^{\lambda h_1 + \sqrt{1+\mathbb{E}[\nu^2]}\nu h_2 - \frac{1}{2}(\lambda^2+\nu^2) - \frac{1}{2}\mathbb{E}[\nu^2]h_2^2}\right]. \tag{162}$$

The extremisation in Eq. 161 is performed over the order parameters $q,\hat{q}$. In particular, we can now define

$$\phi^{\mathrm{BO}}(q) = \max_{\hat{q}} \left\{ -\frac{1}{2}\operatorname{Tr}\left(q\hat{q}^{\top}\right) + \frac{1}{2}\operatorname{Tr}(\mathbb{I}+\hat{q}) - \frac{1}{2}\log\det(\mathbb{I}+\hat{q}) + \alpha\,\mathbb{E}_{\xi}[Z_{\mathrm{out}}^{\star}(\sqrt{q}\,\xi,\,\mathbb{I}-q,\mathbb{I})\,\log Z_{\mathrm{out}}^{\star}(\sqrt{q}\,\xi,\,\mathbb{I}-q,\mathbb{I})] \right\}. \tag{163}$$

to fully characterize Result 3.1.

**Saddle point equations**

To find the solution $q^{BO} = \mathrm{argmax}_{q}\,\phi^{\mathrm{BO}}(q)$ we extremise $\phi^{\mathrm{BO}}$ from which we get the following system of equations:

$$\begin{cases} q = \hat{q}(\mathbb{I}+\hat{q})^{-1}, \\ \hat{q} = \alpha\mathbb{E}_{\xi}\left[Z_{\mathrm{out}}^{\star}(\sqrt{q}\xi,\mathbb{I}-q,\mathbb{I})f_{\mathrm{out}}^{\star}(\sqrt{q}\xi,\mathbb{I}-q,\mathbb{I})^{\otimes 2}\right]. \end{cases} \tag{164}$$

We often will represent this system of equations with $q = F^{BO}(q)$.

## C.2. Derivation of Result 3.2

In this section, we derive Result 3.2 starting from stating the Approximate Message Passing (AMP) algorithm and linearising its corresponding state evolution equations.

### C.2.1. APPROXIMATE MESSAGE PASSING (AMP)

First, we state the particular Approximate Message Passing algorithm we used for the experiments shown in Figure 1, algorithm 1. This particular AMP algorithm was derived from relaxed-BP, following the lines of (Aubin et al., 2018). Note that we have introduced $f_Q^{\star}$, another function derived from $P_{\mathrm{out}}^{\star}$ as in Equation (165):

$$f_Q^{\star}(\omega,V,Q) \equiv \partial_Q \log Z_{\mathrm{out}}^{\star}(\omega,V,Q) = \frac{1}{Z_{\mathrm{out}}^{\star}(\omega,V,Q)}\int dz\,\mathcal{N}(z;\omega,V)\partial_Q P_{\mathrm{out}}^{\star}(z,Q). \tag{165}$$

From Figure 1 it can be seen that the performance of AMP, at least for the $P_{\mathrm{latents}}$ used in those experiments, perfectly matches the Bayes optimal estimator. In fact, it is known that the iterates of AMP follow so-called *state evolution* equations (Berthier et al., 2020),

$$q_{t+1}^{AMP} = F^{BO}(q_t^{AMP}) \tag{166}$$

where $F^{BO}$ is exactly the same as in equation Equation (164).

### C.2.2. LINEAR STABILITY OF AMP AND WEAK RECOVERY

We are interested in computing for which values of $\alpha$ the AMP iterate started from random initialization obtains positive overlap with the hidden spikes, i.e. $q_{11} > 0$ for $\mathbf{u}^{\star}$ and $q_{22} > 0$ for $\mathbf{v}^{\star}$. In order to do this, we linearize state evolution, Equation (166) around its fixed point $q = \mathbb{O}$ (zero matrix), and look for which value of $\alpha$ the fixed point is linearly unstable. Note that $q = \mathbb{O}$ is always a fixed point of Equation (166), since

$$Z_{\mathrm{out}}^{\star}(0,\mathbb{I},\mathbb{I}) = 1, \qquad f_{\mathrm{out}}^{\star}(0,\mathbb{I},\mathbb{I}) = 0. \tag{167}$$

---

**Algorithm 1** Spiked Gaussian Model AMP

---

1: **Input:** Data $\mathbf{X} \in \mathbb{R}^{n \times d}$.
2: **Initialize:** $\widehat{\mathbf{W}}^{t=0} \in \mathbb{R}^{K \times d}$, $\widehat{\mathbf{C}}_i^{t=0} \in \mathcal{S}_K^+$ for $i \in [d]$, $\mathbf{f}^{t=0} \in \mathbb{R}^{n \times K}$.
3: **for** $t \leq T$ **do**
4:     */* Update likelihood mean and variance */*
5:     For each $\mu \in [n]$,
6:        $\mathbf{V}_\mu^t = \frac{1}{d} \sum_{i=1}^d X_{\mu i}^2 \widehat{\mathbf{C}}_i^t \in \mathbb{R}^{K \times K}$,       $\boldsymbol{\omega}_\mu^t = \frac{1}{\sqrt{d}} \sum_{i=1}^d X_{\mu i} \widehat{\mathbf{W}}_i^t - \mathbf{V}_\mu^t \mathbf{f}_\mu^{t-1} \in \mathbb{R}^K$.
7:        $\mathbf{Q}^t = \frac{1}{d} \sum_{i=1}^d \left( \widehat{\mathbf{C}}_i^t + \widehat{\mathbf{W}}_i^t \widehat{\mathbf{W}}_i^{t\top} \right) \in \mathbb{R}^{K \times K}$.
8:     For each $\mu \in [n]$,
9:        $\mathbf{f}_\mu^t = f_{\text{out}}^\star \left( \boldsymbol{\omega}_\mu^t, \mathbf{V}_\mu^t, \mathbf{Q}^t \right) \in \mathbb{R}^K$,      $\partial \mathbf{f}_\mu^t = \partial_{\boldsymbol{\omega}} f_{\text{out}}^\star \left( \boldsymbol{\omega}_\mu^t, \mathbf{V}_\mu^t, \mathbf{Q}^t \right) \in \mathbb{R}^{K \times K}$.
10:       $\mathbf{f}_{Q\mu}^t = f_Q^\star \left( \boldsymbol{\omega}_\mu^t, \mathbf{V}_\mu^t, \mathbf{Q}^t \right) \in \mathbb{R}^{K \times K}$.
11:     */* Update prior first and second moments */*
12:     For each $i \in [d]$,
13:        $\mathbf{A}_i^t = -\frac{1}{d} \sum_{\mu=1}^n X_{\mu i}^2 \, \partial \mathbf{f}_\mu^t - \frac{2}{d} \sum_{\mu=1}^n \mathbf{f}_{Q\mu}^t \in \mathbb{R}^{K \times K}$.
14:        $\mathbf{b}_i^t = \frac{1}{\sqrt{d}} \sum_{\mu \in [n]} X_{\mu i} \mathbf{f}_\mu^t - \frac{1}{d} \sum_{\mu=1}^n X_{\mu i}^2 \, \partial \mathbf{f}_\mu^t \, \widehat{\mathbf{W}}_i^t \in \mathbb{R}^K$.
15:        $\widehat{\mathbf{W}}_i^{t+1} = \left( \mathbf{I}_s + \mathbf{A}_i^t \right)^{-1} \mathbf{b}_i^t \in \mathbb{R}^K$,      $\widehat{\mathbf{C}}_i^{t+1} = \left( \mathbf{I}_s + \mathbf{A}_i^t \right)^{-1} \in \mathbb{R}^{K \times K}$.
16: **end for**
17: **Return:** estimators $\widehat{\mathbf{W}}_{\text{amp}} \in \mathbb{R}^{d \times K}$, $\widehat{\mathbf{C}}_{\text{amp}} \in \mathbb{R}^{d \times K \times K}$.

---

Note that by definition, $q \in \mathbb{S}_+^2$, the cone of positive semi-definite matrices. We linearize around the fixed point $q = \mathbb{O}$, with a perturbation $\delta q \in \mathbb{S}_+^2$:

$$F^{BO}(q) \approx \alpha \mathcal{F}(\delta q) + \mathcal{O}(\|\delta q\|_F^2) \tag{168}$$

with

$$\mathcal{F}(q) = \partial_\omega f_{\text{out}}^\star(\omega = 0, V = \mathbb{I}, Q = \mathbb{I}) q \partial_\omega f_{\text{out}}^\star(\omega = 0, V = \mathbb{I}, Q = \mathbb{I})^\top \tag{169}$$

and we look at which value of alpha $\alpha_c$ the perturbation starts growing during the iteration, i.e.

$$\frac{1}{\alpha_c} = \sup_{q \in \mathbb{S}_+^2, \|q\|_F = 1} \|\mathcal{F}(q)\|_F. \tag{170}$$

Let $\lambda_+$ be the largest eigenvalue of $\partial_\omega f_{\text{out}}(0, \mathbb{I})$ and $v_+$ be the corresponding eigenvector, then

$$\frac{1}{\alpha_c} = \sup_{q \in \mathbb{S}_+^2, \|q\|_F = 1} \|\mathcal{F}(q)\|_F = \lambda_+^2 \tag{171}$$

and the perturbation along which the fixed point becomes unstable is in direction of $v_+^{\otimes 2}$. Indeed, $\partial_\omega f_{\text{out}}^\star$ is symmetric therefore diagonalizable, so let $\partial_\omega f_{\text{out}}^\star = Q \Lambda Q^\top$ with $\Lambda = \begin{pmatrix} \lambda_1 & 0 \\ 0 & \lambda_2 \end{pmatrix}$ ordered such that $|\lambda_1| > |\lambda_2|$. The Frobenius norm is invariant under orthogonal change of basis, so we have:

$$\sup_{q \in \mathbb{S}_+^2, \|q\|_F = 1} \|\mathcal{F}(q)\|_F = \sup_{q' \in \mathbb{S}_+^2, \|q'\|_F = 1} \|\Lambda q' \Lambda\|_F \tag{172}$$

with $q' \equiv Q^\top q Q$. Since $\|\Lambda q' \Lambda\|_F^2 = \lambda_1^4 q_{11}'^2 + 2\lambda_1^2 \lambda_2^2 q_{12}'^2 + \lambda_2^4 q_{22}'^2$ and $|\lambda_1| > |\lambda_2|$, this is maximized by $q' = \begin{pmatrix} 1 & 0 \\ 0 & 0 \end{pmatrix}$ and

$$\sup_{q \in \mathbb{S}_+^2, \|q\|_F = 1} \|\mathcal{F}(q)\|_F = \lambda_+^2 \tag{173}$$

where the supremum is reached for a perturbation in the direction

$$q_{\text{pert}} = v_+^{\otimes 2} = Q \begin{pmatrix} 1 & 0 \\ 0 & 0 \end{pmatrix} Q^\top. \tag{174}$$

**Weak recovery of $\mathbf{u}^\star$.** Now, using the explicit form

$$\partial_\omega f_{\text{out}}^\star(0, \mathbb{I}, \mathbb{I}) = \begin{pmatrix} \mathbb{E}[\lambda^2] & \sqrt{\frac{1}{1+\mathbb{E}[\nu^2]}}\mathbb{E}[\lambda\nu] \\ \sqrt{\frac{1}{1+\mathbb{E}[\nu^2]}}\mathbb{E}[\lambda\nu] & 0 \end{pmatrix} \tag{175}$$

we have, letting $\gamma \equiv \frac{\mathbb{E}[\lambda\nu]^2}{\mathbb{E}[\lambda^2]^2(1+\mathbb{E}[\nu^2])}$, that the eigenvalues of $\partial_\omega f_{\text{out}}^\star$ are

$$\lambda_\pm = \frac{\mathbb{E}[\lambda^2](1 \pm \sqrt{1+4\gamma})}{2} \tag{176}$$

with

$$v_+ \propto \begin{pmatrix} 1 \\ \frac{2\sqrt{\gamma}}{1+\sqrt{1+4\gamma}} \end{pmatrix} \quad \text{such that} \|v_+\| = 1. \tag{177}$$

This gives

$$\alpha_c = \frac{1}{\mathbb{E}[\lambda^2]^2} f(\mathbb{E}[\lambda^2], \mathbb{E}[\nu^2], \mathbb{E}[\lambda\nu]) \tag{178}$$

with

$$f(\mathbb{E}[\lambda^2], \mathbb{E}[\nu^2], \mathbb{E}[\lambda\nu]) = \frac{4}{\left(1 + \sqrt{1 + \frac{4\mathbb{E}[\lambda\nu]^2}{\mathbb{E}[\lambda^2]^2(1+\mathbb{E}[\nu^2])}}\right)^2} \tag{179}$$

and

$$q_{\text{pert}} = v_+^{\otimes 2} \propto \begin{pmatrix} 1 & \frac{2\sqrt{\gamma}}{1+\sqrt{1+4\gamma}} \\ \frac{2\sqrt{\gamma}}{1+\sqrt{1+4\gamma}} & \frac{2\gamma}{1+2\gamma+\sqrt{1+4\gamma}} \end{pmatrix}. \tag{180}$$

This shows that for $\alpha > \alpha_c$, the AMP iteration started at initialization will develop a linear instability in the $u$ direction (direction of $q_{11}$ positive), and hence achieve weak recovery of $\mathbf{u}^\star$.

**Weak recovery of $\mathbf{v}^\star$.** In the case of $\mathbb{E}[\lambda\nu] \neq 0$, we have $\gamma > 0$ and we see from Equation (180) that at $\alpha = \alpha_c$ the AMP iteration started at initialization will develop a linear instability also in the $v$ direction (direction of $q_{22} \neq 0$), and hence achieve weak recovery of $\mathbf{v}^\star$.

The case of $\mathbb{E}[\lambda\nu] = 0$ is trickier. There, the linear instability around initialization only magnetizes against $\mathbf{u}^\star$ (i.e. in the direction of $q_{11} > 0$). To assess weak recovery of $\mathbf{v}^\star$, we then study how the iteration proceeds from an overlap of the form

$$\begin{pmatrix} q_u & 0 \\ 0 & 0 \end{pmatrix} \tag{181}$$

with $q_u$ small but positive. At the next iteration, we will have that the overlaps evolve into

$$F^{BO}\left(\begin{pmatrix} q_u & 0 \\ 0 & 0 \end{pmatrix}\right) \tag{182}$$

and we ask whether the $(2,2)$ component (that we call from now on $q_v$) of the resulting overlap remains zero or not.

*Claim*: after one additional AMP step, letting $k^\star$ be the correlation exponent of $P_{\text{latents}}$ (recall Def. Definition 1.1), the leading order term of the expansion of $q_v$ as a function of $q_u$ is of the form

$$q_v \equiv F^{BO}\left(\begin{pmatrix} q_u & 0 \\ 0 & 0 \end{pmatrix}\right)_{22} = \frac{\alpha}{1+\mathbb{E}[\nu^2]} \frac{q_u^{k^\star}\mathbb{E}[\nu\lambda^{k^\star}]^2}{k^\star!} + \mathcal{O}(q_u^{k^\star+1}). \tag{183}$$

Then, any small magnetization along $\mathbf{u}^\star$ will lead to a small magnetization along $\mathbf{v}^\star$, i.e. weak recovery of $\mathbf{u}^\star$ implies weak recovery of $\mathbf{v}^\star$. We thus show that if $k^\star$ is finite, AMP achieves non-zero cosine similarity with both spikes for all $\alpha > \alpha_c$, thus showing result Result 3.2.

*Proof*:

We are left to prove Equation (183) using the state evolution equations Equation (164).

We start with the equation for $q$, letting

$$\hat{q} = \begin{pmatrix} \hat{q}_u & \hat{q}_{uv} \\ \hat{q}_{uv} & \hat{q}_v \end{pmatrix}, \qquad q_v \equiv F^{BO}\left(\begin{pmatrix} q_u & 0 \\ 0 & 0 \end{pmatrix}\right). \tag{184}$$

We have from Equation (164) that

$$q = \hat{q}(\mathbb{I} + \hat{q})^{-1} \tag{185}$$

which implies

$$q_v = \frac{\hat{q}_v(1 + \hat{q}_v) - \hat{q}_{uv}^2}{(1 + \hat{q}_v)(1 + \hat{q}_u)} \tag{186}$$

evaluated at the point $\begin{pmatrix} q_u & 0 \\ 0 & 0 \end{pmatrix}$. After much work of simplification, one can reach the following expressions for $\hat{q}_u, \hat{q}_v, \hat{q}_{uv}$:

$$\hat{q}_u = \alpha \frac{\mathbb{E}[\lambda^2]^2 q_u}{(1 + \mathbb{E}[\lambda^2]q_u)^2},$$

$$\hat{q}_{uv} = \alpha \sqrt{\frac{1}{1 + \mathbb{E}[\nu^2]}} \frac{\mathbb{E}[\lambda^2]q_u(1 + \mathbb{E}[\lambda^2]q_u)}{(1 + 2\mathbb{E}[\lambda^2]q_u)^{3/2}} \mathbb{E}\left[\lambda \nu \exp\left(-\frac{1}{2}\frac{q_u^2\lambda^2}{1 + 2\mathbb{E}[\lambda^2]q_u}\right)\right], \tag{187}$$

$$\hat{q}_v = \frac{\alpha}{1 + \mathbb{E}[\nu^2]}(1 + \mathbb{E}[\lambda^2]q_u)\,\mathbb{E}_\xi\left[e^{-\frac{\mathbb{E}[\lambda^2]q_u\xi^2}{1 + \mathbb{E}[\lambda^2]q_u}}\left(\mathbb{E}_{\lambda,\nu}\left[\nu e^{-\frac{q_u\lambda^2}{2} + \sqrt{q_u}\,\xi\lambda}\right]\right)^2\right].$$

From Equation (187), we see that for all three elements of $\hat{q}$, the leading order term is at least $\mathcal{O}(q_u)$.

From Equation (186) we will be able conclude that the leading order term of $q_v$ will be the leading order term of $\hat{q}_v$, in two steps: first, we show that if $k^\star$ is the correlation exponent, then the leading order term of $\hat{q}_v = a_{k^\star}\mathbb{E}[\lambda^{k^\star}\nu]^2 q_u^{k^\star} + \mathcal{O}(q_u^{k^\star+1})$, i.e., that whichever moment is the first nonzero one, $\hat{q}_v$ contains it with a nonzero coefficient $a_{k^\star}$. With that, it is clear that the denominator will only contribute with terms of order at least $\mathcal{O}(q_u^{k^\star+1})$, $\hat{q}_v^2 = \mathcal{O}(q_u^{2k^\star})$ and $\hat{q}_{uv}^2 = \mathcal{O}(q_u^{2k^\star})$, from which we get that also $q_v = a_{k^\star}\mathbb{E}[\lambda^{k^\star}\nu]^2 q_u^{k^\star} + \mathcal{O}(q_u^{k^\star+1})$.

To conclude, we remain to show that if $k^\star$ is the correlation exponent, then the leading order term of $\hat{q}_v = a_{k^\star}\mathbb{E}[\lambda^{k^\star}\nu]^2 q_u^{k^\star} + \mathcal{O}(q_u^{k^\star+1})$. Note that we can write

$$e^{-\frac{q_u\lambda^2}{2} + \sqrt{q_u}\,\xi\lambda} = \sum_{n=0}^{\infty} H_n(\xi)\frac{q_u^{n/2}\lambda^n}{n!}, \tag{188}$$

and then

$$\hat{q}_v = \frac{\alpha(1 + \mathbb{E}[\lambda^2]q_u)}{1 + \mathbb{E}[\nu^2]}\mathbb{E}_\xi\left[e^{-\frac{\mathbb{E}[\lambda^2]q_u\xi^2}{1 + \mathbb{E}[\lambda^2]q_u}}\left(\sum_{n=1}^{\infty} H_n(\xi)\frac{q_u^{n/2}\mathbb{E}[\nu\lambda^n]}{n!}\right)^2\right]. \tag{189}$$

Using our assumption that $\mathbb{E}[\nu\lambda^{k^\star}]$ is the first nonzero moment of the type $\mathbb{E}[\nu\lambda^n]$, we get that the leading order term of the above is

$$\hat{q}_v = \frac{\alpha}{1+\mathbb{E}[\nu^2]}\mathbb{E}_\xi\left[\left(H_{k^\star}(\xi)\frac{q_u^{k^\star/2}\mathbb{E}[\nu\lambda^{k^\star}]}{k^\star!}\right)^2\right] + \mathcal{O}(q_u^{k^\star+1}) = \frac{\alpha}{1+\mathbb{E}[\nu^2]}\frac{q_u^{k^\star}\mathbb{E}[\nu\lambda^{k^\star}]^2}{k^\star!} + \mathcal{O}(q_u^{k^\star+1}) \tag{190}$$

and clearly $\frac{\alpha}{1+\mathbb{E}[\nu^2]}\frac{1}{k^\star!} > 0$ so that we prove Equation (183).

**The case of $k^\star = \infty$ and the independent latents example**

What happens when $k^\star = \infty$? One such case we studied was the *independent* latents setting, in which $\lambda \perp\!\!\!\perp \nu$. In this setting, it can be shown that

$$q = \begin{pmatrix} q_u(\alpha) & 0 \\ 0 & 0 \end{pmatrix} \tag{191}$$

is a superstable fixed point of AMP's state evolution for all $\alpha$. This suggests that efficient weak recovery of $\mathbf{v}^\star$ requires $n = \mathcal{O}(d^2)$ samples, in accordance to the results previously found in (Bardone & Goldt, 2024).

Nevertheless, there are multiple other choices for $P_{\text{latents}}$ with $k^\star = \infty$. For example, taking $\lambda \sim \mathcal{N}(0,1)$ and $\nu = |\lambda|\xi$ with $\xi \sim \frac{1}{2}\delta_{1,\xi} + \frac{1}{2}\delta_{\xi,-1}$. Characterizing such settings is an interesting direction left for future work.

**Fixed point iteration to solve Equation (164)**

We solve the system of equations Equation (164) numerically via fixed-point iteration. We initialize the order parameter $q$ to

$$q^{(0)} = \begin{pmatrix} 0.001 & 0.0001 \\ 0.0001 & 0.001 \end{pmatrix} \tag{192}$$

to represent an uninformative initialization.

At each iteration, we first estimate the conjugate variable $\hat{q}$ using the last equation of Equation (164). For the $P_{\text{latents}}$ setting we considered (see Figures 1 and 2), the function $f_{\text{out}}^\star$ can be computed analytically in terms of Gaussian tail functions. However, the outer integral over $\xi$ cannot be solved analytically, and we used Monte Carlo integration to estimate it with $N_{\text{samples}} \approx 10^6$. Using this estimate, we compute $q_{\text{new}}$ using the first equation of Equation (164).

To ensure convergence, we apply damping to the updates. Let $q^{(t)}$ denote the state at iteration $t$. The update rule is given by:

$$q^{(t+1)} = \epsilon q^{(t)} + (1-\epsilon)q_{\text{new}}. \tag{193}$$

We use a damping factor of $\epsilon = 0.6$ and perform $\approx 200$ iterations, which was the typical time to convergence. The solver is implemented in Python using NumPy (Harris et al., 2020) and parallelized over multiple CPUs using `mpi4py` (Dalcin et al., 2011).

**Explicit form of $P_{\text{out}}^\star$, $f_{\text{out}}^\star$ for the $P_{\text{latents}}$ setting considered in Figures 1 and 2**

Simplifying further equation Equation (162) one gets

$$P_{\text{out}}^\star(h,\mathbb{I}) = \sqrt{1+\mathbb{E}[\nu^2]}e^{-\frac{1}{2}\mathbb{E}[\nu^2](1+h_2^2)}\mathbb{E}_{\lambda,v}\left[e^{-\frac{1}{2}\lambda^2}e^{\lambda h_1}e^{\sqrt{1+\mathbb{E}[\nu^2]}\nu h_2}\right]. \tag{194}$$

Additionally, we simplify $f_{\text{out}}^\star(\omega,V,\mathbb{I})$ to get

$$f_{\text{out}}^\star(\omega,V,\mathbb{I}) = (\mathbb{I}+\Gamma V)^{-1}\left(\frac{\mathbb{E}_{\nu,\lambda}\left[\exp(-\frac{1}{2}\lambda^2)\exp\left(\frac{1}{2}\beta^\top(V^{-1}+\Gamma)^{-1}\beta + w^\top V^{-1}(V^{-1}+\Gamma)^{-1}\beta\right)\beta\right]}{\mathbb{E}_{\nu,\lambda}\left[\exp(-\frac{1}{2}\lambda^2)\exp\left(\frac{1}{2}\beta^\top(V^{-1}+\Gamma)^{-1}\beta + w^\top V^{-1}(V^{-1}+\Gamma)^{-1}\beta\right)\right]} - \Gamma\omega\right) \tag{195}$$

with $\Gamma \equiv \begin{pmatrix} 0 & 0 \\ 0 & \mathbb{E}[\nu^2] \end{pmatrix}$ and $\beta \equiv \begin{pmatrix} \lambda & \sqrt{1+\mathbb{E}[\nu^2]}\nu \end{pmatrix}$.

The necessary expectations over $P_{\text{latents}}$ are given by, letting $t \equiv \Phi^{-1}(0.75)$ and $Q(x) \equiv \int_x^\infty \frac{1}{\sqrt{2\pi}} e^{-t^2/2}\, dt$:

$$
\mathbb{E}_{\lambda,\nu}\left[e^{\left(-\frac{a-1}{2}\lambda^2 + b\lambda + c\nu + d\lambda\nu\right)}\right] = \frac{1}{\sqrt{a}}\left[e^{-c}e^{\frac{(d-b)^2}{2a}}\left(\frac{1}{2}Q\left(\sqrt{\frac{a}{2}}\left(-t-\frac{b-d}{a}\right)\right) - \frac{1}{2}Q\left(\sqrt{\frac{a}{2}}\left(t-\frac{b-d}{a}\right)\right)\right)\right.
$$
$$
\left. + e^c e^{\frac{(b+d)^2}{2a}}\left(\frac{1}{2}Q\left(\sqrt{\frac{a}{2}}\left(t+\frac{b+d}{a}\right)\right) + \frac{1}{2}Q\left(\sqrt{\frac{a}{2}}\left(t-\frac{b+d}{a}\right)\right)\right)\right], \tag{196}
$$

$$
\mathbb{E}_{\lambda,\nu}\left[\lambda e^{\left(-\frac{a-1}{2}\lambda^2 + b\lambda + c\nu + d\lambda\nu\right)}\right] = \frac{1}{\sqrt{a}}\left[-e^c e^{\frac{(b+d)^2}{2a}}\left(-\frac{b+d}{a}\frac{1}{2}Q\left(\sqrt{\frac{a}{2}}\left(t+\frac{b+d}{a}\right)\right) + \frac{1}{\sqrt{2\pi a}}e^{-\frac{a}{2}\left(t+\frac{b+d}{a}\right)^2}\right)\right.
$$
$$
+ e^{-c}e^{\frac{(d-b)^2}{2a}}\left[\frac{b-d}{a}\left(\frac{1}{2}Q\left(\sqrt{\frac{a}{2}}\left(-t-\frac{b-d}{a}\right)\right) - \frac{1}{2}Q\left(\sqrt{\frac{a}{2}}\left(t-\frac{b-d}{a}\right)\right)\right)\right.
$$
$$
\left. + \frac{1}{\sqrt{2\pi a}}\left(e^{-\frac{a}{2}\left(-t-\frac{b-d}{a}\right)^2} - e^{-\frac{a}{2}\left(t-\frac{b-d}{a}\right)^2}\right)\right]
$$
$$
\left. + e^c e^{\frac{(b+d)^2}{2a}}\left(\frac{b+d}{a}\frac{1}{2}Q\left(\sqrt{\frac{a}{2}}\left(t-\frac{b+d}{a}\right)\right) + \frac{1}{\sqrt{2\pi a}}e^{-\frac{a}{2}\left(t-\frac{b+d}{a}\right)^2}\right)\right], \tag{197}
$$

$$
\mathbb{E}_{\lambda,\nu}\left[\nu e^{\left(-\frac{a-1}{2}\lambda^2 + b\lambda + c\nu + d\lambda\nu\right)}\right] = \frac{1}{\sqrt{a}}\left[-e^{-c}e^{\frac{(d-b)^2}{2a}}\left(\frac{1}{2}Q\left(\sqrt{\frac{a}{2}}\left(-t-\frac{b-d}{a}\right)\right) - \frac{1}{2}Q\left(\sqrt{\frac{a}{2}}\left(t-\frac{b-d}{a}\right)\right)\right)\right.
$$
$$
\left. + e^c e^{\frac{(b+d)^2}{2a}}\left(\frac{1}{2}Q\left(\sqrt{\frac{a}{2}}\left(t+\frac{b+d}{a}\right)\right) + \frac{1}{2}Q\left(\sqrt{\frac{a}{2}}\left(t-\frac{b+d}{a}\right)\right)\right)\right]. \tag{198}
$$

# D. Derivation of Result 5.1: ERM for the auto-encoder

### D.1. Derivation of Result 5.1

In order to obtain Result 5.1 from the main text, we use the result of the more generic computation detailed in Appendix B in order to study the empirical risk Equation (6). We consider the Empirical Risk Minimisation (ERM) free entropy $\Phi^{ERM}$ starting from Equation (139), Equation (149) and Equation (154), as well as the corresponding maximization conditions in Equation (154) and specialise them for the data model Equation (1) by setting $p = 1$, $K = 2$, $\Lambda_\mu = (\lambda_\mu, \nu_\mu)$, $\mathbf{w}^\star$ standard Gaussian in each component,

$$
F_1(\lambda, \nu) = \begin{pmatrix} 0 & 0 \\ 0 & -\frac{\mathbb{E}[\nu^2]}{1+\mathbb{E}[\nu^2]+\sqrt{1+\mathbb{E}[\nu^2]}} \end{pmatrix} \quad \text{and} \quad F_2(\lambda, \nu) = \begin{pmatrix} \lambda \\ \frac{\nu}{\sqrt{1+\mathbb{E}[\nu^2]}} \end{pmatrix}. \tag{199}
$$

Additionally, we must specialize the equations for the empirical risk Equation (6), by specifying the loss

$$
\ell(h, q) = -h\sigma(h) + \frac{1}{2}q\sigma(h)^2. \tag{200}
$$

Comparing Equation (200) with the empirical loss we aim to analyze (Equation (6)), we have ignored the constant $\|\mathbf{x}\|^2$ that does not affect the minimizer. In addition, $\mathbb{E}[\|\mathbf{x}\|^2] = d + \mathbb{E}[\lambda^2] = \mathcal{O}(d)$, which dominates the rest of the terms in the loss. To compute results such as in Fig. 2, we ignore this constant term by computing differences with respect to a baseline.

In addition, we will keep for generality the $l^2$ regularization $r(w) = \tilde{\lambda}\frac{w^2}{2}$ that was considered in the generic derivation in Appendix B, but we note that all results in the main text are obtained setting $\tilde{\lambda} = 0$.

Finally, we remark that the order parameters will be $q, \hat{q}, V, \hat{V} \in \mathbb{R}$ and $m, \hat{m} \in \mathbb{R}^{1\times 2}$ (see Appendix B.1 for the notations).

**Free entropy**

Putting everything together, from Equation (139),Equation (149) and Equation (153), we find that the ERM free entropy reduces to

$$\Phi = \max_{m,q,V,\hat{m},\hat{q},\hat{V}} \left( -m\hat{m}^\top - \frac{1}{2}V\hat{q} + \frac{1}{2}q\hat{V} + \frac{1}{2}\frac{\hat{q}+\hat{m}\hat{m}^\top}{\tilde{\lambda}+\hat{V}} - \alpha\mathbb{E}_\xi \left[ Z^\star_{\text{out}}(m^\top q^{-1/2}\xi, \mathbb{I} - m^\top q^{-1}m, \mathbb{I})\mathcal{M}_\ell(\sqrt{q}\xi, V, q) \right] \right) \quad (201)$$

with, from Equation (152),

$$\mathcal{M}_\ell(\sqrt{q}\xi, V, q) = \inf_h \left( -h\sigma(h) + \frac{1}{2}q\sigma(h)^2 + \frac{1}{2}\frac{(h-\sqrt{q}\xi)^2}{V} \right). \quad (202)$$

Finally, we recall the definition $\mathcal{Z}^\star_{\text{out}}(\omega, V, Q) = \mathbb{E}_{h\sim\mathcal{N}(\omega, V)}[P^\star_{\text{out}}(h, Q)]$ from Equation (143). We specify our data model Equation (1) following Equation (95) and Equation (98) to get its component $P^\star_{\text{out}}(h, Q = \mathbb{I})$:

$$P^\star_{\text{out}}(h, \mathbb{I}) = \sqrt{1+\mathbb{E}[\nu^2]}\mathbb{E}_{\lambda,\nu} \left[ e^{\lambda h_1 + \sqrt{1+\mathbb{E}[\nu^2]}\nu h_2 - \frac{1}{2}(\lambda^2+\nu^2) - \frac{1}{2}\mathbb{E}[\nu^2]h_2^2} \right]. \quad (203)$$

The extremisation in Eq. 201 is performed over the order parameters $m, q, V, \hat{m}, \hat{q}, \hat{V}$. In particular, we can now define

$$\phi_{\text{ER}}(m_u, m_v, q)$$
$$= \max_{V,\hat{m},\hat{q},\hat{V}} \left( -m\hat{m}^\top + \frac{1}{2}(q\hat{V} - V\hat{q}) + \frac{1}{2}\frac{\hat{q}+\hat{m}\hat{m}^\top}{\tilde{\lambda}+\hat{V}} - \alpha\mathbb{E}_\xi \left[ Z^\star_{\text{out}}(m^\top q^{-1/2}\xi, \mathbb{I} - m^\top q^{-1}m, \mathbb{I})\mathcal{M}_\ell(\sqrt{q}\xi, V, q) \right] \right) \quad (204)$$

to fully characterize Result 5.1. We write $m = (m_u, m_v)$.

**Saddle point equations**

Starting from Equation (154), and specialising to our setting, we get the following saddle point equations:

$$\begin{cases} m = \dfrac{\hat{m}}{\hat{V}+\tilde{\lambda}}, \\[2mm] q = \dfrac{\hat{q}+\hat{m}\hat{m}^\top}{(\hat{V}+\tilde{\lambda})^2}, \\[2mm] V = \dfrac{1}{\hat{V}+\tilde{\lambda}}, \\[2mm] \hat{m} = \alpha\mathbb{E}_\xi \left[ Z^\star_{\text{out}}\left( m^\top q^{-1/2}\xi, \mathbb{I} - m^\top q^{-1}m \right) f_{\text{out}}(\sqrt{q}\xi, V, q) f^{\star\top}_{\text{out}}\left( m^\top q^{-1/2}\xi, \mathbb{I} - \dfrac{m^\top m}{q} \right) \right], \\[2mm] \hat{q} = \alpha\mathbb{E}_\xi \left[ Z^\star_{\text{out}}\left( m^\top q^{-1/2}\xi, \mathbb{I} - \dfrac{m^\top m}{q} \right) (f_{\text{out}}(\sqrt{q}\xi, V, q))^2 \right], \\[2mm] \hat{V} = \alpha\mathbb{E}_\xi \left[ Z^\star_{\text{out}}\left( m^\top q^{-1/2}\xi, \mathbb{I} - \dfrac{m^\top m}{q} \right) \left( \sigma(\text{prox}_h)^2 - f_{\text{out}}(y, \sqrt{q}\xi, V, q) q^{-1/2}\xi \right) \right] + \dfrac{\hat{m}m^\top}{q}. \end{cases} \quad (205)$$

Here, we recall the definitions of $f^\star_{\text{out}}(\omega, V, Q)$ and $f_{\text{out}}(\omega, V, q)$ from Equations (143) and (155). Finally, specifying our loss $l(h, q)$, we get from Eq. 151

$$\text{prox}_\ell(\omega, V, q) = \arg\min_h \left\{ -h\sigma(h) + \frac{1}{2}q\sigma(h)^2 + \frac{1}{2}\frac{(h-\omega)^2}{V} \right\}. \quad (206)$$

### D.2. Numerical solution of the saddle point equations

To numerically solve the saddle point equations Equation (205) efficiently , we derive an alternative version of the equations for the conjugate order parameters $\hat{m}, \hat{q}, \hat{V}$. Both methods were implemented numerically and provide matching results.

**Rewriting the fixed-point equation.** The alternative equations can be derived starting from (135) without doing the substitution (136). Taking the $s \to 0$ and $\beta \to \infty$ limit, the ERM free entropy (201) reads in that case

$$
\Phi = \max_{m,q,V,\hat{m},\hat{q},\hat{V}} \left( -m\hat{m}^\top - \frac{1}{2}V\hat{q} + \frac{1}{2}q\hat{V} + \frac{1}{2}\frac{\hat{q} + \hat{m}\hat{m}^\top}{\tilde{\lambda} + \hat{V}} - \alpha \underbrace{\mathbb{E}_\xi \mathbb{E}_{h^\star \sim \mathcal{N}(0,Q^\star)} \left[ P_{\text{out}}^\star(h^\star, Q^\star) \mathcal{M}_\ell(h^\star, \xi, q, V, m) \right]}_{-\Phi_{\text{output}}} \right)
\tag{207}
$$

with

$$
\mathcal{M}_\ell(h^\star, \xi, q, V, m) = \min_h \left( -h\sigma(h) + \frac{1}{2}q\sigma(h)^2 + \frac{1}{2V}\left[ h - V(\sqrt{-\tilde{q}_0}\xi - \tilde{m}_0 h^\star) \right]^2 \right).
\tag{208}
$$

$\tilde{q}_0$ and $\tilde{m}_0$ are defined in (133) as $\tilde{m}_0 = -V^{-1}m$ and $\tilde{q}_0 = -V^{-1}qV^{-1} + V^{-1}mm^\top V^{-1}$. Then, we plug-in the specific expression (203) for $P_{\text{out}}^\star$ to obtain an expectation over the latent variables $\Lambda$. Recall additionally that we set $Q^\star = \mathbb{I}$. Grouping the terms introduced by $P_{\text{out}}^\star$ with $h^\star$ to make Gaussians appear, the output term reads

$$
\Phi_{\text{output}} = \mathbb{E}_{\xi,\Lambda} \left[ \sqrt{1 + \mathbb{E}[\nu^2]} \int \frac{dh^\star}{2\pi} e^{-\frac{1}{2}(h_1^\star - \lambda)^2} e^{-\frac{1}{2}(\sqrt{1+\mathbb{E}[\nu^2]}h_2^\star - \nu)^2} \mathcal{M}_\ell(h^\star, \xi, q, V, m) \right].
\tag{209}
$$

As in the main text, we stay in the $p = 1, K = 2$ setting, and note with subscripts 1 and 2 the first and second components of the vector $h^\star$. We now proceed to the change of variables $h_{0,1}^\star = h_1^\star - \lambda, h_{0,2}^\star = \sqrt{1 + \mathbb{E}[\nu^2]}h_2^\star - \nu$. This transformation inverts the relation (158). The output term then reads

$$
\Phi_{\text{output}} = \mathbb{E}_{\xi,\Lambda,h_0^\star \sim \mathcal{N}(0,\mathbb{I})} \min_h \left( -h\sigma(h) + \frac{1}{2}q\sigma(h)^2 + \frac{1}{2V}\left[ h - V\sqrt{-\tilde{q}_0}\xi - m\left( \begin{smallmatrix} h_{0,1}^\star + \lambda \\ \frac{h_{0,2}^\star + \nu}{\sqrt{1+\mathbb{E}[\nu^2]}} \end{smallmatrix} \right) \right]^2 \right).
\tag{210}
$$

Written in terms of $F_1$ and $F_2$, the free entropy reads

$$
\Phi = \max \left( -m\hat{m}^\top - \frac{1}{2}V\hat{q} + \frac{1}{2}q\hat{V} + \frac{1}{2}\frac{\hat{q} + \hat{m}\hat{m}^\top}{\tilde{\lambda} + \hat{V}} - \alpha \mathbb{E}_{\Lambda,\xi} \mathbb{E}_{h_0^\star \sim \mathcal{N}(0,\mathbb{I})} \mathcal{M}_\ell(h_0^\star, \xi, q, V, m) \right)
\tag{211}
$$

with

$$
\mathcal{M}_\ell(h_0^\star, \xi, q, V, m, \Lambda) = \inf_h \left( \frac{1}{2}\left[ -2h\sigma(h) + q\sigma(h)^2 \right] + \frac{1}{2}V^{-1}\left[ h - V(\sqrt{-\tilde{q}_0}\xi - \tilde{m}_0 h_0^\star) - m(F_1 h_0^\star + F_2) \right]^2 \right).
\tag{212}
$$

While we derived (211) and (212) with $F_1$ and $F_2$ specified by (95), these expressions hold for generic $F_1$ and $F_2$.

We write

$$
Y = m(F_1 h_0^\star + F_2) - V\tilde{m}_0 h_0^\star + V\sqrt{-\tilde{q}_0}\xi = m\left( F_1 h_0^\star + F_2 + h_0^\star \right) + \sqrt{q - mm^T}\xi.
\tag{213}
$$

We note that

$$
Y \sim \mathcal{N}\left( \underbrace{m\left( F_1 h_0^\star + F_2 + h_0^\star \right)}_{=\mu_Y}, \underbrace{q - mm^\top}_{=\Sigma_Y} \right).
\tag{214}
$$

Thus, we can write the Moreau envelope as

$$
\mathcal{M}_\ell = \inf_h \left( -h\sigma(h) + \frac{q}{2}\sigma(h)^2 + \frac{1}{2}V^{-1}(h - Y)^2 \right).
\tag{215}
$$

To take the derivative, we use the envelope theorem, which states that for a generic function $f$

$$
\mathcal{M}(\theta) = \inf_x f(x, \theta) \Rightarrow \nabla_\theta \mathcal{M}(\theta) = \nabla_\theta f(x, \theta)|_{x = \text{prox}_x}
\tag{216}
$$

where $\text{prox}_x$ minimizes $f$.

The gradient of the energetic term with respect to $m$ is

$$
\hat{m} = \alpha \partial_m \Phi_{\text{output}} = -\alpha \mathbb{E}_{\lambda,\xi,h_0^\star} \left[ \partial_m \mathcal{M}_\ell \right].
\tag{217}
$$

Recalling that the loss term in the transformed variables does not depend on $m$, we obtain

$$\hat{m} = \alpha V^{-1} \mathbb{E}_\Lambda \mathbb{E}_{h_0^\star} \left[ \mathbb{E}_\xi [\text{prox}_\ell - Y] (F_1 h_0^\star + F_2 + h_0^\star)^\top - \mathbb{E}_\xi [(\text{prox}_\ell - Y)\xi](q - mm^\top)^{-1/2} m \right]. \tag{218}$$

The integral over $\xi$ can be solved explicitly for a few terms:

$$\hat{m} = \alpha V^{-1} \mathbb{E}_{\Lambda, h_0^\star} \left[ (\mathbb{E}_\xi [\text{prox}_\ell] - \mu_Y) (F_1 h_0^\star + F_2 + h_0^\star)^\top - \mathbb{E}_\xi [\text{prox}_\ell \xi] \Sigma_Y^{-1/2} m \right] + \alpha V^{-1} m, \tag{219}$$

where we recall that $\mu_Y$ and $\Sigma_Y$ are the mean and variance of $Y$.

For $\hat{q}$, we have

$$\hat{q} = 2\alpha \partial_V \Phi_{\text{output}} = -2\alpha \mathbb{E}_{\Lambda, h_0^\star, \xi} [\partial_V \mathcal{M}_\ell]. \tag{220}$$

$Y$ does not depend on $V$, and neither does the loss term. Thus, we simply have

$$\partial_V \mathcal{M}_\ell = -\frac{1}{2} V^{-2} (\text{prox}_\ell - Y)^2, \tag{221}$$

so that

$$\hat{q} = \alpha \mathbb{E}_{\Lambda, h_0^\star, \xi} \left[ V^{-2} (\text{prox}_\ell - Y) \right]. \tag{222}$$

Performing the integrals over $\xi$ that we can solve explicitly, we obtain

$$\hat{q} = \alpha V^{-2} \mathbb{E}_{\Lambda, h_0^\star} \left[ \mathbb{E}_\xi [(\text{prox}_\ell - \mu_Y)^2] - 2\mathbb{E}_\xi [\text{prox}_\ell \xi] \sqrt{\Sigma_Y} + \Sigma_Y \right]. \tag{223}$$

For $\hat{V}$, we have

$$\hat{V} = -2\alpha \partial_q \Phi_{\text{output}} = 2\alpha \mathbb{E}_{\Lambda, h_0^\star, \xi} [\partial_q \mathcal{M}_\ell]. \tag{224}$$

$q$ appears only in the loss and covariance $\Sigma_Y$. Thus,

$$\hat{V} = 2\alpha \mathbb{E}_{\Lambda, h_0^\star} \mathbb{E}_\xi [\frac{1}{2} \sigma(\text{prox}_\ell)^2 - \frac{1}{2} V^{-1} (\text{prox}_\ell - Y) \xi \Sigma_Y^{-1/2}]$$
$$= \alpha \mathbb{E}_{\Lambda, h_0^\star, \xi} \left[ \sigma(\text{prox}_\ell)^2 \right] - \alpha V^{-1} \mathbb{E}_{\Lambda, h_0^\star} \left( \mathbb{E}_\xi [\text{prox}_\ell \xi^\top] \Sigma_Y^{-1/2} \right) + \alpha V^{-1} \tag{225}$$

where in the last step we solved the Gaussian integral over $\xi$ for the $-Y \xi^\top$ term.

Our alternative derivation does not change the prior or trace term, so that the expressions for $m, q, V$ remain the same. Thus, the fixed-point equations that we solve numerically read

$$
\begin{cases}
m = \dfrac{\hat{m}}{\hat{V} + \tilde{\lambda}}, \\[2ex]
q = \dfrac{\hat{q} + \hat{m}\hat{m}^\top}{(\hat{V} + \tilde{\lambda})^2}, \\[2ex]
V = \dfrac{1}{\hat{V} + \tilde{\lambda}}, \\[2ex]
\hat{m} = \alpha V^{-1} \mathbb{E}_{\lambda, h_0^\star} \left[ (\mathbb{E}_\xi [\text{prox}_\ell] - \mu_Y)(F_1 h_0^\star + F_2 + h_0^\star)^\top - \mathbb{E}_\xi [\text{prox}_\ell \xi] \Sigma_Y^{-1/2} m \right] + \alpha V^{-1} m, \\[2ex]
\hat{q} = \alpha V^{-2} \mathbb{E}_{\lambda, h_0^\star} \left[ \mathbb{E}_\xi \left[ (\text{prox}_\ell - \mu_Y)^2 \right] - 2\mathbb{E}_\xi [\text{prox}_\ell \xi^\top] \sqrt{\Sigma_Y} + \Sigma_Y \right], \\[2ex]
\hat{V} = \alpha \mathbb{E}_{\lambda, h_0^\star, \xi} \left[ \sigma(\text{prox}_\ell)^2 \right] - \alpha V^{-1} \mathbb{E}_{\lambda, h_0^\star} \mathbb{E}_\xi [\text{prox}_\ell \xi] \Sigma_Y^{-1/2} + \alpha V^{-1}.
\end{cases} \tag{226}
$$

The values of $F_1$ and $F_2$ are given by Equation (95). The proximal term is given by

$$\text{prox}_\ell(h_0^\star, \xi, q, m, V, \lambda, \nu) = \underset{h}{\text{arginf}} \left( \frac{1}{2} [-2h\sigma(h) + \sigma(h)q\sigma(h)] + \frac{1}{2} (h - Y)^2 V^{-1} \right). \tag{227}$$

We recall that $\xi \sim \mathcal{N}(0,1)$, $h_0^\star \sim \mathcal{N}(0,\mathbb{I})$ and $\Lambda = (\lambda, \nu)$, with $\lambda \sim \mathcal{N}(0,1)$ for the choice $P_{\text{latents}}$ of Figures 1, 2. Additionally, $\nu$ is deterministically obtained from $\lambda$ for this distribution, and thus does not appear in the expectation. Finally, we recall that

$$Y = m\left(F_1 h_0^\star + F_2 + h_0^\star\right) + \sqrt{q - mm^T}\,\xi \tag{228}$$

where

$$\mu_Y = m\left(F_1 h_0^\star + F_2 + h_0^\star\right), \tag{229}$$

$$\Sigma_Y = q - mm^\top. \tag{230}$$

**Fixed-point iteration.** We solve the system of equations Equation (226) numerically via fixed-point iteration. We initialize the order parameters $m, q,$ and $V$ prior to the iteration. Specifically, $m$ is initialized to the small non-zero vector $(10^{-3}, 10^{-3})$ to represent an uninformative initialization. The initial values of $q$ and $V$ are chosen to ensure stability and depend on the specific activation function used. The exact initialization parameters for each simulation can be found in the Supplementary Material.

At each step, we first estimate the conjugate parameters $\hat{m}, \hat{q}, \hat{V}$ using the last three equations of Equation (226). These expectations are approximated via Monte Carlo integration with $N_{\text{samples}} = 10^6$. We draw $N_{\text{samples}}$ independent tuples of the random variables $(\lambda, h_0^\star, \xi)$ from their respective distributions, evaluate the integrands, and compute the empirical mean. Using these estimates, we compute the candidate updates $\theta_{\text{new}} = (m_{\text{new}}, q_{\text{new}}, V_{\text{new}})$ via the first three equations of Equation (226).

To ensure convergence, we apply damping to the updates. Let $\theta^{(t)} = (m, q, V)^{(t)}$ denote the state at iteration $t$. The update rule is given by:

$$\theta^{(t+1)} = \epsilon\theta^{(t)} + (1-\epsilon)\theta_{\text{new}}. \tag{231}$$

We use a damping factor of $\epsilon = 0.95$ and perform 5000 iterations. To mitigate finite-sample noise inherent to the Monte Carlo integration, the order parameters used to compute the losses are the average of the values over the final 200 iterations. The solver is implemented in Python using NumPy (Harris et al., 2020) and parallelized over multiple CPUs using mpi4py (Dalcin et al., 2011).

Crucially, evaluating the expectations requires computing the proximal operator $\text{prox}_\ell$. We detail the numerical handling of this step below.

**Solving the proximal operator.** In general, the minimization problem Equation (227) does not admit a closed-form solution. However, for linear and ReLU activation functions, exact analytical solutions can be derived by finding the stationary points of the objective. For the linear case, the solution is given by:

$$\text{prox}_\ell^{\text{linear}}(Y) = \frac{Y}{1 + V(q-2)}. \tag{232}$$

For the ReLU activation, the solution is piecewise:

$$\text{prox}_\ell^{\text{ReLU}}(Y) = \begin{cases} \frac{Y}{1+V(q-2)} & \text{if } \frac{Y}{1+V(q-2)} > 0, \\ Y & \text{otherwise.} \end{cases} \tag{233}$$

For generic activation functions, we compute the proximal operator numerically in two steps. First, we perform a grid search over an interval $h \in [-c, c]$ using $N_{\text{grid}} = 1000$ points, where the bound $c$ is selected heuristically based on the activation function. Second, we refine the estimate using golden-section search within a window $[h_0^\star - \Delta, h_0^\star + \Delta]$ around the optimal grid point $h_0^\star$ for 25 iterations, where $\Delta = 2c/(N_{\text{grid}} - 1)$. The solver is implemented in JAX (Bradbury et al., 2018) to take advantage of parallelization on an NVIDIA RTX A6000.

**Train loss.** The training loss is given by minus the free entropy (see Equation (159)). Thus, we simply plug-in the order parameters obtained from the fixed-point iteration into the argument of Equation (211):

$$-\epsilon_t = -m\hat{m}^\top - \frac{1}{2}V\hat{q}^\top + \frac{1}{2}q\hat{V}^\top + \frac{1}{2}\frac{\hat{q} + \hat{m}\hat{m}^\top}{\tilde{\lambda} + \hat{V}} - \alpha\mathbb{E}_{\Lambda, h_0^\star, \xi}\mathcal{M}_\ell \tag{234}$$

with

$$\mathcal{M}_\ell = \inf_h \left( \frac{1}{2} \left[ -2h\sigma(h) + \sigma(h)q\sigma(h) \right] + \frac{1}{2}(h - Y)^\top V^{-1}(h - Y) \right). \tag{235}$$

$Y$ is defined above. The expectation is again estimated using Monte-Carlo integration, with $N_{\text{samples}} = 10^7$ samples, and the minimization problem is solved in the same manner as the proximal operator. Recall that we ignored the $\|\mathbf{x}\|^2 = d + \mathbb{E}[\lambda^2]$ term that dominates the error. This is the reason why we compare differences of losses in Figure 2. Additionally, to match the empirical loss Equation (200), one additionally divides $\epsilon_t$ by $\alpha$, as the empirical loss is divided by $n$ and not $d$.

**Test loss.** The test loss (or generalization error) is defined in Equation (156). We recall that the pre-activation is written $h = \frac{\mathbf{w}^\top \mathbf{x}}{\sqrt{d}}$. The specific loss that we consider is given by Equation (5). Expanding this loss and considering the $d \to \infty$ limit, we have

$$\epsilon_g = \mathbb{E}_{\lambda,h_0,h_0^\star} \left[ -2h\sigma(h) + q\sigma(h)^2 \right] \tag{236}$$

Again, we omit the $\|\mathbf{x}\|^2$ term. From Equation (158), we have that $h = h_0 + m(F_1 h_0^\star + F_2)$ where

$$\begin{pmatrix} h_0^\star \\ h_0 \end{pmatrix} \sim \mathcal{N} \left( 0, \begin{pmatrix} \mathbb{I} & m^\top \\ m & q \end{pmatrix} \right). \tag{237}$$

The expectation is then estimated numerically using Monte-Carlo integration with $N_{\text{samples}} = 10^7$ samples.

**Downstream Task.** The classification error of the downstream task described in Section 6 can be expressed as a function of the order parameters. This leads to the theoretical curve of the rightmost panel in Fig. 2. For simplicity, we again consider the case of one hidden unit ($p = 1$). The loss is defined as

$$\mathcal{L}_{DS} = \frac{1}{4} \mathbb{E}_{\mathbf{x}} \|\text{sign}(\mathbf{x}^\top \hat{\mathbf{w}}) - \text{sign}(\mathbf{x}^\top \mathbf{v}^\star)\|^2. \tag{238}$$

We consider the $d \to \infty$ limit. Using the definitions of the data model Equation (1) and of the order parameters Equation (23), we have

$$\frac{\mathbf{x}^\top \hat{\mathbf{w}}}{\sqrt{d}} = \lambda m_u + \frac{\nu}{\sqrt{1 + \beta_\nu}} m_v + \frac{\mathbf{z}^\top S \hat{\mathbf{w}}}{\sqrt{d}}. \tag{239}$$

We removed the ERM superscript for readability, write $\beta_\nu = \mathbb{E}[\nu^2]$ and rescale by $d^{-1/2}$ to have terms of order 1. The rescaling does not change the sign. The term $\xi_1 = \frac{\mathbf{z}^\top S \hat{\mathbf{w}}}{\sqrt{d}}$ is a gaussian with mean 0 and variance

$$\text{Var}[\xi_1] = q - \frac{\beta_\nu}{1 + \beta_\nu} m_v^2. \tag{240}$$

Similarly, we find

$$\frac{\mathbf{x}^\top \mathbf{v}^\star}{\sqrt{d}} = \frac{1}{\sqrt{1 + \beta_\nu}} \left( \nu + \frac{\mathbf{z}^\top \mathbf{v}^\star}{\sqrt{d}} \right). \tag{241}$$

In the $d \to \infty$ limit, $\frac{\mathbf{z}^\top \mathbf{v}^\star}{\sqrt{d}} = \xi_2$ with $\xi_2$ a standard Gaussian. $\xi_1$ and $\xi_2$ are correlated with covariance

$$\mathbb{E}[\xi_1 \xi_2] = \frac{m_v}{\sqrt{1 + \beta_\nu}}. \tag{242}$$

Thus,

$$\mathcal{L}_{DS} = \frac{1}{4} \mathbb{E}_{\lambda,\nu,\xi_1,\xi_2} \left[ \left\| \text{sign}\left( \lambda m_u + \frac{\nu}{\sqrt{1 + \beta_\nu}} + \xi_1 \right) - \text{sign}(\nu + \xi_2) \right\|^2 \right], \tag{243}$$

where

$$(\xi_1, \xi_2) \sim \mathcal{N} \left( 0, \begin{pmatrix} q - m_v^2 \frac{\beta_\nu}{1 + \beta_\nu} & \frac{m_v}{\sqrt{1 + \beta_\nu}} \\ \frac{m_v}{\sqrt{1 + \beta_\nu}} & 1 \end{pmatrix} \right) \tag{244}$$

These low-dimensional expectations are again estimated using Monte-Carlo integration. For the results shown in the right panel of Fig. 2, we employ an aggregation procedure as described in Appendix E. In each trial, we draw a batch of $M = 100$

samples of the latent variables and noise $(\lambda, \nu, \xi_1, \xi_2)$. These samples are partitioned into two subsets according to the sign of the teacher term, $\text{sign}(\nu + \xi_2)$. Within each subset, we average the student's pre-activation fields (the argument of the first sign function) to produce a single aggregated field. This yields two aggregated data points per batch, one for the positive class and one for the negative class. The final downstream loss is estimated by averaging over $N = 10^7$ such independent trials.

# E. Numerical details on the experiments

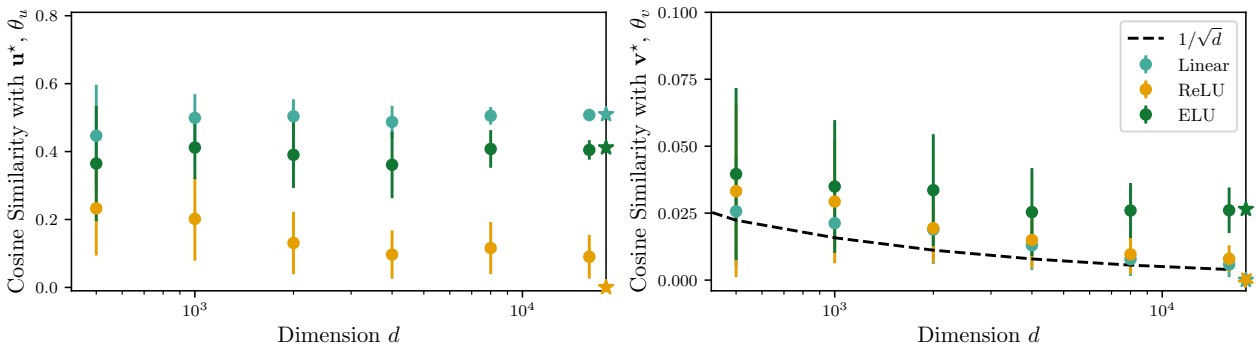

*Figure 3.* Cosine similarity of the ERM estimators with, respectively, $\mathbf{u}^\star$ and $\mathbf{v}^\star$, for different choices of the activation of the autoencoder and dimension $d$. Results are shown at fixed sample ratio $\alpha = n/d = 1.7$ and the same setting as Fig. 1, showcasing the finite-size effects. The dashed line indicates the standard scaling of the cosine similarity with an independent random direction. Points are numerical simulations, and the stars are the predicted value (Section 5), which agree in the high-dimensional limit.

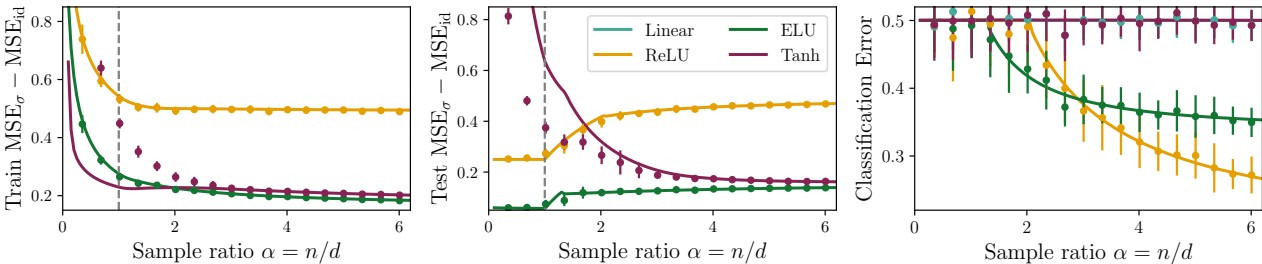

*Figure 4.* Replication of Fig. 2 with the standard $\tanh$ activation. Train loss (left panel) and test loss (center panel) of the non-linear autoencoders minus the corresponding loss of the linear autoencoder, $\sigma = \text{id}$, as a function of the sample complexity. The vertical dashed line is $\alpha_u^{\text{lin}}$. The parameters and $P_{\text{latents}}$ are the same as in Fig. 1. Right panel: Classification error on the downstream task.

We use the spiked cumulant model, Equations (1) and (2), with a latent distribution that has correlation exponent $k^\star = 2$. Concretely, $\lambda \sim \mathcal{N}(0, 1)$ and $\nu = -\sqrt{2}$ if $|\lambda| < \Phi^{-1}(0.75)$ and $\nu = +\sqrt{2}$ otherwise, where $\Phi$ is the standard Gaussian cdf. For each sample ratio $\alpha = n/d$, we generate $n = \lfloor \alpha d \rfloor$ i.i.d. training samples and evaluate the test reconstruction MSE on $n_{\text{test}} = 2 \times 10^6$ i.i.d. samples drawn from the same data model.

Section E.4 considers a different latent distribution with $k^\star = 3$.

**Optimization details.** We optimize Equation (6) using full-batch Adam (Kingma & Ba, 2015). We use $\eta = 0.1$, 800 epochs and no weight decay. We report performance metrics computed from the solution returned by the optimizer at the end of training (final epoch), averaged across instances. An instance corresponds to (i) a fresh random initialization $\mathbf{w} \sim \mathcal{N}(0, \mathbb{I}_d)$ and (ii) a freshly generated training set, while keeping $\mathbf{u}^\star, \mathbf{v}^\star$ and the latent distribution fixed. For each dimension value $d$, we sample $\mathbf{u}^\star, \mathbf{v}^\star \sim \mathcal{N}(0, \mathbb{I}_d)$ once and then normalize so that $\|\mathbf{u}^\star\| = \|\mathbf{v}^\star\| = \sqrt{d}$. We keep these teacher spikes fixed across all instances at that $d$.

**Downstream task.** To construct the downstream task dataset, we draw batches of 100 i.i.d. samples $\{x_\mu\}_{\mu=1}^{100}$ from the data model (Equations (1) and (2)). We split indices into $\mathcal{I}_+ = \{\mu : \mathbf{x}_\mu^\top \mathbf{v}^\star \geq 0\}$ and $\mathcal{I}_- = \{\mu : \mathbf{x}_\mu^\top \mathbf{v}^\star < 0\}$. We form

two averaged inputs $x_+ = \frac{1}{|\mathcal{I}_+|} \sum_{\mu \in \mathcal{I}_+} x_\mu$ and $x_- = \frac{1}{|\mathcal{I}_-|} \sum_{\mu \in \mathcal{I}_-} x_\mu$, yielding two labeled pairs $(x_+, +1), (x_-, -1)$. We resample the batch until collecting a total of $10^4$ independent labeled pairs. For the simulations of Fig. 2, we compute the averaged classification error for each trained autoencoder ($\mathbf{w}$) on the collected labeled samples. The error bars correspond to one standard deviation.

All experiments use the same data-generation and optimization described above; deviations are specified in the corresponding figure captions.

### E.1. Finite-size effects at fixed sample ratio

In Fig. 1 of the main text, we observe the presence of finite-size effects for $\alpha \lesssim 2$. Figure 3 assesses the effect of the dimension $d$ value by probing the cosine similarities, $\theta_u$ and $\theta_v$, at fixed sample ratio $\alpha = 1.7$. The overall pattern shows that the error bars, one standard deviation across 30 instances, reduce as we increase the $d$ value, while the points reach the predicted high-dimensional limit (stars) at moderate $d$ values. The overlap with the PCA-invisible spike $\mathbf{v}^\star$ (right) exhibits the expected $d^{-1/2}$ scaling (cosine similarity with an independent random direction) below the weak-recovery threshold $\alpha_{\text{weak}}^\sigma$ for Linear and ReLU activations.

### E.2. $\tanh$ **scaling with the dimension size** $d$

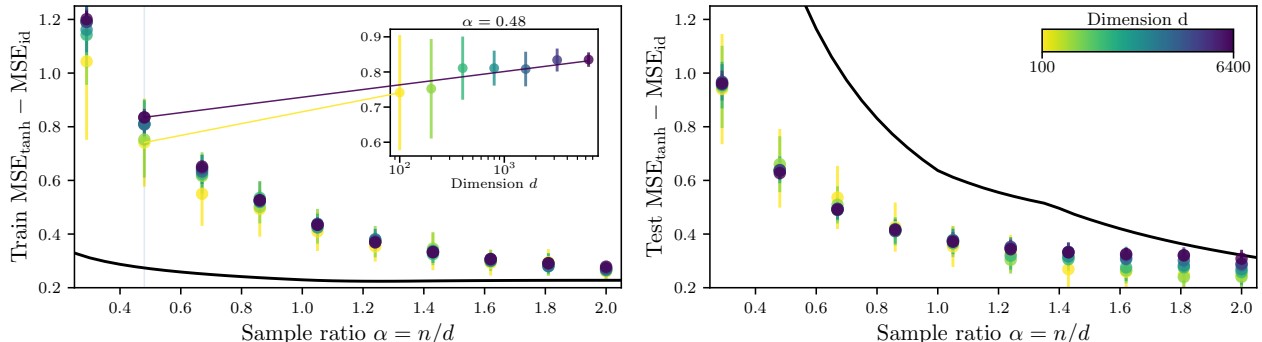

*Figure 5.* Train loss (left panel) and test loss (right panel) of the non-linear autoencoder with $\sigma = \tanh$ activation minus the corresponding loss of the linear autoencoder, $\sigma = \text{id}$, as a function of the sample ratio $\alpha$. Colored dots depend on the dimension $d$. Inset: scaling of the training loss at fixed $\alpha = 0.48$ with respect to $d$. Solid line corresponds to the theoretical prediction of ERM (Section 5).

If instead of plotting $\tanh(\cdot/4)$ (Fig. 2 of the main text), we plot $\tanh$ (Fig. 4), we see a discrepancy between simulations and the asymptotic ERM predictions at low-$\alpha$. Figure 5 tests whether the low sample ratio mismatch for tanh is a finite-size artifact by varying $d \in [100, 6400]$. Both the train and test losses show only a small drift with $d$ and its error bars decrease at moderate dimensions, indicating that the mismatch at small $\alpha$ is unlikely to be explained solely by finite-size effects (for the tested experiments).

Figure 6 complements the analysis of the $\tanh$ activation at low sample ratio ($\alpha = 0.48$). Each curve corresponds to an independent run with a fresh random initialization and an independently generated training dataset. We see that the training trajectories are qualitatively consistent across runs, showing no evidence of training instabilities in this regime for the chosen optimizer.

Overall, the empirical objective is highly nonconvex and exhibits multiple competing minima; correspondingly, the replica-symmetric ERM prediction can deviate from what stochastic gradient based optimization finds. A refined analysis, including replica-symmetry breaking, is left as future work.

### E.3. Additional activations for correlation exponent $k^\star = 2$

Figure 7 extends Fig. 1 to additional nonlinearities. In particular, we notice that GELU, Swish and Sigmoid weakly recover the $\mathbf{v}^\star$ spike as predicted from the theory in the main text, Table 1, when the correlation exponent is $k^\star = 2$. See also Appendix A.4 for the $He_2 + He_1$ activation.

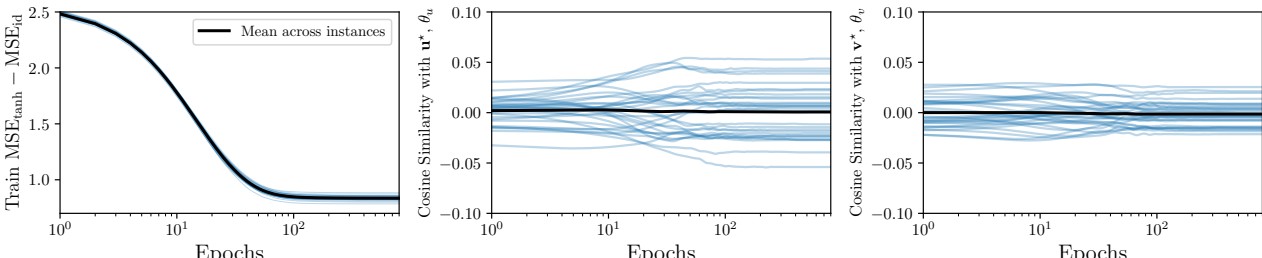

*Figure 6.* Left: Train loss of the non-linear autoencoder with $\sigma = \tanh$ activation minus the corresponding loss of the linear autoencoder, $\sigma = \mathrm{id}$, during training. Center and Right: signed cosine similarity of the autoencoders weights, without the absolute value of the definition, with the $\mathbf{u}^\star$ and $\mathbf{v}^\star$ spikes ($\theta_u$, $\theta_v$). Simulations at fixed sample rate $\alpha = 0.48$, $d = 6400$ and 30 instances. Each blue line corresponds to a single instance, different initialization and training dataset, and the black line is the mean of them. Gray fill represents one standard deviation at each epoch.

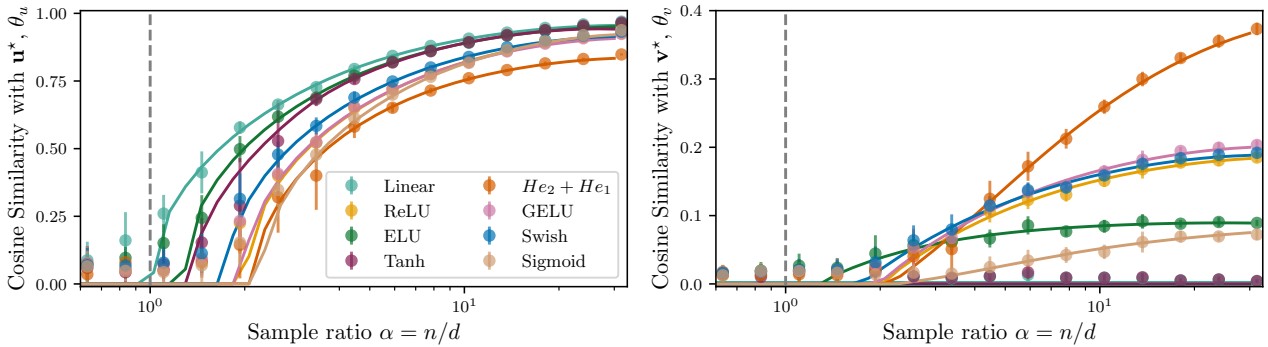

*Figure 7.* Cosine similarity of the ERM estimators with, respectively, $\mathbf{u}^\star$ and $\mathbf{v}^\star$, for different choices of the activation of the autoencoder. We use the same $P_{\text{latents}}$ as in Fig. 1. Colored dots are numerical simulations of minimization with full-batch Adam, with learning rate $\eta = 0.1$ over 800 epochs and no weight decay ($d = 1000$, averaged over 10 instances, error bars represent one standard deviation). Solid lines are theoretical predictions obtained following the procedure described in Appendix D. Vertical dashed line is $\alpha_u^{\text{lin}}$. We see additional activations that weakly recover the $\mathbf{v}^\star$ spike, with $He_2 + He_1$ being the one with the largest overlap. We use standard definitions: $\text{Swish}(x) = x\,\text{sigmoid}(x)$, and $\text{GELU}(x) = x\,\Phi(x)$ ($\Phi$ is the Gaussian cdf). For $He_2 + He_1$, we use probabilists' Hermite polynomials.

## E.4. Experiments for correlation exponent $k^\star = 3$

In Figure 8 we consider a different latent distribution with correlation exponent $k^\star = 3$, for which: $\lambda \sim \mathcal{N}(0, 1)$ and $\nu = \text{sign}(\lambda)\,\text{sign}(|\lambda| - \sqrt{2\ln 2})$. This choice ensures $\mathbb{E}[\lambda\nu] = \mathbb{E}[\lambda^2\nu] = 0$ while $\mathbb{E}[\lambda^3\nu] \neq 0$. The red dashed line of the right panel is the cosine similarity value with an independent random direction from which, if $\theta_v$ is below it, we cannot empirically assess from the simulations whether we have weak recovery of $\mathbf{v}^\star$ or not. As predicted from the theory in the main text (Table 1), at large enough sample ratio $\alpha$ the autoencoder weights with tanh and sigmoid activations start to weakly recover the $\mathbf{v}^\star$ spike, whereas Linear and ReLU activations do not.

## E.5. Experiments for multiple hidden neurons

In this section, we provide additional experiments in the spike cumulant model with correlation exponent $k^\star = 2$ but adding a second neuron in the bottleneck. Extending the architecture of the main text (Eq. 4):

$$\hat{\mathbf{x}}_\mu = \frac{\mathbf{w}_1}{\sqrt{d}}\sigma\left(\frac{\mathbf{w}_1^\top \mathbf{x}_\mu}{\sqrt{d}}\right) + \frac{\mathbf{w}_2}{\sqrt{d}}\sigma\left(\frac{\mathbf{w}_2^\top \mathbf{x}_\mu}{\sqrt{d}}\right) \tag{245}$$

with $\mathbf{w}_1$ and $\mathbf{w}_2$ sampled i.i.d. from $\mathcal{N}(0, \mathbb{I}_d)$ at initialization. Different instances could in principle align to different spikes, hence we will keep track of the maximum cosine similarity: $\theta_u = \max(\theta_u(\mathbf{w}_1), \theta_u(\mathbf{w}_2))$. Another metric we keep track is

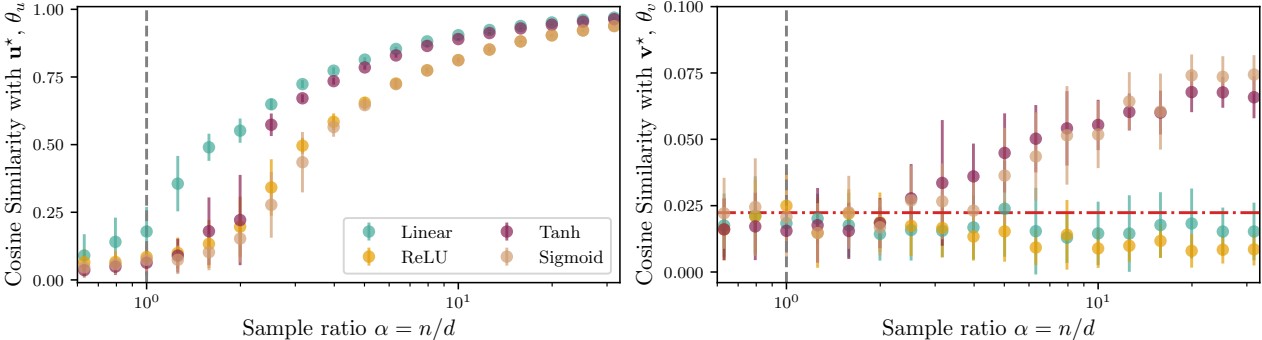

*Figure 8.* Cosine similarity of the ERM estimators with, respectively, $\mathbf{u}^\star$ and $\mathbf{v}^\star$, for different choices of the activation of the autoencoder. We use the same $P_{\text{latents}}$ as in Section E.4. Colored dots are numerical simulations of minimization with full-batch Adam, with learning rate $\eta = 0.1$ over 1200 epochs and no weight decay ($d = 2000$, averaged over 20 instances, error bars represent one standard deviation). Vertical dashed line is $\alpha_u^{\text{lin}}$. Horizontal dashed line at $d^{-1/2}$ shows the scale of the cosine similarity with an independent random direction, from which we can empirically evidence that autoencoder weights with ReLU and Linear activations cannot weakly recover the $\mathbf{v}^\star$ spike, whereas tanh and sigmoid do. This behavior is predicted from Table 1 in our specific latent distribution setting (correlation exponent $k^\star = 3$).

the cosine similarity between both weights defined as:

$$\tilde{\theta} = \frac{|\mathbf{w}_1^\top \mathbf{w}_2|}{||\mathbf{w}_1|| \, ||\mathbf{w}_2||} \tag{246}$$

Figures 9-10 display the total and relative (with respect to the linear activation) train and test MSE for single and double hidden neurons architectures. Overall, increasing the number of neurons decreases the error for all tested activations while reducing the generalization (lower cosine similarities in Fig. 11).

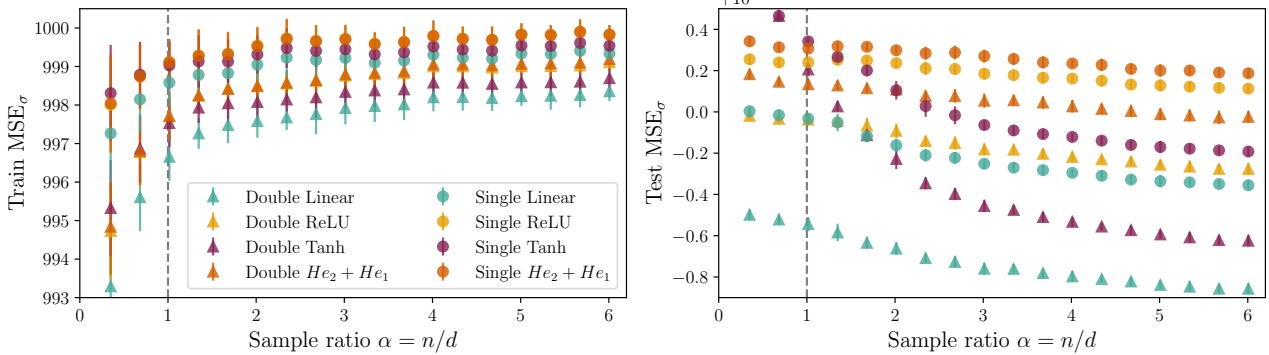

*Figure 9.* Train loss (left panel) and test loss (right panel) of the non-linear tied autoencoders as function of the sample complexity with single (circles) or double (triangles) hidden units. Simulations run at $d = 2000$ and averaged with 30 instances. We see that increasing the number of units reduces the train and test error for all the tested activations.

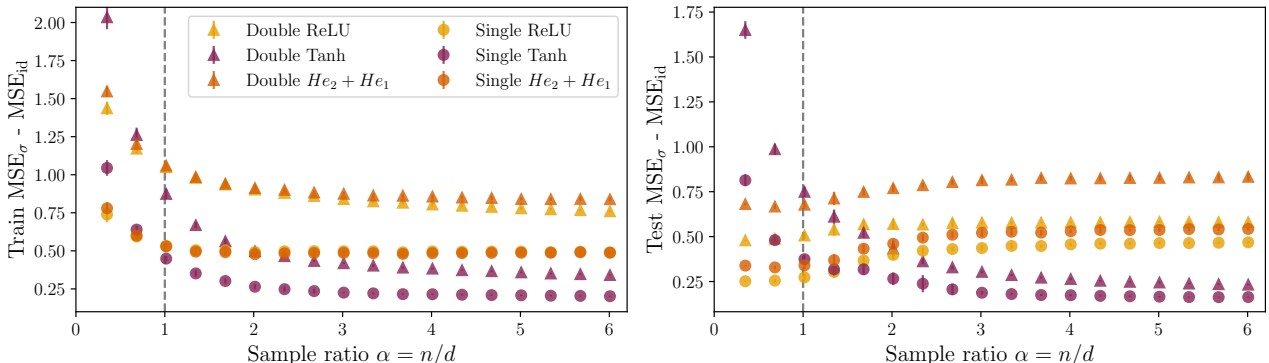

*Figure 10.* Train loss (left panel) and test loss (right panel) of the non-linear tied autoencoders minus the corresponding loss of the linear autoencoder for single (circles) or double (triangles) hidden units. Simulations run at $d = 2000$ and averaged with 30 instances. For the tested activations, increasing the number of neurons increases the gap between the linear and non-linear activation even when the total loss is lower.

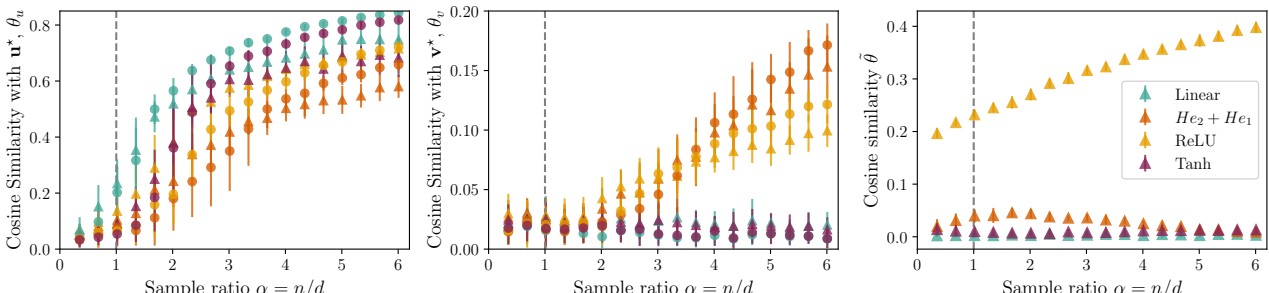

*Figure 11.* Cosine similarity with respect to the teacher spikes for single (circles) and double (triangles) hidden units architectures. Notice that the cosine similarity marginally decreases by increasing the number of neurons. Right panel shows the alignment between the two autoencoder weights with different activations.

