# OpenReview forum: "A Solvable High-Dimensional Model Where Nonlinear Autoencoders Learn Structure Invisible to PCA While Test Loss Misaligns With Generalization"
_ICML.cc/2026/Conference — ICML 2026 regular_

### Official Review · Reviewer_kERy · 2026-03-10

**Soundness:** 3
**Presentation:** 2
**Significance:** 3
**Originality:** 3
**Overall Recommendation:** 5
**Confidence:** 3

**Summary:**

This paper investigate the capabilities of minimal nonlinear autoencoders in capturing latent structures that elude linear models. Specifically, the authors employ a spiked cumulant model parameterized with $u^\ast$ and $v^\ast$. In the data generation condition on the paper, only $u^\ast$ is visible on the linear autoencoder (PCA), and $v^\ast$ is inherently undetectable. The study establishes the conditions for weak recovery on $u^\ast$ and $v^\ast$ using minimal nonlinear architecture. Furthermore, the findings also reveal that while linear model may yield superior test reconstruction loss, they fail to identify higher-order structure of the model, which denotes that reconstruction loss is not always a robust metric for evaluating the quality of learned representations.

**Compliance With Llm Reviewing Policy:**

Affirmed.

**Final Justification:**

Authors have adequately addressed my concerns, and I have raised my score.

**Key Questions For Authors:**

1. Can you explain more in detail about the connection between representation quality of the latent variable in the autoencoder model is with the weak recovery property (cosine similarity)? Cosine similarity neglects the magnitude of vectors, which may include important information on the data.

1. Can you extend your results to autoencoders with multiple channels (i.e., $w \in \mathbb{R}^{d \times m}, m > 1$)?

1. Can you extend your results on regularized autoencoder models, e.g., Variational Autoencoders or Wasserstien autoencoders?

**Limitations:**

yes

**Strengths And Weaknesses:**

Strengths

* The paper provides a rigorous theoretical framework about the conditions of the weak recovery on the latent parameters. The theoretical recovery property of linear model and nonlinear model with various active functions are also well supported by the empirical evidences (Fig 1 and 2).

Weaknesses

* The weak recovery proven in the paper is a asymptotic property for $d \to \infty$. The result of the proven theorems may be hard to be applied on the case of training with finite-sample data on a fixed architecture

* The paper is focused on the weak recovery (cosine similarity) between the latent parameter $u^\ast$ and $v^\ast$ with ERM estimator w. Practically, however, it is perhaps suboptimal to attempt the recovery of independent parameters through the same estimator $w$ instead of assigning separate estimator for each.

---

> ### Author Rebuttal · Authors · 2026-03-27
>
> We thank the reviewer for the careful reading and detailed comments.
>
> >Weak recovery is an asymptotic property... hard to apply to finite-sample training
>
> We agree that this point should be explained more clearly. Our use of the asymptotic limit is primarily methodological: it is what makes it possible to derive tractable analytical equations for both the population gradient flow and the empirical risk minimization. Hence, the asymptotic regime makes the model solvable, rather than creating the phenomenon itself.
>
> All figures contain simulations for finite $d$ that show high agreement with our asymptotic predictions, corroborating that the asymptotics are relevant already for dimensionalities commonly used in ML practice. Appendix E.1 further shows that empirical overlaps approach the theoretical predictions at moderate $d$, with the main finite-$d$ effects being smoother transitions and larger variability across instances.
>
> >Recovery of independent parameters through the same estimator is perhaps suboptimal
>
> This is a good point, and the restriction is intentional. The paper studies the single-neuron autoencoder, so the question is precisely what a single learned representation can encode under a self-supervised reconstruction objective. The goal is not to design the statistically optimal estimator for each spike separately, but to understand whether one minimal nonlinear representation can simultaneously capture structure that linear reconstruction misses. The generic framework in Appendix B can be readily extended to larger architectures, allowing to study recovery of the two spikes separately.
>
> For completeness, we also included a Bayes-optimal estimator that jointly estimates both spikes as an information-theoretic baseline. What is interesting is that, once we have weak recovery of $\mathbf{u}^\star$, the dependence between the latents can induce weak recovery of $\mathbf{v}^\star$ as well. This is explicit in the AMP analysis, where weak recovery of both spikes occurs at the same threshold whenever the correlation exponent is finite.
>
> >Cosine similarity neglects the magnitude of vectors
>
> In our setting, cosine similarity is the natural quantity to characterize directional alignment with the latent spikes. At the same time, our theory does not ignore magnitude: we explicitly track $m_u^{ER}=\hat{w}^\top u^\star /d$, $m_v^{ER}=\hat{w}^\top v^\star /d$, and $q^{\text{ER}}=\hat{w}^\top \hat{w} /d$ (Eq. 20), where the norm is encoded through $q^{\text{ER}}$. We will revise to make the link more explicit: $\theta_s^\text{ER} = m_{s}^\text{ER}/\sqrt{q^{\text{ER}}}$ for $s \in \{u,v\}$.
>
> >Extension to multiple channels ($m>1$)?
>
> The current paper focuses on $m=1$ in the main text, but the generic replica framework in Appendix B holds for any $m=O(1)$ and planted structure of any finite rank. In the revised version, we will add supporting simulations for $m=2$, matching the number of hidden neurons to the number of spikes.
>
> >Extension to VAEs or Wasserstein autoencoders?
>
> The empirical risk derivation already includes a ridge regularization term on the weights, although all results reported in the main text are obtained at zero regularization. Variational or Wasserstein autoencoders, however, involve different objectives and additional latent-variable regularization.
>
> Specifically, we can adapt recent statistical physics analyses, such as the replica calculations developed in Ichikawa & Hukushima (2025) for the linear VAE, to our non-linear setting and spiked cumulant data model.
> While the underlying mathematical derivation would be relatively similar to our current replica computation, VAEs and Wasserstein autoencoders optimize different generative losses rather than the reconstruction loss studied here. As a result, we expect them to exhibit different phenomenologies and learning transitions (such as posterior collapse).
>
> Because the resulting physical picture would be distinct, we think that characterizing the behavior of these generative architectures falls outside the scope of the present paper. Nevertheless, the mathematical framework and conceptual insights developed here, particularly regarding how to handle non-linearities and higher-order dependencies within the replica method, provide a necessary foundation for such an extension. We completely agree that it is a relevant and exciting direction for future work.
>
> Ichikawa, Yuma, and Koji Hukushima. "High-dimensional asymptotics of VAEs: threshold of posterior collapse and dataset-size dependence of rate-distortion curve." _Journal of Statistical Mechanics: Theory and Experiment 2025.7 (2025): 073402._

---

> > ### Author Rebuttal · Reviewer_kERy · 2026-04-02
> >
> > I thank the authors for their thorough responses to my questions. My concerns have been adequately addressed, and I intend to raise my score to 5.

---

### Official Review · Reviewer_YSA7 · 2026-03-11

**Soundness:** 2
**Presentation:** 4
**Significance:** 2
**Originality:** 3
**Overall Recommendation:** 3
**Confidence:** 3

**Summary:**

The paper analyzes a stylized high-dimensional data model with two latent directions- one that shows up in covariance and one that does not, but is still statistically tied to the first through higher-order dependence. In this setting, the authors show that a linear auto-encoder reduces to PCA and can only recover the visible direction, while a single-neuron nonlinear auto-encoder with a suitable activation can also recover the hidden one. The evidence comes in three layers; an information-theoretic and AMP analysis for weak recovery, a population gradient-flow analysis under a spherical constraint, and an empirical-risk analysis based on replica calculations, together with simulations. The paper also make a second point that linear auto-encoders achieve lower train and test reconstruction loss, yet the nonlinear models that recover the hidden factor do better on the toy downstream classification task.

**Compliance With Llm Reviewing Policy:**

Affirmed.

**Key Questions For Authors:**

- How much of the main result depends on using this very small one-neuron autoencoder, and what do you expect would stay true for a slightly larger model?

- How much does the downstream result depend on labels being aligned with the hidden spike?

- Which claim do you see as the paper’s real contribution? nonlinear recovery beyond PCA, or the failure of reconstruction loss as a validation signal?

**Limitations:**

yes

**Strengths And Weaknesses:**

Soundness:

- The hidden direction is genuinely absent from second-order statistics by construction, so the paper’s comparison is built around a real structural limitation of linear methods.
- The paper distinguishes reasonably clearly between what is proved, what is derived from replica calculations, and what is supported numerically.
- The population-level training analysis gives a mathematically interpretable account of why certain nonlinearities can reach weak recovery.

- The strongest end-to-end claim of the paper rests on mixed evidence, because the rigorous population result is narrower than the broader empirical story.
- The tanh mismatch at low sample ratio matters because it shows that one of the paper’s main analytical approximations is not reliable in all of the regimes emphasized by the paper.
- The main optimization theory only establishes weak recovery in a limited regime, not a full characterization of the final learned representation. This is supported by the paper’s own discussion that Theorem 4.2 only controls dynamics up to weak recovery and not beyond that regime.

Presentation:

- The paper usually states the caveats, but some of the strongest framing comes earlier and more prominently than the limits of the actual analysis.
- The paper’s main takeaway is phrased at the level of nonlinear autoencoders, but the technical analysis is for a single-neuron tied architecture with additional geometric constraints, so the formal scope is narrower than the headline claim.
- The downstream classification task is helpful, but it is still a fairly direct test of hidden-factor recovery rather than a broad measure of representation quality.

Significance:

- The paper gives a clean solvable setting in which nonlinear unsupervised learning accesses information that PCA and linear auto-encoders provably cannot use.
- The paper gives a simple, analyzable example where reconstruction loss can point to the wrong model.

- The paper’s significance is mainly conceptual, since it clarifies a phenomenon rather than introducing a method with broad practical use.
- The significance would be stronger if the same misalignment appeared across a broader range of architectures or data models rather than in one carefully designed example.
- The work gives a strong diagnosis of one failure mode, but not yet a broader framework for deciding which unsupervised objectives preserve useful latent structure

Originality:

- The correlation-exponent viewpoint is a useful way to organize what kind of latent dependence is needed for recovery.
- The paper turns a broad intuition about nonlinear representation learning into a specific solvable example with multiple analytical angles.

- Again, the paper is best understood as a sharp special case, not as a broad characterization of when nonlinear auto-encoders outperform linear ones.

---

> ### Author Rebuttal · Authors · 2026-03-27
>
> We thank the reviewer for the detailed feedback.
>
> >Q: How much depends on the one-neuron architecture?
>
> The single hidden unit was chosen for mathematical tractability of the population dynamics. However, the qualitative phenomenology extends to larger models. The Bayes-optimal and empirical risk analyses in Appendix B are carried out for any $k = O(1)$ hidden spikes and autoencoders with any $p = O(1)$ tied weights. It can be empirically checked that with two hidden units the qualitative behavior is consistent with the main text; we will add additional simulations in the revised version.
>
> >Q: How much does the downstream result depend on label alignment with the hidden spike?
>
> The alignment is a core design choice. The downstream task illustrates how self-supervised methods can be evaluated without direct access to the data-generating process. In our theoretical analysis we can measure cosine similarities with the teacher spikes directly, but in practice this cannot be done. The downstream task models the realistic scenario where one has only indirect access to latent structure through a small labeled dataset. If labels were unrelated to the latent structure, there would be no hope of distinguishing whether the representation captures structure beyond what PCA recovers. We agree the task is deliberately simple, and we view this as a feature: a minimal classification task depending on $\mathbf{v}^\star$ is the cleanest way to isolate whether the learned representation captures structure that reconstruction loss does not reward. A more complex evaluation would conflate multiple effects and obscure the advantage.
>
> >On the lack of a broader evaluation framework
>
> We believe that downstream evaluation is itself a natural consequence of our analysis. The underlying principle aligns with the widely adopted paradigm of pre-training followed by fine-tuning. In practice, it is realistic to assume access to small labeled datasets that are insufficient to train the self-supervised model but can probe whether the representation captures task-relevant structure. We will emphasize this prescriptive aspect more clearly in the revision.
>
> > Which claim is the paper's real contribution?
>
> Both are central. The first, that nonlinear autoencoders provably recover latent structure invisible to PCA, demonstrates advantage of nonlinearity in a minimal, fully analyzable setting. The second, the misalignment between test loss and representation quality, addresses a question of practical relevance, as self-supervised and generative pipelines still widely use validation loss for model selection.
>
> >On the mixed evidence for the end-to-end claim
>
> The broader claims rest not merely on empirical findings, but on exact theoretical predictions via the replica method, cross-validated with simulations. The agreement between multiple independent analytical approaches (information-theoretic analysis, AMP, population gradient flow, replica-based ERM) strengthens the reliability of the overall picture.
>
> >On the tanh mismatch at low sample ratio
>
> As discussed in Appendix E.2, the mismatch arises from the highly nonconvex landscape exhibiting multiple competing minima, causing the replica-symmetric prediction to deviate from gradient-based optimization. A refined analysis incorporating replica-symmetry breaking would be needed to resolve this. Importantly, the mismatch is localized to tanh in a specific regime (low $\alpha$) and does not affect the main conclusions.
>
> >On optimization theory limited to weak recovery
>
> We focused on dynamics up to weak recovery for two reasons. First, this transition is the genuinely high-dimensional phase: once weak recovery is achieved, the dynamics become effectively low-dimensional. Second, the loss expansion (Eq. 16) breaks down beyond this regime, requiring case-by-case analysis with population losses that typically lack closed-form expressions, leading to results comparable in nature and scope to the ERM study in Section 5.
>
> >On significance across broader architectures
>
> The key ingredients underlying the misalignment, namely latent structure encoded in higher-order statistics and the gap between reconstruction fidelity and representation quality, are not artifacts of our construction but broadly present in real data. Our model is intentionally minimal to allow exact analytical treatment, following the tradition in high-dimensional statistics where solvable models first isolate a phenomenon before extensions to richer scenarios follow. We view this as the foundational case; extending to multi-neuron architectures and other self-supervised objectives is an exciting direction discussed in the conclusion.
> >On the formal scope being narrower than the headline claim
>
> We appreciate this and will revise the framing accordingly.

---

> > ### Author Rebuttal · Reviewer_YSA7 · 2026-04-03
> >
> > The rebuttal improves the explanation and I appreciate the clarity, but it does not change the main limits of the paper. The core positive result is still shown for a very specific one-neuron tied autoencoder, the training theory only proves early weak recovery in an idealized constrained setting, and the finite-sample ERM story still depends on an assumption that the paper itself says may fail in part of the reported regime.

---

> > > ### Author Response · Authors · 2026-04-05
> > >
> > > We thank the reviewer for acknowledging the improved clarity. We would like to address the three remaining concerns.
> > >
> > > **On the one-neuron architecture:**
> > > The single nonlinear neuron is the minimal architecture sufficient to recover the hidden structure. Studying larger architectures for this data model would not reveal additional mechanisms or improve performance on the core phenomenon. The value of the single-neuron analysis is precisely that it identifies the minimal ingredients enabling recovery of higher-order structure. Our framework in Appendix B already covers $p = O(1)$ hidden units and rank-$k$ planted structure, and we will add multi-neuron simulations in the revision to confirm that the phenomenology is unchanged.
> > >
> > >
> > > **On only proving weak recovery:**
> > > The weak recovery threshold is the fundamental high-dimensional phase transition, and characterizing it is the standard contribution in related literature. High-impact works such as the BBP transition (Baik et al., 2005) for spiked covariance models and the information-theoretic thresholds in Lelarge & Miolane (2017) focus precisely on this threshold as their central result. Our population analysis identifies conditions on the activation function under which weak recovery is or is not achievable, which is the natural question in our setting. Yet closer to our setting, Bardone & Goldt, ICML 2024, study a closely related model and similarly focus their entire theoretical analysis on characterizing weak recovery.
> > >
> > > **"the finite-sample ERM story still depends on an assumption that the paper itself says may fail in part of the reported regime "**
> > >
> > > We need to make a clarification here. The validity of the "replica symmetric" assumption is, in our case, explicitly verifiable via the replicon condition (see Vilucchio et al., 2025). In the revised version, we will evaluate this condition, thus explicitly identifying regimes where it holds and where it does not. The numerical agreement between the theory and GD heuristically indicates regimes in which it is satisfied. To support the main claims of our paper, we only use those regimes. The regimes in which the replicon condition is not satisfied are inconsequential for the main claim in our paper. This thus does not weaken our main claims.

---

### Official Review · Reviewer_AHLE · 2026-03-12

**Soundness:** 3
**Presentation:** 3
**Significance:** 3
**Originality:** 3
**Overall Recommendation:** 5
**Confidence:** 2

**Summary:**

This paper studies a non-Gaussian spiked model and shows an interesting misalignment between reconstruction loss and representation quality. Despite linear autoencoders achieving lower test loss, they fail to recover identifiable latent factors. In contrast, nonlinear autoencoders can learn the hidden structure using higher-order polynomials.

**Compliance With Llm Reviewing Policy:**

Affirmed.

**Final Justification:**

I am not familar with this area but I think the model and conclusions are interesting.

**Key Questions For Authors:**

What will the main conclusions change when the dataset size and dimension are finite?

**Limitations:**

see weaknesses.

**Strengths And Weaknesses:**

**Strength**

This paper provides a rigorous theoretical framework to prove that non-linear activation functions can help learn good representations while linear autoencoders cannot.

**Weaknesses**
1. The data is deliberately using explicit whitening to hide features. This data model may not coincide with practice.
2. The paper's conclusions highly rely on asymptotic limits ($n,d\to \infty$).

---

> ### Author Rebuttal · Authors · 2026-03-27
>
> We thank the reviewer for their comments and for the positive assessment.
>
> >The data is deliberately using explicit whitening to hide features. This data model may not coincide with practice.
>
> We agree that the role of the whitening step should be clarified better. Our intent is not to claim that explicit whitening matches how real-world data is, but rather to isolate a regime that is well known in statistics and machine learning: latent structure may be invisible to second-order methods, while still being recoverable from higher-order dependencies. Classical approaches such as ICA or tensor-based methods are designed precisely to exploit this type of information once covariance is no longer informative. In that sense, the whitening step is not meant as an artificial trick, but as a clean way to separate structure visible at the covariance level from structure that only appears in higher-order statistics. Our goal is then to understand for a solvable datamodel if a shallow neural network can recover such higher-order structure, and how this depends on the activation function. We will revise the paper to make this motivation and connection to higher-order statistical methods more explicit.
>
> >The paper's conclusions highly rely on asymptotic limits ($n,d\rightarrow \infty$).
>
> We agree that the finite-size interpretation should be made more explicit. Our view is that the high-dimensional limit is primarily a methodological tool, allowing us to obtain tractable analytical equations for both the population gradient flow and the empirical risk minimization problem. In that sense, the asymptotic regime is what makes the model solvable, rather than what creates the phenomenon itself. At the same time, the main qualitative conclusions remain visible at finite dimensions. In appendix E.1, we explicitly study finite-size effects and show that the empirical overlaps approach the theoretical high-dimensional predictions already at moderate $d$.
>
> >What will the main conclusions change when the dataset size and dimension are finite?
>
> Our answer is that the qualitative conclusions do not change, but finite-size effects blur the thresholds and increase variability, especially near the transition. In particular, at finite $d$ one should expect:
> (1) smoother transitions in recovery as a function of $\alpha$, and
> (2) larger fluctuations between instances near the BBP threshold.
> However, the central picture remains the same in the experiments already included in the paper: linear methods recover the covariance-visible structure, while suitable nonlinear activations recover information about the higher-order hidden structure.
> We will revise the paper to make these two points clearer: first, that whitening is a path to isolate the higher-order mechanism rather than a claim of real datamodels; and second, that the asymptotic theory is meant to predict the finite-dimensional phenomenology up to finite-size and optimization effects, which we already showed in appendix E.

---

> > ### Author Rebuttal · Reviewer_AHLE · 2026-04-03
> >
> > Thanks for the clarifications. I will maintain my score.

---

### Official Review · Reviewer_q9Lm · 2026-03-13

**Soundness:** 2
**Presentation:** 3
**Significance:** 2
**Originality:** 3
**Overall Recommendation:** 4
**Confidence:** 3

**Summary:**

This paper introduces a tractable high-dimensional model called the "spiked cumulant model" where nonlinear autoencoders can extract hidden data structures that are invisible to linear methods like PCA.

The model contains two latent factors: one visible to covariance (PCA) and one invisible, requiring higher-order moments. While linear autoencoders achieve lower reconstruction (test) loss, they fail to recover the hidden structure. In contrast, nonlinear autoencoders successfully learn both factors despite higher test loss, demonstrating a clear misalignment between test loss and representation quality.

The paper provides rigorous analysis of population gradient flow and empirical risk minimization, establishing a theoretical framework to study nonlinear representation learning and evaluation in self-supervised settings.

**Compliance With Llm Reviewing Policy:**

Affirmed.

**Key Questions For Authors:**

Could you provide more discussion on Section 6? Moreover, could you elaborate on the rationale for using downstream tasks to evaluate the nonlinear learne

**Limitations:**

Yes.

**Strengths And Weaknesses:**

Strengths:
The concrete model construction is interesting. The primary strength, as summarized earlier, lies in providing a rigorous, solvable theoretical framework to demonstrate the superiority of nonlinear representation learning and the critical misalignment between test loss and generalization.
Weaknesses:
Although I appreciate this work, the proposed "spiked cumulant model" itself appears somewhat restrictive. The findings would constitute a more substantial and convincing contribution if the authors could argue that similar principles—where nonlinear methods uncover structures hidden to linear ones and where loss fails to indicate representation quality—hold in broader, more abstract, or more realistic model classes.

---

> ### Author Rebuttal · Authors · 2026-03-27
>
> We thank the reviewer for their insightful comments and for highlighting what we also see as the core contribution of the paper.
>
> > The primary strength, as summarized earlier, lies in providing a rigorous, solvable theoretical framework to demonstrate the superiority of nonlinear representation learning and the critical misalignment between test loss and generalization.
>
> We are glad this central message came across. Our goal is precisely to introduce a minimal solvable high-dimensional model where standard covariance-based methods such as PCA, and equivalently linear autoencoders, fail to recover a hidden latent factor, while simple nonlinear autoencoders can provably recover it. We analyze this phenomenon both at the population level and at the empirical-risk level in the proportional high-dimensional regime.
>
> > Although I appreciate this work, the proposed "spiked cumulant model" itself appears somewhat restrictive.
>
> We agree that this point deserves to be clarified better in the paper. Our intent is not to present the model as realistic. Instead, we use it as a minimal solvable setting that isolates a broader mechanism: recovery of latent structure through higher order dependence when second order statistics are uninformative. This is already reflected in the paper through the correlation exponent $k^\star$.
> Our model is intentionally minimal to allow exact analytical treatment, following the tradition in high-dimensional statistics where solvable models first isolate a phenomenon before extensions to richer scenarios follow. We view this as the foundational case; extending to multi-neuron architectures and other self-supervised objectives is an exciting direction discussed in the conclusion. We finally remark that our analysis already provides the starting ground for extensions to less restricted settings, since the framework for replica derivation detailed in appendix B is set in a larger class of data models in which planted structure can have any finite rank $k$ and also the autoencoder can have any O(1) number of units.
>
> > Could you provide more discussion on Section 6? Moreover, could you elaborate on the rationale for using downstream tasks to evaluate the nonlinear learne
>
> Our motivation for using a downstream task is the same as in modern self-supervised learning: in general, one does not have access to the hidden true structure of the data, so quantities such as cosine similarity with the teacher spikes cannot be measured in practice. A more realistic evaluation scenario takes inspiration from pre-training followed by fine-tuning pipelines. One assumes access to a large unlabeled dataset for training the autoencoder, together with a small labeled dataset that provides indirect measurements of the latent structure. The labeled set is too small to train the representation itself, but sufficient to probe whether the learned features capture task-relevant information. This is exactly the perspective we wanted to capture in Section 6. In our model, the downstream labels are chosen to depend on the hidden direction $\mathbf{v}^\star$, so the downstream task provides a principled way to test whether pretraining has encoded the true structure of the data. We will expand Section 6 to make this motivation clearer and to better explain why downstream evaluation is the appropriate metric in this self-supervised setting.

---

> > ### Author Rebuttal · Reviewer_q9Lm · 2026-04-03
> >
> > Thank you for your replies. I’m satisfied with them.

---

### Decision · Program_Chairs · 2026-04-30

**Decision:**

Accept (regular)

**Comment:**

This paper is well summarised by one reviewer. A tractable high-dimensional model is introduced, where nonlinear autoencoders can extract hidden data structures that are invisible to linear methods (PCA). The model contains two latent factors: one visible to covariance (PCA) and one invisible, requiring higher-order moments. While linear autoencoders achieve lower reconstruction (test) loss, they fail to recover the hidden structure. In contrast, nonlinear autoencoders successfully learn both factors despite higher test loss, demonstrating a clear misalignment between test loss and representation quality. The paper provides a rigorous analysis of population gradient flow and empirical risk minimization, establishing a theoretical framework to study nonlinear representation learning and evaluation in self-supervised settings.

The reviewers mostly appreciate the explicit formulation of the problem and identify the work as a significant conceptual contribution. One of the reviewers raise concerns with the simplicity of the 1-neuron model, but this may not be a reason for rejection given that the work seems to provide considerable value in providing understanding the benefit of non-linearity in unsupervised/self-supervised models.